# Tumor-derived GDF-15 blocks LFA-1 dependent T cell recruitment and suppresses responses to anti-PD-1 treatment

Immune checkpoint blockade therapy is beneficial and even curative for some cancer patients. However, the majority don't respond to immune therapy. Across different tumor types, pre-existing T cell infiltrates predict response to checkpoint-based immunotherapy. Based on in vitro pharmacological studies, mouse models and analyses of human melanoma patients, we show that the cytokine GDF-15 impairs LFA-1/β2-integrin-mediated adhesion of T cells to activated endothelial cells, which is a pre-requisite of T cell extravasation. In melanoma patients, GDF-15 serum levels strongly correlate with failure of PD-1-based immune checkpoint blockade therapy. Neutralization of GDF-15 improves both T cell trafficking and therapy efficiency in murine tumor models. Thus GDF-15, beside its known role in cancer-related anorexia and cachexia, emerges as a regulator of T cell extravasation into the tumor microenvironment, which provides an even stronger rationale for therapeutic anti-GDF-15 antibody development.

Immune checkpoint blockade has achieved unprecedented durable responses in patients with advanced and metastatic cancer. However, across 75 trials in 29 tumor types, only 1,568 out of 8,692 patients (18%) were classified as responders to anti-PD(L)1 monotherapy[1]. Responses require contact-dependent killing of cancer cells by immune cells. Infiltration of cytotoxic immune effector cells into the tumor microenvironment is thus a prerequisite for successful immunotherapy[2]. The infiltrated ("hot") vs. non-infiltrated ("cold") phenotype of the tumor microenvironment is, however, hardly related to the availability of immunogenic (neo)antigens[3]. Instead, malignant cells often orchestrate a T cell-excluding microenvironment which confers resistance to immunotherapy[4–6]. Further, tumor-infiltrating T cells gradually assume an epigenetically imprinted, irreversibly exhausted state[7]. Successful responses to anti-PD-1 thus depend on freshly immigrating T cells[8]. Roadblocks to T cell infiltration ("T cell repellents") are hence of major interest, both as biomarkers and as promising targets for therapeutic intervention.

While detrimental in cancer, tolerance towards neoantigen-expressing cells is essential during pregnancy. Accordingly, T cells in placenta and decidua are functionally inhibited, and sparse. In mice, they only account for 3% of decidual leukocytes on embryonic day 8.5[9].

T cell extravasation from the well-vascularized feto-maternal interface is strictly regulated. As T cells depend on active leukocyte function-associated antigen (LFA)−1 for adhesion to, rolling on and transmigration across endothelial barriers[10,11], inhibition of the interaction between LFA-1 and its ligand intercellular adhesion molecule (ICAM)−1 rescued challenged pregnancies[12]. Conversely, adoptive transfer of LFA-1-positive T cells induced rejection in abortion-prone mouse models. LFA-1 activity thus represents a critical immune checkpoint in pregnancy. Interestingly, LFA-1-deficient mice can still mount anti-viral immune responses, but lose the ability to clear immunogenic tumors[13]. Conversely, LFA-1 activation enriches tumor-specific T cells in "cold" tumors and synergizes with CTLA-4 blockade[14]. Inhibiting (conformational) LFA-1 activation could thus also enable tumor immune escape.

Growth/differentiation factor-15 (GDF-15, also known as macrophage inhibitory cytokine MIC-1)[15] is a divergent member of the transforming growth-factor beta (TGF-β) superfamily. Between rat, mouse and human, GDF-15 shows <70% sequence conservation[16]. In mice and monkeys, GDF-15 induces anorexia and cachexia[17–19] via the brainstem-restricted receptor Glial Cell Line-Derived Neurotrophic Factor (GDNF) family receptor alpha-like (GFRAL)[20–22]. In humans, the highest physiological GDF-15 expression occurs during pregnancy.

✉e-mail: Wischhusen_J@ukw.de

Low GDF-15 serum levels predict miscarriage[23]. Anorexia or cachexia are, however, rarely observed during pregnancy and unlikely to protect the fetus. Among the various functions of GDF-15, inhibition of dendritic cell-mediated T cell stimulation[24] and induction of regulatory T cell activity[25] might support feto-maternal semi-allograft tolerance. Overexpression of GDF-15 in dendritic cells can even confer cardiac allograft tolerance between BALB/c and C57Bl/6 J mice[26]. In the liver, GDF-15 can prevent myeloid cell activation[27]. In mice subjected to cardiac ischemia-reperfusion injury, induction of GDF-15 (or administration of recombinant human (rh)GDF-15) inhibits LFA-1/β2-integrin activation on polymorphonuclear neutrophils (PMN), thus preventing excessive influx of PMN into the infarcted myocardium[28].

In the tumor secretome, GDF-15 is the most prominently over-expressed cytokine[29]. Elevated GDF-15 levels correlate with poor survival[30], absence of T cell infiltrates[31] and impaired T cell priming[24,26,31]. We thus hypothesized that GDF-15-mediated inhibition of leukocyte integrin activation might extend beyond the previously described effect on murine granulocytes and macrophages[28]. In this study, we show that tumor-derived GDF-15 inhibits the LFA-1/ICAM-1 axis in T cells and T cell migration into tumor tissue. Neutralization of GDF-15 enhances T cell infiltration and efficacy of immune checkpoint blockade in murine GDF-15-expressing tumor models. Most importantly, clinical data from two independent melanoma patient cohorts indicate that elevated GDF-15 serum levels correlate with resistance to PD-1-based immune checkpoint blockade. GDF-15 thus represents a predictive biomarker for failure of immune checkpoint blockade, and a new synergistic target for cancer immunotherapy.

## Results

### GDF-15 inhibits adhesion of human T cells to activated endothelial cells

Assays with anti-coagulated whole blood showed that adhesion of chemokine-stimulated CD45$^+$ immune cells on activated human lymphatic endothelial cells (huLEC) is impaired by GDF-15 (Fig. 1a). Applying a significance level of $p < 0.05$, GDF-15 significantly reduced adhesion of CD4$^+$ and CD8$^+$ T cells stimulated with C-X-C Motif Chemokine Ligand 12α/stromal cell-derived factor 1 (CXCL12α/SDF-1)[32] (Fig. 1b). Focussing on T cells as key effectors of cancer immunotherapy, and applying physiological flow conditions, a brief treatment with rhGDF-15 reduced adhesion of stimulated human T cells on activated huLEC almost to the level observed with unstimulated huLEC, which essentially lack adhesion molecules (Fig. 1c). GDF-15 similarly reduced adhesion and rolling of T cells on activated human umbilical vein endothelial cells (HUVEC) (Fig. 1d, Supplementary Figure 1a). Further, the speed of rolling was slightly, but significantly increased by GDF-15 (Supplementary Figure 1b). Applying GDF-15 to endothelial cells rather than to T cells yielded much smaller effects, compatible with GDF-15 sticking to glycosaminoglycans on HUVEC and thereby being presented to T cells (Fig. 1d)[32]. In the presence of CXCL9 and CXCL10, which can recruit T cells into tumors[33], GDF-15-mediated inhibition of T cell adhesion to HUVEC was also highly significant (Fig. 1e). Antibody-mediated blockade of the brainstem GDF-15 receptor GFRAL did not affect the impact of GDF-15 on T cell adhesion (Fig. 1f). Dose-response curves showed that EC50 and EC90 values for inhibition of T cell adhesion under flow conditions are in a concentration range (Fig. 1g) likely to be achieved in the tumor microenvironment. Phase-contrast microscopy revealed no impact of GDF-15 during transendothelial T cell migration. Lower recruitment thus results from reduced adhesion (Fig. 1h–j). Still, by interfering with T cell adhesion to activated human endothelial cells, GDF-15 may reduce T cell infiltration into tumor tissue.

### GDF-15 interferes with LFA-1-dependent adhesion to immobilized ICAM-1

To test whether GDF-15 impairs activation of Lymphocyte function-associated antigen (LFA)−1 on human T cells (similar to a previously described effect on murine myeloid cells[28]), we ran stimulated T cells over a layer of immobilized intercellular adhesion molecule (ICAM)−1. Adhesion was effectively prevented by GDF-15 or by the blocking anti-LFA-1 antibody TS1/18 (Fig. 2a–c). GDF-15 had, in contrast, no significant effect on T cell adhesion to mucosal vascular addressing cell adhesion molecule 1 (MAdCAM-1) or vascular cell adhesion protein 1 (VCAM-1) (Fig. 2b, c). Adding GDF-15 on top of a blocking anti-LFA-1 antibody showed no additive effect when CD4$^+$ or CD8$^+$ T cells were run over a layer of activated HUVEC (Fig. 2d, e). Thus, GDF-15 impairs T cell adhesion primarily by interfering with the ICAM-1:LFA-1-axis. Surprisingly, flow cytometry failed to show effects of GDF-15 on the binding of the conformation-specific anti-active-LFA-1 antibody mAb24 (Fig. 2f) or an Fc-tagged ICAM-1 complex (Fig. 2g) to CD8$^+$ T cells from healthy donors or cancer patients. A quantitative assessment of ICAM-1-Fc or mAb24 binding to CD8$^+$ T cells on a single-molecule level by *direct* stochastic optical reconstruction microscopy (*d*STORM)[34,35] still revealed a slight, but significant impairment of CXCL12-induced LFA-1 activation (Fig. 2h–j). Adhesion, however, requires *ligand binding and a stable linkage* of LFA-1 to the actin cytoskeleton. The latter is triggered by mechanical tension resulting from interactions with immobilized (but not soluble) ICAM-1[36]. We thus immobilized ICAM-Fc and E-selectin-Fc on beads, added activated T cells ±GDF-15, and lysed these cells after 7 min. This revealed a much lower level of Talin Ser$^{425}$ phosphorylation in GDF-15-treated T cells (Fig. 2k, l). As phosphorylation is required to relieve Talin auto-inhibition[37], GDF-15-treated T cells cannot establish a stable link between LFA-1 and the cytoskeleton[38]. GDF-15 thus modulates adhesion of T cells to the endothelium mainly by interfering with the intracellular stabilization of extracellular LFA-1:ICAM-1 interactions. Notwithstanding effects on other immune cell subsets, GDF-15 is therefore likely to affect both T cell homing to lymphoid organs and T cell trafficking to tumor tissue.

### GDF-15 broadly affects T cell adhesion to endothelial cells

To identify the T cell subsets affected by GDF-15, purified immune cells isolated from healthy donors were briefly exposed to GDF-15 or anti-LFA-1 antibody (TS1/18) before being run over a layer of activated HUVEC. Adhesion was induced by CXCL12α. Strong effects of GDF-15 or anti-LFA-1 were found for central or effector memory CD4$^+$ T cells, and for naive, memory, effector memory or pan CD8$^+$ T cells. Effects on pan or naive CD4$^+$ T cells, which express lower levels of LFA-1[39], became significant when the sample size was expanded. With the exception of CD8$^+$ T$_M$ cells, effects of GDF-15 were generally similar to those of the anti-LFA-1 antibody (Supplementary Fig. 2a–h). Activated and expanded, >90% Foxp3$^+$ T$_{reg}$ showed moderately reduced adhesion to huLEC upon GDF-15 treatment (Supplementary Fig. 2i). However, GDF-15 enhances the suppressive capacity of T$_{reg}$ by stabilizing Foxp3[25]. The spectrum of GDF-15-responsive cell types is thus aligned with the known anti-inflammatory function of GDF-15. Of note, individual assays showing no effect of GDF-15 were not associated with a general lack of response in the respective donor. As GDF-15 is prone to adhere on plastic, a loss of active GDF-15 during such assays appears more likely.

### GDF-15 interferes with T cell trafficking and PD-1-based immune checkpoint blockade in the sub-cutaneous MC38 colon cancer mouse model

While GDF-15 is highly overexpressed in about 50% of solid human tumors (Supplementary Figure 3), elevated GDF-15 levels are rare in murine tumor cell lines. The Open Access Crown Biotech Oncology Databases (https://www.crownbio.com/oncology/oncology-databases) indicate a median of 45.9 fragments per kilobase of exon model per million reads mapped (FKPM) for *gdf15* across 544 patient-derived cell lines, of 18.4 FKPM across 1141 established human cancer cell lines, but of only 1.9 FKPM across 125 murine tumor cell lines. To enable experiments with an anti-human GDF-15 antibody, we selected

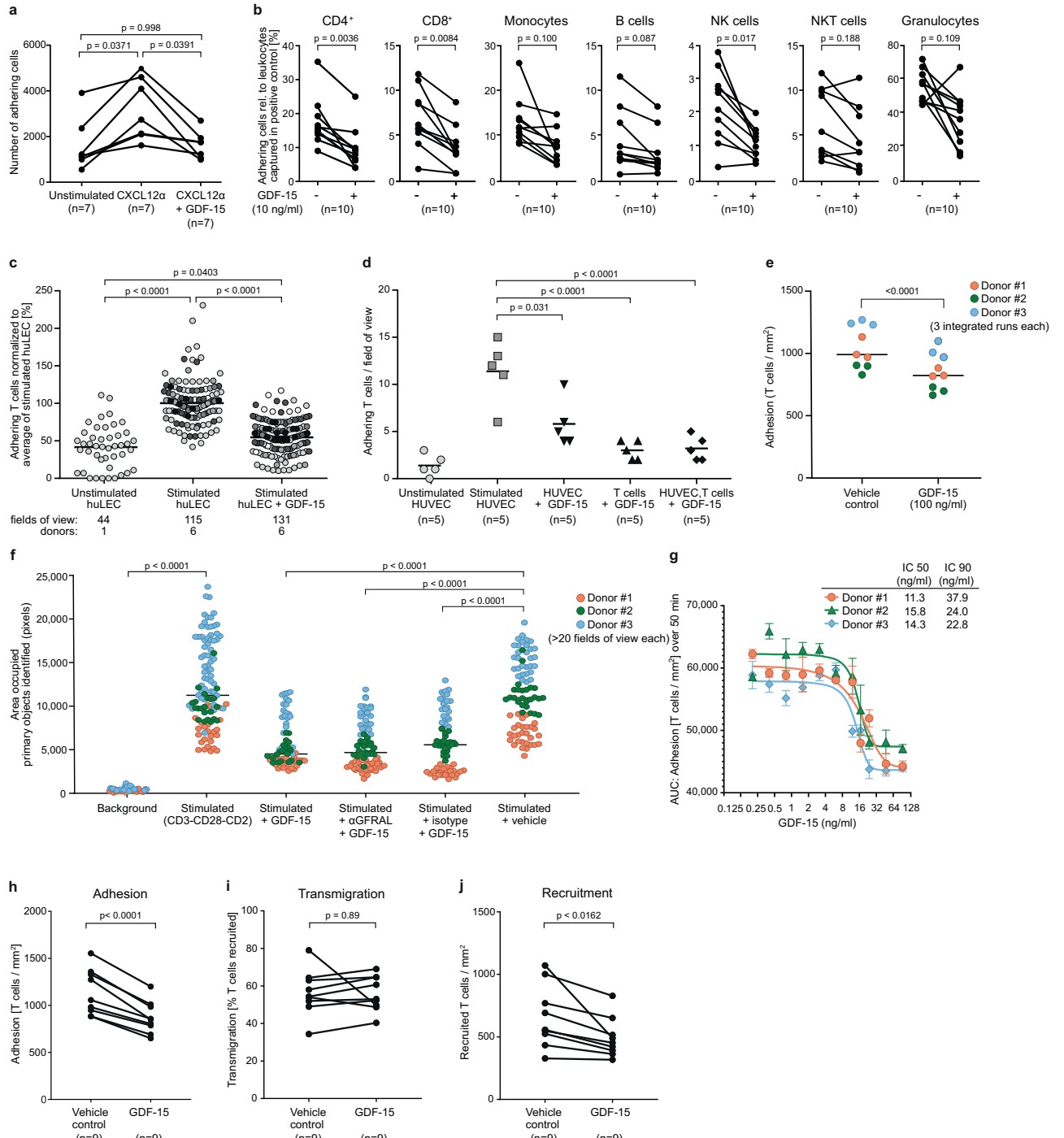

**Fig. 1 | GDF-15 interferes with T cell adhesion to activated endothelial cells.**
**a** Effects of recombinant human (rh)GDF-15 (added for 10 min) on CXCL12α
−mediated adhesion of whole blood-derived CD45⁺ cells to activated human lym-
phatic endothelial cells (huLEC) were analyzed. Adherence was calculated based on
the number of CD45⁺ cells as enumerated by flow cytometry ($n = 7$ experiments). In
**b**, adhering leukocytes were further characterized by multicolor staining ($n = 10$
experiments). **c** Stimulated T cells from 6 donors were treated or not with rhGDF-15
for 20 min before being run in μ-slides over a layer of activated huLEC. 10 pre-
defined fields of view were video-imaged for 5 s and the number of T cells adhering
under hydrodynamic flow conditions was counted. Different shadings indicate
different T cell donors. For reference, adhesion to non-activated huLEC is shown for
one donor. **d** Enumeration of T cells adhering to human umbilical vein endothelial
cells (HUVEC). 5 predefined fields of view per sample were analyzed in a repre-
sentative experiment. In **e**, CXCL9 and CXCL10 were used to induce adhesion of

untreated or GDF-15-treated CD8⁺ T cells from 3 different donors on stimulated
huLEC. In **f**, stimulated CD8⁺ T cells from 3 different donors were treated with
rhGDF-15 and anti-GFRAL or isotype control antibodies for 20 min before being run
in μ-slides over a layer of activated huLEC. In **g**–**j**, phase-contrast microscopy in
chamber slides to assess effects of rhGDF-15 on T cell adhesion to activated HUVEC.
An EC₅₀ value for rhGDF-15-mediated adhesion inhibition on pan T cells from 3
different donors was determined in **g** (logICF=logIC50 + (1/HillSlope)*log(F/(100-
F)). **h**–**j** Using pan T cells from 9 different donors, effects of rhGDF-15 on T cell
adhesion (**h**), transmigration (**i**) and recruitment (**j**) were analyzed. Statistical ana-
lyses were performed by one-way ANOVA in **a**, **d**, **f**, by two-sided paired Student´s *t*-
tests in **b**, **e**, **h**, **i**, **j**, by mixed-effects analysis in **c**. To correct for multiple compar-
isons, Tukey´s post hoc test was applied in **a**, **c**, **d**, Bonferroni´s method in **b**. In
**c**, **d**, **g**, mean values with SEM, in **e**, **f**, median values are indicated as horizontal lines.
Source data are provided as a Source Data file.

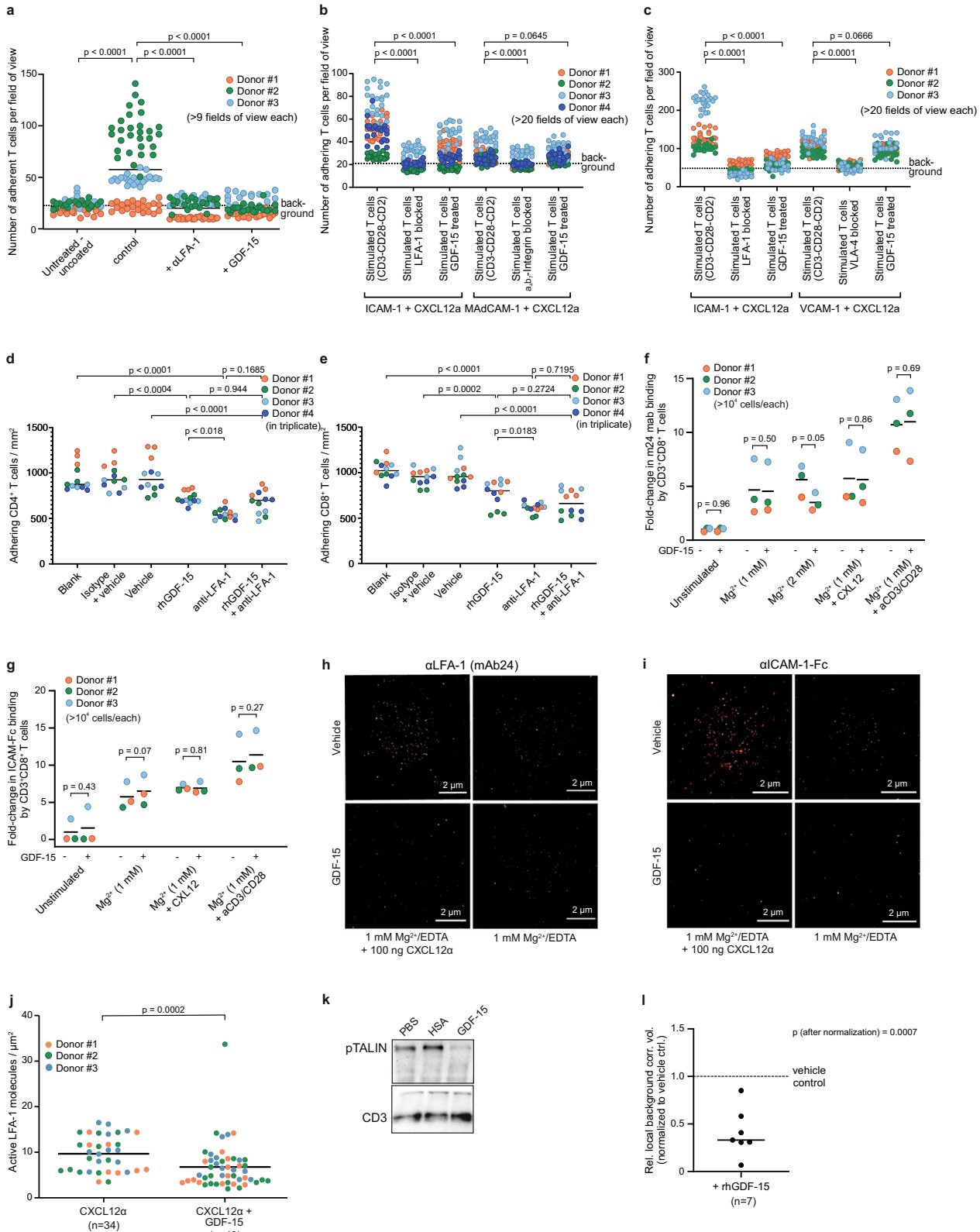

the MC38 colon cancer cell line, which showed no detectable expression of murine GDF-15 under standard conditions. These cells had been instrumental for the development of PD-1 and CTLA-4 inhibitors[40]. MC38 cells were transfected with either human *gdf15* or empty control vector. Resulting GDF-15 expression levels were similar to those observed in human cancer cell lines (Supplementary Fig. 3c). In NCI[nu/nu]-mice, MC38 cells expressing transgenic human GDF-15 (MC38[tghGDF-15]

cells) induced cachexia (Fig. 3a) and grew more slowly than control-transfected MC38[blank] cells (Fig. 3b). In immunocompetent C57BL/6NCrl-mice, however, MC38[tghGDF-15] cells trended towards a higher tumor take rate (96% vs. 80%, *p* = 0.133, Fig. 3c) and showed a significantly faster tumor growth than MC38[blank] control groups across four experiments. A representative Kaplan–Meier plot is shown in Fig. 3d (also compare Fig. 3l, Table S1 and Supplementary Fig. 4). These

**Fig. 2 | GDF-15 interferes with LFA-1-dependent adhesion of human T cells.**
**a**–**c** μ-slides were coated with CXCL12α and vehicle or ICAM-1-Fc (**a**–**c**), MAdCAM-1-Fc (**b**), or VCAM-1-Fc (**c**). Stained primary human T cells were stimulated with anti-CD3/CD28 before GDF-15, or vehicle, or antibodies against adhesion molecules LFA-1, α4β7 integrin, or VCAM-1 were added for 30 min. T cells were perfused for 6 min over the coated μ-slides. Adhesion was recorded by live microscopy and analyzed using CellProfiler software. In **d**, **e**, CD4[+] and CD8[+] T cells were pre-treated for 20 min with GDF-15, or blocking anti-LFA-1 antibody TS1/18, or both, and run over activated HUVEC as in (**1e**–**i**). **f**, **g** Binding of conformation-specific anti-active LFA-1 antibody mAb24 (**f**) or ICAM-Fc (**g**) to CD8[+] T cells was analyzed. Whole blood from healthy volunteers was maintained at 37 °C and treated or not with GDF-15 10 min prior to LFA-1 activation. Fluorescence-conjugated antibodies and complexed soluble ICAM-1-Fc were added for another 10 min. Cells were fixed and analyzed on an Attune Nxt flow cytometer. Mean fluorescent intensity (MFI) values were normalized to control conditions by z-transformation. **h**, **i**, **j** Human PBMC were stimulated for 30 min with CXCL12α and Mg[2+] ± rhGDF-15. Cells were stained with the conformation-specific Alexa Fluor 647-labeled anti-LFA-1 antibody mAb24 (**h**) or hICAM-1-Fc-AF647 (**i**). The number of active LFA-1 molecules per single CD3[+] T cell was quantified by *direct* stochastic optical reconstruction microscopy. Representative single cell images are shown in **h**, **i**. Data obtained with mAb24 across three different donors are summarized in **j**. **k**, **l** T cells were added to ICAM-1- and E-Selectin-coated Protein G beads, in the absence or presence of GDF-15. After lysis, Talin phosphorylation was assessed by Western blotting, with CD3ε as loading control. A representative blot is shown in **k**. Protein quantification data from 7 different samples normalized to vehicle (human serum albumin) control are displayed in **l**. Statistics were calculated by Kruskal–Wallis with Dunn´s post hoc test (**a**), by one-way ANOVA with Tukey´s correction for multiple comparisons (**b**–**e**), and by two-sided paired *t*-tests (**f**, **g**, **j**, **l**). Horizontal bars indicate mean (**a**, **f**, **g**) or median (**d**, **e**, **j**, **l**) values. Source data are provided as Source Data file.

MC38[tghGDF-15] tumors gave rise to heterogeneous hGDF-15 serum levels (Fig. 3e), not exceeding the second quartile of the levels observed in human melanoma patients[30]. Correlations between GDF-15 serum levels and tumor size were weaker than expected. Some mice with large tumors showed hardly detectable hGDF-15 levels (Fig. 3f). Measured GDF-15 serum levels were instead negatively linked to the development of endogenous antibodies against human GDF-15 (Fig. 3g), which likely precluded ELISA-based detection of human GDF-15 in about 50% of the inoculated mice (Fig. 3h). Introducing a human transgene and a resistance gene, which both provide additional targets for spontaneous immune responses, thus entailed caveats. Still, immunosuppressive effects of human GDF-15 apparently prevented antibody formation in 50% of the mice, and immunosuppressive effects of human GDF-15 more than outweighed its immunogenicity in mice. Immunohistochemistry from tumor-bearing mice euthanized after 28 to 33 days showed that transgenic human GDF-15 reduced spontaneous CD8[+] T cell infiltration ($p = 0.024$) without altering the total number of tumor-infiltrating leukocytes ($p = 0.91$) (Fig. 3i–k). We then used a proprietary anti-GDF-15 antibody that binds human GDF-15 with high ($K_D \leq 0.06$ pM) and murine GDF-15 with intermediate affinity ($K_D \leq 6$ nM). Its neutralizing function was first confirmed by its ability to prevent GDF-15-induced cancer cachexia in nude mice xenograft models (Supplementary Fig. 4a). Antibody-mediated blockade of GDF-15 partly reverted the growth advantage of MC38[tghGDF-15] cells (Fig. 3l). While the number of tumors that could be harvested and assessed by flow cytometry after day 23 was too small to allow for valid conclusions, effects on immune cell infiltration still showed a trend (Fig. 3m–o). To test the impact of GDF-15 under T cell-stimulating conditions, mice were co-treated with an anti-PD-1 antibody (clone RMP1-14) and anti-GDF-15. The immunohistochemical evaluation of infiltrating CD8[+] T cells from this experiment indicated a synergistically enhanced T cell recruitment by anti-PD-1 and anti-GDF-15 (Fig. 3p, q), with significantly ($p = 0.0033$) increased CD8[+] T cell counts next to necrotic tumor areas (Fig. 3r). Strikingly, blank-transfected MC38 tumors were eradicated in 9/10 mice by anti-PD-1 antibody treatment (clone RMP1-14) (Figs. 3s and S4c). MC38[tghGDF-15] tumors, in contrast, resisted anti-PD-1 treatment in 9/9 mice ($p \leq 0.0001$). Adding anti-GDF-15 to anti-PD-1, however, resulted in complete clearance of MC38[tghGDF-15] tumors in 4/10 mice (Fig. 3t and Supplementary Fig. 4d). Cox proportional hazard models confirmed that anti-GDF-15 and anti-PD-1-based immune checkpoint blockade synergize to increase survival in mice over the effect of anti-PD-1 monotherapy (hazard ratio (HR): 0.28, 95% CI 0.08-0.96, $p = 0.044$).

### GDF-15 blockade improves T cell trafficking and cancer immunotherapy in the orthotopic Panc02 pancreatic cancer mouse model
While the MC38[tghGDF-15] model proved valuable to show effects of GDF-15 overexpression and of an anti-human GDF-15 antibody, spontaneous antibodies against human GDF-15 were a confounding factor. As GDF-15 is critical for engraftment of orthotopically implanted Panc02 pancreatic cancer cells[41], we tested whether neutralization of GDF-15 could sensitize established Panc02 tumors towards PD-1-based immunotherapy. We thus inoculated $1 \times 10^4$ Luciferase-expressing Panc02 cells into the head of the pancreas (Fig. 4a), or injected $1 \times 10^6$ Panc02-Luc cells into the pancreas tail (Fig. 4b). Tumor-bearing animals were randomized by bioluminescence in vivo imaging on day 5, and subsequently treated with either anti-PD-1, or anti-GDF-15, or both, or isotype control antibody. In both settings, combination treatment induced tumor regressions (Fig. 4a, b). Taking either the value on day 29 or the last bioluminescent measurement as endpoint, only the combination treatment achieved a statistically significant survival advantage over vehicle in the aggressive "pancreas head" model (Fig. 4a). In the "pancreas tail" model there were no early drop-outs. However, as the $p$ value lost significance after correction for multiple testing (Fig. 4b), the pancreas head model was chosen to assess T cell infiltration. Flow cytometry on day 12 after treatment revealed that anti-GDF-15 tended to enhance CD4[+] and CD8[+] T cell infiltration, with statistical significance in the treatment-responsive combination therapy group (Fig. 4c, d). No significant differences were seen for myeloid cell infiltration (Supplementary Fig. 5). Immunohistochemical stainings further confirmed a higher abundance of intratumoral Granzyme B-positive CD8[+] T cells and only scarce Foxp3[+] T cells upon combination treatment (Fig. 4e).

### GDF-15 blockade improves T cell trafficking to tumor-draining lymph nodes in the orthotopic EMT6 breast cancer mouse model
While T cell infiltration into the tumor is a prerequisite for successful immune checkpoint blockade, T cell dynamics in tumor-draining lymph nodes is critically important for the priming of anti-tumor immune responses[42]. With LFA-1 being also implicated in T cell trafficking to lymph nodes, we asked whether GDF-15 might also interfere at this level. Unfortunately, the network of interlobular lymphatic vessels in the pancreas is so complex, that there is no standardized classification of pancreatic nodes. Hence, we addressed this question in the orthotopic EMT6 breast cancer model, where lymph node dynamics have already been successfully studied[43]. Tumor-bearing mice were treated with anti-GDF-15 on day 6, 9, and 12 after tumor implantation. CFSE-labeled T cells were adoptively transferred on day 13. 24 h later, axillary and brachial lymph nodes were explanted. Anti-GDF-15 treatment indeed enhanced the trafficking of CD4[+] and CD8[+] T cells to tumor-draining lymph nodes (Fig. 4f, g). Caliper-based assessments of tumor size showed no effect of anti-GDF-15 on the palpated tumor size in the mammary fat pad (Fig. 4h). However, weighing of explanted tumors suggested a reduced tumor mass after anti-GDF-15 treatment (Fig. 4i).

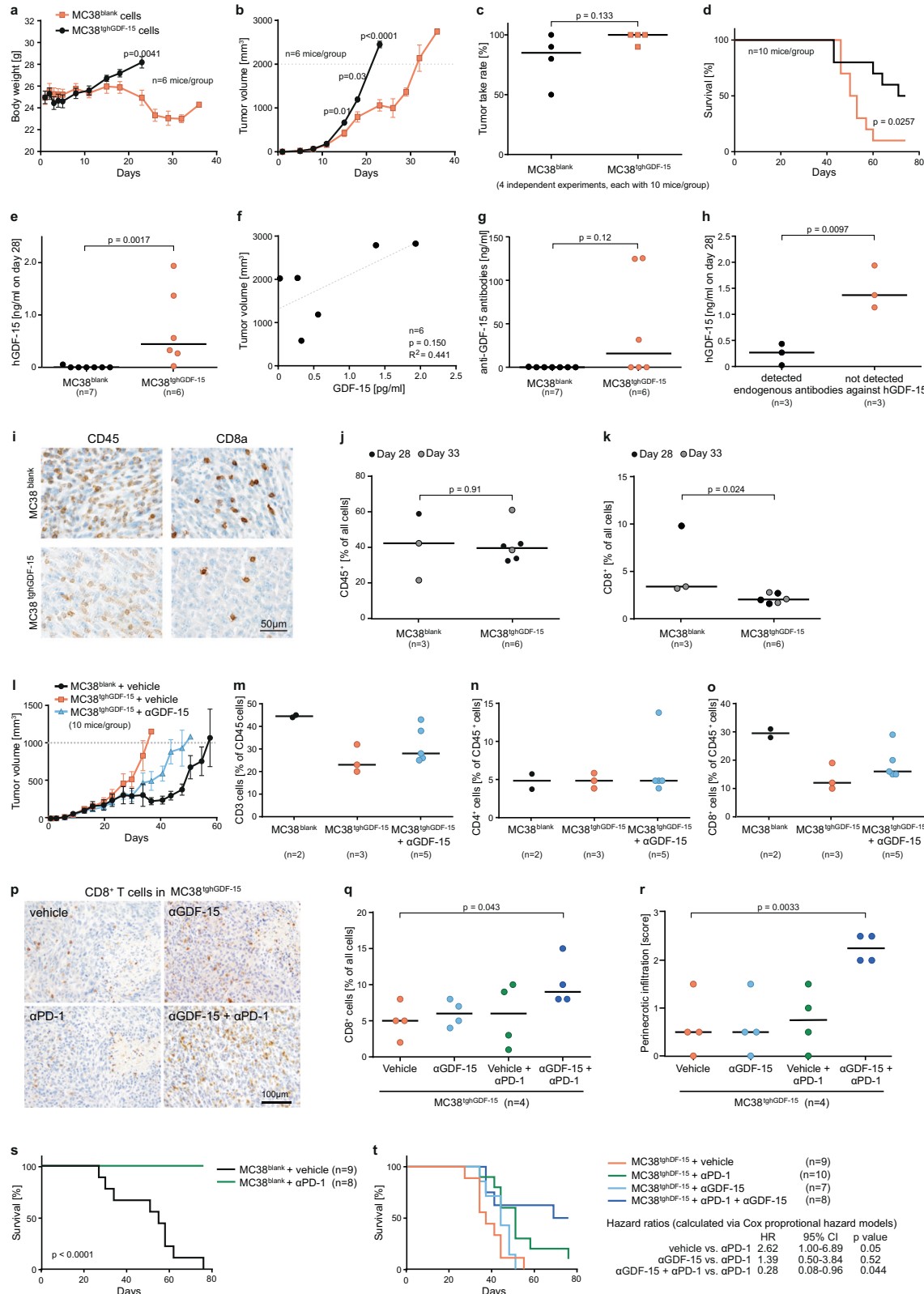

## Humanized mice show enhanced immune cell recruitment in the presence of a blocking anti-GDF-15 antibody

To assess anti-GDF-15 treatment against human cancer in vivo, we reconstituted NOD/SCID/γc$^{-/-}$/FcγR$^{-/-}$ mice with human umbilical cord-derived CD34$^+$ human hematopoietic stem cells. While T cells develop properly in these humanized mice, dendritic cells are largely lacking[44]. Hence, effects of human-specific biologicals on T cell trafficking can be

studied in the absence of endogenous antibody formation or tumor rejection. In these mice, GDF-15 expression of the inoculated patient-derived HV-18-MK melanoma transplant resulted in similar GDF-15 serum levels as in human melanoma patients (Fig. 4j)[30]. Analyses from three isotype control-treated mice suggested a correlation between GDF-15 serum levels and tumor size in this model (Fig. 4k). Due to the overall low TIL numbers and the resulting sampling bias,

**Fig. 3 | GDF-15 interferes with immune infiltration and immune-mediated tumor rejection in the MC38 colon cancer model in vivo. a, b** NCI[nu/nu]-mice (6 mice/group) were subcutaneously injected with $5 \times 10^5$ MC38[blank] or MC38[tghGDF-15] colon cancer cells. Body weight (**a**) and tumor sizes (**b**) were determined twice weekly. **c–h** C57BL/6NCrl-mice were subcutaneously injected with $5 \times 10^5$ MC38[blank] or MC38[tghGDF-15] cells. **c** Tumor take rates. **d** Representative survival curves (termination criterion: tumor volume > 1200 mm³) shown as Kaplan–Meier plot (detailed statistics in Supplementary Table 1). **e** hGDF-15 serum levels on day 28 were analyzed by ELISA and **f** correlated with tumor size. **g** Antibodies against hGDF-15 were assessed in sera and **h** correlated with hGDF-15 levels. **i–t** C57BL/6NCrl-mice were subcutaneously injected with MC38[blank] or MC38[tghGDF-15] cells. In **i–k**, tumors were explanted when ≥1000 mm³. Tumor-infiltrating CD45⁺ and CD8⁺ cells were stained (**i**) and quantified (**j, k**). In **l–o**, effects of anti-hGDF-15 antibody on tumor growth (**l**), and infiltration by CD45⁺ (**m**), CD4⁺ (**n**) and CD8⁺ (**o**) cells are shown. Infiltration was analyzed by flow cytometry from gently dissociated, 300–500 mm³-sized tumors. Preferential rejection of MC38[blank] tumors and tumor-unrelated deaths caused

imbalances between the groups. In **p–r**, MC38[tghGDF-15] tumors were treated with vehicle/anti-hGDF-15/anti-PD-1/anti-hGDF-15+anti-PD1. Representative pictures (**p**), the percentage of tumor-infiltrating CD8⁺ T cells (**q**) and a score for perinecrotic CD8⁺ T cell infiltration (**r**) are shown. **s, t** C57BL/6NCRL mice were subcutaneously inoculated with $5 \times 10^5$ MC38[blank] or MC38[tghGDF-15] cells. Mice were randomized across the different treatment groups and treated or not with anti-PD-1 antibody ± anti-GDF-15 antibody (**t**). Kaplan–Meier plots based on survival are displayed. **a, b, c, h, q**, and **r** were analyzed by two-sided unpaired *t*-tests, **d** and **s** by log-rank (Mantel-Cox) test. Wilcoxon–Mann–Whitney test was applied to **e, g, j**, and **k**. Pearson's linear regression was calculated in **f**. In **t**, tumor growth was compared by Cox proportional hazard models. Individual tumor growth curves for **s** and **t** are shown in Supplementary Fig. S4. Horizontal bars depict mean values ± SEM in **a, b, l**, and median values in **c, e, g, j, k, m, n, o, q, r**. All experiments were performed at least three times. In **a, b, d–t**, representative experiments are shown. Source data are provided as a Source Data file.

immunohistochemical analyses were inferior to flow cytometry-based infiltration assessments. These showed that an ADCC-deficient humanized anti-GDF-15 antibody significantly enhanced the number of tumor-infiltrating CD45⁺ cells ($p = 0.0374$) and the proportion of CD3⁺ T cells in the infiltrate ($p = 0.0106$), while CD19⁺ ($p = 0.3087$) cells included as a less LFA-1-sensitive cell population were not increased (Fig. 4l, two independent experiments with a total of 9 mice per group). T cell subset analyses performed from 4 mice/group further showed that treatment with anti-GDF-15 significantly enhanced infiltration of CD4⁺ T cells ($p = 0.037$) and enabled de novo infiltration of CD8⁺ T cells, which were completely excluded from tumors in three out of four isotype-treated control mice ($p = 0.069$) (Fig. 4m). GDF-15 blockade thus enables human effector T cells to infiltrate patient-derived melanomas in humanized mice.

## GDF-15 expression and T cell infiltration are inversely correlated in human melanoma brain metastases and in human oropharyngeal squamous cell carcinomas (OPSCC)

Having established a link between GDF-15 and T cell infiltration in wild-type and in humanized mice, we assessed GDF-15 expression and immune infiltrates in tissue microarrays from melanoma brain metastases[45]. Stainings for CD3⁺ and CD8⁺ T cells as well as for the GDF-15 pro-domain (which adheres to the extracellular matrix) are shown in Fig. 5a. A GDF-15 score based on the percentage of GDF-15-positive cells and their respective staining intensities revealed highly significant negative correlations between GDF-15 expression and CD3⁺ and CD8⁺ T cell infiltration ($n = 70$) (Fig. 5b, c). For Foxp3⁺ T_reg, a weaker, though still significant correlation was found (Fig. 5d). As infiltrating CD8⁺ T cells are a well-established, good prognostic factor in melanoma[46], these findings may explain the strong association between GDF-15 expression and poor survival in this disease[30].

Similar associations between immune infiltration and survival have also been described for oropharyngeal squamous cell carcinoma (OPSCC)[47,48]. When GDF-15 serum levels were assessed from patients with OPSCC and correlated with CD8⁺ T cell infiltration data (available for 37 HPV⁺ OPSCC), a significant inverse correlation between GDF-15 serum levels and CD8⁺ T cell infiltration was found (Fig. 5e). HPV⁻ OPSCC are, in contrast, poorly immunogenic and hardly infiltrated by T cells[49]. GDF-15 serum levels could therefore not be correlated with immune infiltration in this subset. Still, disease-specific survival data from 86 patients with OPSCC showed that elevated GDF-15 serum levels are associated with poor survival across the whole OPSCC cohort (Fig. 5f). Associations between GDF-15 and survival were, however, non-significant in patients with (poorly immunogenic) HPV⁻ ($n = 32$) (Fig. 5g), but highly pronounced in patients with more immunogenic HPV⁺ ($n = 54$) OPSCC (Fig. 5h).

## Elevated GDF-15 predicts failure of anti-PD-1 antibodies and poor survival in melanoma patients treated with immune checkpoint blockade

Having shown GDF-15 to interfere with immunotherapy in vivo, we explored its potential predictive role for tumor immune escape in humans. Therefore, we analyzed pre-treatment serum samples from patients with metastatic melanoma at baseline of immune checkpoint blockade. Clinical follow-up according to RECIST1.1 criteria[50] showed that GDF-15 serum levels were similar between responders to anti-CTLA-4/ipilimumab ($n = 11$, median: 2.2 ng/ml) and non-responders ($n = 26$, median: 1.9 ng/ml) ($p = 0.59$, Fig. 6a). However, all 5 durable responders to ipilimumab showed GDF-15 levels ≤2.0 ng/ml, while none of the 6 initial responders with GDF-15 levels ≥2.0 ng/ml had a durable response over >1 year (median [GDF-15]: 1.05 ng/ml in patients with durable response vs. 7.42 ng/ml in non-survivors, range: 0.75–1.84 ng/ml in durable responders vs. 2.19–29.8 ng/ml in non-survivors, $p = 0.0043$, Fig. 6b). In a first (Zurich-based) cohort of patients treated with the anti-PD-1 antibody pembrolizumab (Keytruda™), responders (median 1.2 ng/ml, $n = 19$) showed significantly lower GDF-15 serum levels than non-responders (median: 4.6 ng/ml, $n = 10$, Fig. 6c). No single patient with GDF-15 levels ≥4.1 ng/ml responded to therapy. A logistic regression model based on these 29 patients estimates a 60% decrease in the probability of response to treatment per 1 ng/ml increase in serum GDF-15 (OR = 0.39, 95% CI [0.16,0.70], $p = 0.01$, Fig. 6d). Serum levels of lactate dehydrogenase (sLDH), which are linked to tumor mass[51] and inversely associated with response to anti-PD-1 treatment[52], were a weaker predictor (OR = 0.997, 95% CI [0.994,0.999], per U/ml LDH, $p = 0.0727$) (Supplementary Fig. 6a). According to Akaike´s information criterion, additional consideration of sLDH did not even improve the GDF-15-based prediction of response. Consequently, patients with higher GDF-15 serum levels were more likely to die from melanoma (HR = 1.27, 95% CI [1.10;1.47] per ng/ml GDF-15, $p = 0.001$). Survival from the onset of anti-PD-1 treatment was thus poor in patients with high GDF-15 (≥2 ng/ml, Fig. 6e). Next-generation sequencing of tumor tissue ($n = 40$) as well as TCGA database mining confirmed that GDF-15 mRNA levels are unrelated to mutational load ($p = 0.5$ for both test cohort and TCGA-based analysis, Supplementary Fig. 6b, c). Further, 3 out of 4 TCGA data sets show no significant correlation between GDF-15 and PD-L1 expression, while one cohort indicates a weak inverse correlation (Supplementary Fig. 6d). As a marker for the failure of anti-PD-1 therapy, GDF-15 is hence independent from PD-L1 or tumor mutational burden. A second (Tübingen-based) cohort with pre-treatment serum samples from 88 melanoma patients treated with the anti-PD-1 antibodies pembrolizumab or nivolumab confirmed the negative correlation between GDF-15 serum levels and success of anti-PD-1 treatment. Again, disease control ($p = 0.0085$) and response ($p = 0.0315$) were more likely to be

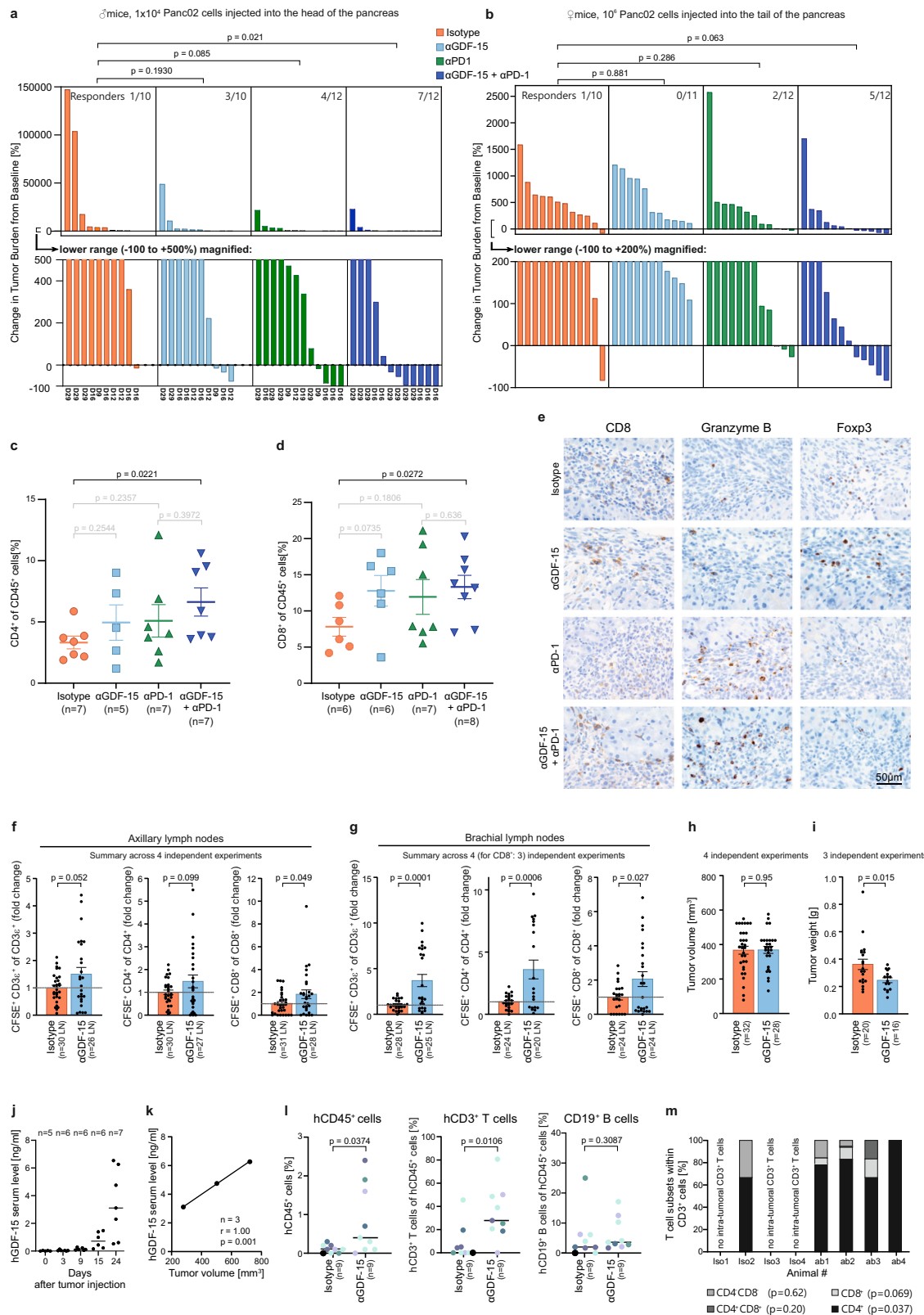

observed in patients with low GDF-15 serum levels, while none of the patients with elevated GDF-15 levels showed a lasting complete response to therapy (Fig. 6f). Unlike the patient collective from Zurich, this Tübingen cohort also comprises non-responders with low GDF-15, indicating the presence of other escape mechanisms. Still, a logistic regression model confirmed the significant negative association between GDF-15 and the probability of response or disease control.

With regard to survival, the huge dynamic range of GDF-15 serum levels in this cohort (0-1640 ng/ml) results in an apparently low hazard ratio, as the HR describes the change in survival probability per integer change in GDF-15 level measured in ng/ml (Fig. 6g, h, Supplementary Table 2). Still, a dichotomized multi-variate analysis (Supplementary Fig. 7) indicates a hazard ratio of 3.312 (95%CI: 1.617–6.783) for GDF-15 serum levels ≥2.0 ng/ml, which are found in 28/88 patients. Multi-

**Fig. 4 | GDF-15 blockade synergizes with anti-PD-1 in orthotopic Panc02 tumors and enhances T cell recruitment in syngeneic and humanized mice.** In **a**, **c**, **d**, **e**, $1 \times 10^4$ luciferase-transgenic Panc02 cells (Panc02-luc) were inoculated into the pancreatic head of male albino C57Bl/6J mice. In **b**, female albino C57Bl/6J mice received $1 \times 10^6$ Panc02-Luc cells into the pancreatic tail. After bioluminescence-based randomization on day 5, animals were treated twice weekly with vehicle/anti-hGDF-15/anti-PD-1/anti-hGDF-15+anti-PD1. Bioluminescent in vivo imaging (1–2 times/week) enabled monitoring of tumor growth (counts$_{final\ measurement}$/counts$_{randomization}$). In **a**, animals were euthanized when symptomatic or on day 29 (see x-axis), in **b** on day 35. CD4$^+$ (**c**) and CD8$^+$ (**d**) T cell infiltration on day 12 were assessed by flow cytometry from disseminated tumors. **e** Representative stainings for tumor-infiltrating CD8$^+$, Granzyme B$^+$ and Foxp3$^+$ cells. **f–i** $2 \times 10^5$ EMT6 murine breast cancer cells were orthotopically injected in female 7–10-week-old BALB/c mice. Anti-GDF-15 or isotype control were administered on days 6, 9, 12. Carboxyfluorescein-succinimidylester (CFSE)-labeled T cells were adoptively transferred on day 13. On day 14, infiltration of transferred CD3$^+$, CD4$^+$ and CD8$^+$

T cells into (explanted) axillary (**f**) and brachial (**g**) lymph nodes was assessed by flow cytometry. Tumor size (**h**) and weight (**i**) were recorded. **j–m** NOD/SCID/γc$^{-/-}$FcRγ$^{-/-}$ mice were reconstituted with human hematopoietic stem cells (HSC), injected with patient-derived HV-18-MK (GDF-15$^{high}$) melanoma transplants, and treated with anti-hGDF-15 or control antibody. Day 24 hGDF-15 serum levels (**j**) correlated with tumor size for the 3 isotype-treated tumors that could be explanted without being disrupted (Pearson correlation) (**k**). Tumor-infiltrating human CD45$^+$, CD3$^+$, and CD19$^+$ cells were determined by flow cytometry. Two independent experiments ($n = 9$ mice/group) are summarized in **l**. Colors relate to CD34$^+$ HSC donors. In **m**, the composition of CD3$^+$ T cell immune infiltrates on day 24 is shown for the mice from the second independent experiment. In **a**, **b**, tumor growth was analyzed by pairwise Mann–Whitney tests with Bonferroni–Holm correction. In **c**, **d**, **f**, **g**, **h**, **i**, groups were compared by unpaired Student´s $t$-test. In **l** and **m**, Mann–Whitney $U$-test was performed, using overall cell percentages in **m**. Horizontal bars indicate mean ± SEM in **c**, **d**, **f**, **g**, **h**, **i**, median values in **j**, **l**, **m**. Source data are provided as a Source Data file.

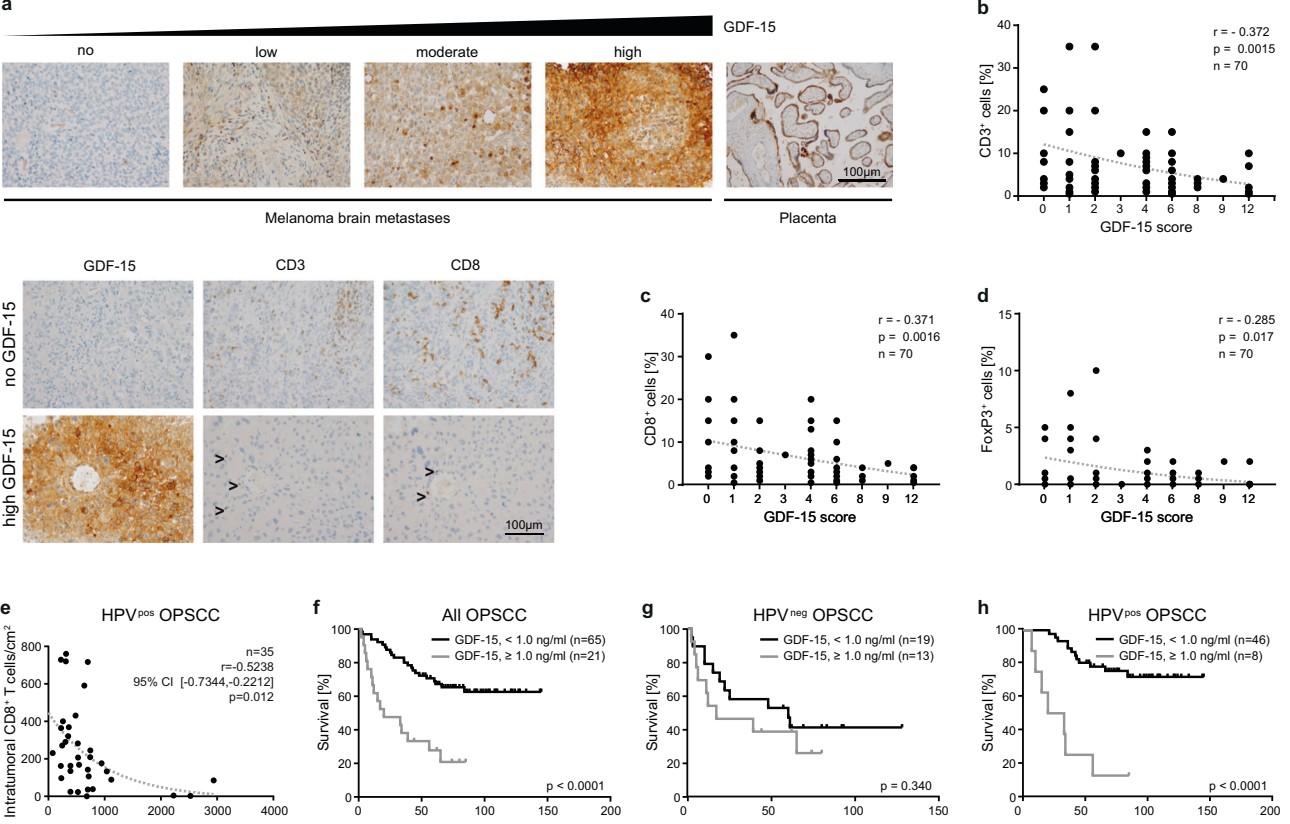

**Fig. 5 | GDF-15 expression is negatively correlated with intratumoral T cell infiltration in brain metastases from melanoma patients and in HPV$^+$ oropharyngeal squamous cell carcinomas. a–d** Formalin-fixed paraffin-embedded tissues from melanoma brain metastases from 70 patients were stained by immunohistochemistry for GDF-15 and for the T cell marker proteins CD3 and CD8. Exemplary stainings are shown in **a**. **b–d** For hGDF-15 expression, a score was based on frequency (0–1% → score 0; 1–10% → score 1; 10–25% → score 2; 25–50% → score 3; >50% → score 4) multiplied with staining intensity (weak → 1, moderate → 2, strong → 3)). **e–h** GDF-15 serum levels were assessed from patients with human papilloma virus (HPV)$^+$ or HPV$^-$ oropharyngeal squamous cell carcinomas (OPSCC).

Tumor-infiltrating CD8$^+$ T cells per area were quantitated for 37 HPV$^+$ tumors and correlated with the corresponding GDF-15 serum levels (**e**). **f–h** GDF-15 serum levels were divided into two groups with either GDF-15 < 1.0 ng/ml or GDF-15 ≥ 1.0 ng/ml. Kaplan–Meier plots for disease-specific survival were plotted for these two groups for patients with OPSCC irrespective of HPV status ($n = 86$) (**f**), as well as for HPV$^-$ ($n = 32$) (**g**) and HPV$^+$ ($n = 54$) (**h**) OPSCC. Spearman´s rank-correlation coefficients ($\rho$) and $p$-values are indicated for **b–e**. Dotted trend lines were added for visualization. Kaplan–Meier curves were compared by log-rank (Mantel-Cox) test (**f–h**). Source data for **b–h** are provided as a Source Data file.

variate analysis further revealed that serum S100B levels, age, sex, and line of treatment had no significant impact, while the presence of brain metastases ($p < 0.001$), serum LDH levels ($p = 0.008$) and GDF-15 levels ≥2 ng/ml ($p = 0.001$, Fig. 6i) were associated with poor survival. sLDH, however, lost significance as predictor for survival in the subgroup of patients with high GDF-15. Moreover, 12 of 25 patients with high sLDH

(and presumably high tumor load) and low GDF-15 were still alive at the end of the observation period, while no long-term survivors were among the 6 patients with low sLDH and high GDF-15. High GDF-15 serum levels thus indicate aggressive tumors that respond poorly to anti-PD-1 therapy (Fig. 6j), irrespective of tumor load. The impact of blocking GDF-15 remains to be explored in clinical studies.

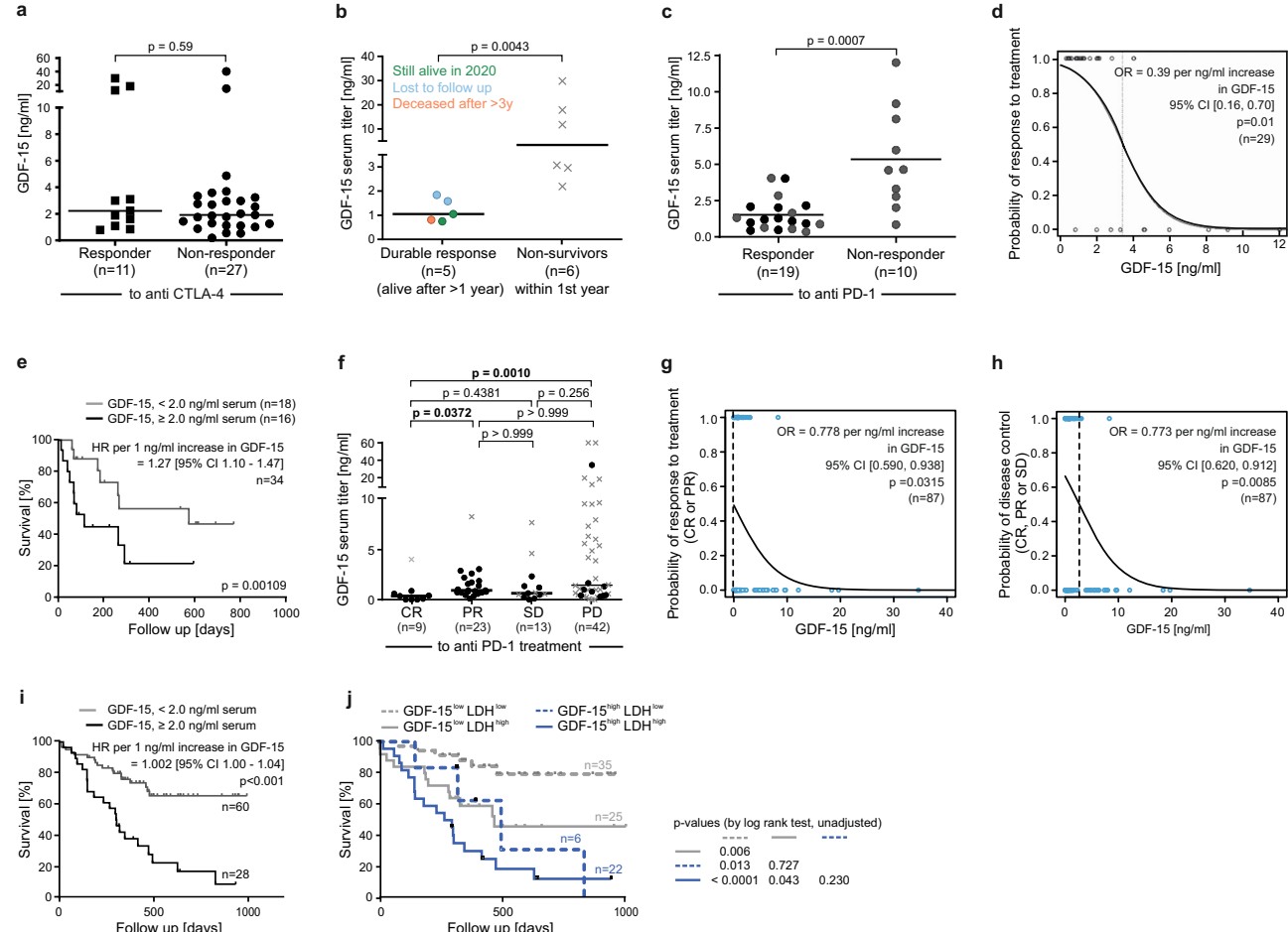

**Fig. 6 | In human melanoma patients GDF-15 serum levels predict response to and survival under therapy with anti-PD-1 antibodies. a, b** GDF-15 levels were analyzed in 37 melanoma patients at baseline of ipilimumab treatment, and correlated with clinical responses based on RECIST v1.1 criteria (**a**), including durability of initial responses (**b**). **c–j** GDF-15 levels were analyzed in pre-treatment sera from 34 melanoma patients prior to pembrolizumab treatment (Zurich cohort, **c–e**), and from 88 patients (Tübingen cohort, **f–j**) prior to treatment with pembrolizumab (n = 48) or nivolumab (n = 40). GDF-15 serum levels were correlated with responses according to RECIST v1.1 (**c, f**). Black circles indicate patients with ongoing responses at the time of analysis. Groups were compared by Mann–Whitney test in **a–c**. In **d, g, h**, logistic regression models were fitted for response (**d, g**) or disease control (**h**) under anti-PD-1 treatment. Dotted vertical lines indicate GDF-15 serum levels (continuous predictor) corresponding to a 50% probability of treatment response. Two extreme values ([GDF-15] ≫ 100 ng/ml) are displayed at the upper

end of the scale. In **e** and **i**, overall survival of 34 (**e**), respectively 88 (**i**) patients was analyzed by Cox proportional hazards model with overall survival (time to death) as outcome variable and GDF-15 as continuous predictor. Kaplan–Meier plots (cut-off: 2.0 ng/ml GDF-15) are shown for visualization (**e, i**). Further Kaplan–Meier curves including serum lactate dehydrogenase (sLDH) as additional predictor were calculated for the Tübingen cohort (**j**). Censoring is indicated by vertical lines. In **f**, p-values were calculated by Kruskal–Wallis test. In **f, g, h**, one patient whose clinical course contradicted the RECIST1.1-, and therefore target lesion-based classification as complete responder was omitted from statistical consideration. One further patient could not be staged and is therefore neither displayed nor assigned to any group. In **i, j**, overall survival between groups was compared using two-sided log-rank tests, including all patients. Horizontal bars in **a, b, c, f** depict median values. Source data for **b–h** are provided as a Source Data file.

## Discussion

GDF-15 has long been known to exert anti-inflammatory effects by inhibiting macrophages, dendritic cells, granulocytes and other cell types. Here we show that GDF-15 also prevents a stable, Talin-dependent linkage between LFA-1 and the actin cytoskeleton. Tumor-derived GDF-15 thus inhibits the LFA-1:ICAM-1 axis in human T cells, thereby interfering with T cell post rolling arrest, firm adhesion to endothelia, crawling and diapedesis across activated blood vessels (Figs. 1 and 2; S1 and S2). In vivo, GDF-15 expression translates into reduced immune infiltration in syngeneic and humanized tumor models (Figs. 3 and 4). GDF-15 blockade further enhances T cell trafficking to tumor-draining lymph nodes. GDF-15 thus interferes with responses to anti-PD-1-based immunotherapy in mice. Antibody-based neutralization of GDF-15 confirms the therapeutic potential of targeting GDF-15 in humanized and in syngeneic immunocompetent mouse models (Figs. 3 and 4). Blocking experiments confirm that the

observed effects depend on GDF-15 rather than on contaminating TGF-β frequently found in GDF-15 preparations[53] (Supplementary Fig. 8a). As melanoma cohorts from the TCGA database show a weakly negative correlation between *tgf-β1/2* and *gdf15* expression (Supplementary Fig. 8b, c), immune cell exclusion in GDF-15-overexpressing tumors cannot be ascribed to TGF-β[54]. Brain metastases from human melanoma are well-suited to assess correlations between GDF-15 and T cell infiltration, as these lesions are typically only resected once they have grown to a size that allows for a proper histopathological assessment. Other metastases are, in contrast, either removed as early as possible, or not at all. In these melanoma metastases infiltrating CD3+ and CD8+ T cells were inversely correlated with GDF-15 tissue levels (Fig. 5). Inverse correlations with tumor-infiltrating regulatory T cells were weaker. In HPV+ OPSCC, GDF-15 serum levels were negatively correlated with intratumoral CD8+ T cell numbers (Fig. 5) and survival. These and previous corroborating findings from colon cancer, glioma and

benign atrophic lesions of the human prostate can be explained by the newly discovered GDF-15-dependent inhibition of LFA-1 on immune effector cells[31,55,56]. GDF-15 thus represents a "T cell repellent" and a roadblock for anti-PD-1 treatment, which has been approved in Europe, the US, and elsewhere for different malignancies. Clinical trials with anti-GDF-15 and anti-PD-1 for immunotherapy refractory tumors are ongoing (NCT04725474) or announced[57]. The lack of association between GDF-15 serum levels and initial responses to CTLA-4 blockade implies that anti-CTLA-4 may overcome immune-inhibitory effects of GDF-15. Nevertheless, among melanoma patients responding to ipilimumab, only those with low GDF-15 levels showed durable responses and survival >1 year (Fig. 6a, b). The lack of durable responses in ipilimumab-treated melanoma patients with high GDF-15 levels may be due to impaired trafficking of T cells into the tumor tissue. Moreover, induction of long-term protective CD8[+] T cell memory requires initial T cell priming to be followed by LFA-1-dependent signals delivered via secondary synaptic circuits during a critical differentiation period[58]. Accordingly, GDF-15 may impair long-term tumor control in patients treated with anti-CTLA-4. Clinically even more striking is the strong correlation between high levels of GDF-15, failure of anti-PD-1 monotherapy and poor survival in two independent cohorts of melanoma patients (Fig. 6c–j). GDF-15 even achieves a better discrimination between responders and non-responders to anti-PD-1 than sLDH. Consequently, patients with high GDF-15 serum levels are less likely to benefit from PD-(L)1 monotherapy. Conveniently, patients likely to benefit from anti-GDF-15 treatment can be easily selected by measuring GDF-15 serum levels[59] with commercially available assays (e.g., Elecsys® assay; with granted Breakthrough Device Designation by FDA). Limitations of our study are the focus on melanoma, and on brain metastases for the analysis of immune infiltration. Moreover, GDF-15 also acts on immune effectors beyond T cells (see[60] and the inhibition of LPS-induced macrophage polarization shown in Supplementary Fig. 5c). Still, the newly discovered effects of GDF-15 on T cells provide a mechanistic basis for T cell exclusion from GDF-15 expressing tumors, and a strong rationale for targeting GDF-15. This is supported by in vitro experiments, mouse models and clinical data. GDF-15 thus acts as "vascular immune checkpoint" that interferes with PD-1-based (and possibly other) immunotherapies. As GDF-15 is overexpressed in >50% of all solid cancers (compare Supplementary Fig. 3), it is a promising actionable new target for cancer immunotherapy.

Potential unwanted clinical effects of prolonged GDF-15 suppression can only be speculated upon at this stage. A mild (≤20%) slowly progressive reduction in motoneurons was just observed in one of several genetic *gdf15* knock-out models[61], and may not be relevant when GDF-15 signaling is inhibited at later stages in life. Instead, the most prominent effect of GDF-15 blockade in animal models is prevention of weight loss[17]. In humans, low GDF-15 levels are even associated with a longer lifespan[62]. Notably, GDF-15 may primarily have local effects, depending on its diffusion gradient and biological $IC_{50}$, thereby excluding immune cells from the tumor microenvironment in the absence of generalized immunosuppression. With regard to intracellular signaling it is noteworthy that the GDF-15 receptor GFRAL, which was initially reported to be brainstem-restricted, was undetectable in mRNA from human T cells (Supplementary Fig. 9). Moreover, a recent study showed that liver-protective, anti-inflammatory effects of colchicine are mediated via induction of GDF-15, and therefore lost in GDF-15[-/-] mice, while GFRAL[-/-] mice still respond normally[27]. Likewise, anti-GFRAL had no effect in our flow adhesion assays (Fig. 1f). Immunomodulatory effects of GDF-15 thus appear to be independent of GFRAL. Clinically, however, GDF-15/GFRAL-dependent metabolic effects like induction of cachexia[17] may further contribute to the poor survival of melanoma patients with high GDF-15 serum levels.

Interestingly, recombinant murine GDF-15 failed to rescue *gdf15* deficiency in knock-out mice in a recent report[27]. This was ascribed to an insufficient bioactivity of recombinant murine GDF-15. Fortunately,

despite some variation in potency, the recombinant human GDF-15 (rhGDF-15) used over the course of this study consistently showed biological effects in line with in vivo observations. With regard to the original physiological function of GDF-15, its strong induction during pregnancy likely protects the semi-allogenic fetus from the maternal immune system[23]. Whether destabilization of the immunological synapse by GDF-15-dependent LFA-1 inhibition[63] augments the tolerogenic function of GDF-15[24,31] remains to be explored. Induction of anorexia and cachexia[17] via GFRAL on brainstem neurons[20–22] is, in contrast, not observed during pregnancy. It thus remains to be determined how and under which conditions GDF-15 induces cachexia, immune tolerance, or both. Identification of the GDF-15 receptor on immune cells and its cell-type specific deletion will further allow to disentangle the pleiotropic immunomodulatory effects of GDF-15. Likewise, exploration of molecular mechanisms and signaling pathways will greatly benefit from receptor identification. From a translational perspective, the potential to improve the response to cancer immunotherapy (pursued in clinical trial NCT04725474) and to ameliorate cancer- and chemotherapy-associated cachexia[17,64] (pursued in studies NCT04815551 & NCT04299048) provides two compelling reasons for the further clinical exploration of GDF-15-neutralizing cancer treatments.

## Methods

### Proteins and antibodies

HIS-tagged recombinant human Growth and Differentiation Factor-15 was produced in HEK293T cells (Invigate GmbH, Jena). The anti-human GDF-15 antibody used in this manuscript was generated in female C57BL/6J [GDF-15-/-] mice kindly provided by Dr. Jens Strelau (Heidelberg, Germany) and validated for binding affinity by surface plasmon resonance at NMI (Reutlingen, Germany). Specificity was tested by tissue cross-reactivity studies. Antibody for cell culture and in vivo experiments was produced by Exbio (Prague, Czech Republic). For experiments in humanized mice, a humanized version of this antibody was generated by grafting the complementarity determining regions on a hinge-stabilized human IgG4 backbone, followed by recombinant expression (Evitria, Schlieren, Switzerland). For experiments in fully immune-competent mice, a surrogate antibody (0297.mIgG1) was also produced by Evitria. To block GFRAL, clone 3P10, described in Suriben, R. et al. Antibody-mediated inhibition of GDF15-GFRAL activity reverses cancer cachexia in mice. Nat Med 26, 1264-1270 (2020), and in US20170306031A1 was used. Isotype antibodies were anti-Fluorescein [4-4-20 (enhanced)] Human IgG4 S228P (Absolute antibodies) or mouse IgG1 MOPC-21 (Biolegend). Unless otherwise indicated, antibodies were used according to the manufacturer´s instructions.

### Adhesion, rolling, and transmigration assays

For whole blood-based immune cell adhesion assays, $4×10^5$ huLEC cells/well were seeded in gelatin-coated 6 well-plates, cultured for 2 days, and stimulated for 5 h with TNF-α and IFN-γ (50 ng/ml each). Heparinized blood was obtained from healthy volunteers and mixed for 10 min with CXCL12α (200 ng/ml) ± GDF-15 (10 ng/ml) as indicated. Keeping all components at 37 °C, medium was exchanged for 2 ml of blood/well. Cells were gently spun down at 100 rpm for 1 min and the plate was placed on a gently rocking shaker set at 200 rpm (at 37 °C). The wells were swirled three times with 2 ml of 37 °C warm PBS with $Ca^{2+}$ and $Mg^{2+}$, using a medical aspirator to remove non-adherent cells. For harvesting, the plate was put on ice and cells were collected in ice-cold PBS without $Ca^{2+}$ and $Mg^{2+}$. Fc-receptors were blocked with TruStainFcX (BioLegend, San Diego, CA) and cells were stained with anti-human CD45-SuperBright 436 clone 2D1 (eBioscience, San Diego, CA), anti-human CD3-APC/Cy7 clone HIT3a, anti-human CD4-BrilliantViolet711 clone A161A1, anti-human CD19-BrilliantViolet711 clone HIB19, anti-human CD56-APC clone 5.1H11, anti-human CD66b-PECy7 clone G10P5 (all from BioLegend), anti-human CD8-PE clone UCHT-4

and anti-human CD14-FITC clone MEM-15 (both from Immunotools, Friesoythe, Germany). After staining, cells were fixed with 4% formaldehyde for 10 min, PBS was added, and cells were stored overnight at 4 °C (to promote erythrocyte lysis), before being analyzed and enumerated on an Attune Nxt flow cytometer (LifeTechnologies, Carlsbad, CA) suitable for whole blood analyses.

For *in-house* adhesion assays under flow conditions, μ-slides VI 0.4 (ibidi GmbH, Martinsried, Germany) were coated with fibronectin (100 μg/ml, 30 μl per loading port) and incubated for 1 h at 37 °C. Fibronectin was aspirated, followed by a wash with huLEC or HUVEC medium (InSCREENeX, Braunschweig, Germany). huLEC (InSCREE-NeX) or HUVECs (Lonza, Basel, Switzerland or freshly isolated) were detached, washed and resuspended at 1×10⁶ cells/ml. 30 μl of this cell suspension were applied to the loading ports of the μ-slide VI which was covered with a lid and endothelial cells were left to grow at 37 °C with 5%CO₂ until 90% confluent (24–48 h). Endothelial cells were activated with TNF-α (5 ng/ml) and IFN-γ (5 ng/ml) in channels 2–5. All media were aspirated from the channels and replaced by pre-warmed media with cytokines. Pan Effector T cells were isolated by negative selection (Miltenyi, Bergisch Gladbach, Germany) and stimulated overnight with ImmunoCult™ CD3/CD28/CD2 T cell Activator cocktail (Stemcell Technologies, Cologne, Germany) before being pretreated or not with rhGDF-15. Regulatory T cells were isolated with the human CD4⁺CD25⁺CD127^{dim/-} regulatory T cell isolation kit II and expanded with the human T_{reg} expansion kit to a purity ≥90%, based on intracellular FoxP3 staining (all reagents from Miltenyi). Where indicated, endothelial cells were pre-incubated for 20 min with GDF-15 (Invigate, Jena, Germany) at the indicated concentration. A heat chamber or a stage top incubator next to the microscope was pre-warmed, and a gas-mix was connected (5% CO₂, 16% O₂, 79% N₂). Untreated T cells, rhGDF-15-treated T cells (1×10⁶ cells/ml each) or medium were perfused through the channels at 0.5 dyn/cm² (0.38 ml/min). T cell flow was established for 5 min while 10 fields of view were predefined on the microscope. Each field of view was video-imaged for 5 s and adherent as well as rolling T cells were counted. Resulting data were compared using Mann–Whitney test for testing of non-normally distributed data.

Further flow adhesion assays were performed at MesenFlow Technologies (Geneva, Switzerland). HUVECs (Lonza, Catalog #:CC-2517, used up to a maximum of 5 passages) were cultured in chamber slides over 2-3-days prior to overnight activation with TNF-α (1000 U/ml) and IFN-β (500 U/ml). Healthy donor PBMCs obtained from buffy coats not older than 24 h on the day of experimentation and MACS® cell separation kits (Miltenyi Biotec) were used to purify pan CD4⁺ T cells (human CD4⁺ T Cell Isolation Kit #130-096-533), naive CD4⁺ T cells (human naive CD4⁺ T Cell Isolation Kit II #130-094-131), CD4⁺ memory T cells (human memory CD4⁺ T Cell Isolation Kit, #130-091-893), CD4⁺ effector memory T cells (human CD4⁺ Effector Memory T Cell Isolation Kit, #130-094-125), pan CD8⁺ T cells (human CD8⁺ T Cell Isolation Kit #130-096-495), naive CD8⁺ T cells (human naive CD8⁺ T Cell Isolation Kit #130-093-244) or CD8⁺ memory T cells (human CD8⁺ memory T Cell Isolation Kit #130-094-412). Purified subsets were stored at 4 °C in 1% BSA and adapted to 37 °C 1 h prior to experimentation. Physiological flow (0.5 Dyns/cm²) was generated through mounted HUVEC culture slides on a heated microscope chamber (37 °C) using a calibrated pump. Chemokines (1 μM CXCL12α or 0.5 μM CXCL9 + 0.5 μM CXCL10) and GDF-15 (100 ng/ml) or vehicle control (100 mM acetic acid) were then perfused over the activated HUVEC monolayer for 5 min followed by 15 min stasis (step 1). Wash-buffer was then pumped over the HUVECs for 10 min to remove any unbound CXCL chemokines or GDF-15. Leukocyte suspensions were pre-treated for 20 min with rhGDF-15 (100 ng/ml), or blocking anti-LFA-1 antibody TS1/18 (20 μg/ml) (# MA1810, ThermoFisher Scientific), or both, or vehicle control before centrifugation and resuspension in washing medium. Leukocytes were then perfused over HUVECs for 6 min (step

2) before wash-buffer at a pressure of 0.1 Pa was run over the cells for 50 min (step-3). Throughout steps 2-3, captured leukocytes were imaged every 30 s by phase-contrast microscopy. The resulting short movie sequences allowed analysis of individual leukocytes over large areas. The total number of adhesion events per unit field (0.19 mm²) was expressed per mm². T cells adherent to the surface of HUVEC had a phase-white/gray appearance, whereas those that had transmigrated (referred to as recruited) had a phase-black appearance. The number of captured cells at each time point equals the total number of adherent cells on either side of the HUVEC layer. Transmigration events (phase black) were presented as a percentage of total T cells (phase gray + black) captured from flow per unit field. The impact of GDF-15 on T cell adhesion was addressed using either pan T cells (Fig. 1f–h) or further T cell subsets (Fig. S2) with the same experimental settings. Depicted time points (30 min or 15 min) are indicated. Three to four donors were used per condition.

For flow adhesion assays with recombinant adhesion molecules, hydrophobic 0.2 mm Luer μ-Slides (ibidi) were coated with CXCL12α (Immunotools) and either recombinant ICAM-1 (#720-IC-200, R&D Systems, Wiesbaden, Germany, and generated in-house) or VCAM-1 (#862-VC-100, R&D Systems, Wiesbaden, Germany) or MAdCAM-1 (#6056-MC-050, R&D Systems) Fc-fusion protein in PBS. After 2 h at 37 °C, the coated slides were blocked with 1.5% PBS-BSA solution for 1.5 h at 37 °C. T cells were purified by negative isolation (Miltenyi), stained with a fluorescent cell tracker (CMFDA, ThermoFisher) and stimulated for 30 min with ImmunoCult™ CD3/CD28/CD2 T cell Activator cocktail (StemCell Technologies). 100 ng/ml rhGDF-15 or vehicle control were added for another 30 min before the stimulated T cells were subjected to flow conditions. Anti-LFA-1 blocking antibody (TS1/18, used at 20 μg/ml), anti-VLA-4 blocking antibody (20 μg/ml of # BE0071, BioXcell, Lebanon, NH, USA) or anti-α4β7 integrin antibody (5 μg/ml of #MAB10078, R&D Systems) were added as indicated. Under a heated CO₂ stage top incubator (ibidi) fixed to a microscope, uni-directional flow was applied to the coated channel slide. Bright field live microscopy images were recorded for 6 min under flow conditions (2.68 dyn/cm²) followed by a 10 min wash with cell culture medium to flush out non-adherent cells. Adherent stained cells were quantified by fluorescent microscopy. Image analysis was performed by counting the number of adherent cells using the open source application "CellProfiler" with an appropriate pipeline to finally quantify the number of adherent T cells.

## Ligand-complex-based binding assays and mAb24 anti-active LFA-1 conformation-specific stainings

For immune complex formation 50 μg/ml ICAM1-Fc (produced in house, alternatively available from R&D Research, Wiesbaden, Germany) were mixed with 40 μg/ml APC-conjugated F(ab)₂ anti-human Fc antibody (Jackson Immunoresearch,) in assay buffer (Ca²⁺ and Mg²⁺-free PBS + 0.5% BSA) and incubated for 30 min at room temperature. AF488-labeled mAb24 anti-active LFA-1 antibody (BioLegend) was used at a 1:300 dilution. As LFA-1 conformation is temperature-dependent all cells were kept at 37 °C. Whole blood was pre-incubated or not with 20 ng/ml rhGDF-15 (Invigate, Lot 682039) for 10 min prior to adding an equal volume of stimulation and staining cocktail, thereby effectively diluting the GDF-15 concentration to a final of 10 ng/ml. CXCL12α was used at 200 ng/ml, αCD3/CD28 was diluted by 1:20. Anti-human CD3-APC/Cy7 clone HIT3a (BioLegend), anti-human CD4-BrilliantViolet711 clone A161A1 (BioLegend), anti-human CD8-PECy7 clone RPA-T8 (eBioscience) defined the cells of interest. After staining, cells were fixed with 4% para-formaldehyde and stored overnight at 4 °C to promote erythrocyte lysis, before being analyzed on an Attune Nxt flow cytometer (ThermoFisher) employing a whole blood filter. To allow for statistical evaluation mean fluorescent intensity (MFI) values were normalized to control conditions by z-transformation. Samples were measured in duplicates.

### *d*STORM super-resolution microscopy for analysis of active versus inactive LFA-1

For immune complex formation 5 μg/ml ICAM1-Fc (produced in house) were mixed with 5 μg/ml AF647-conjugated anti-human IgG antibody (ThermoFisher) in assay buffer ($Ca^{2+}$ and $Mg^{2+}$-free PBS + 0.5% BSA) and incubated for 30 min at room temperature. For microscopy measurements, 8-well LabTek chamber slides (Nunc™ Lab-Tek™ II Chamber Slide™ System, ThermoFisher) were coated with poly-D-lysine (Merck) and $2\times10^5$ PBMCs or CD8$^+$ T-cells were seeded 1 h prior staining at 37 °C. Afterwards, cells were treated for 30 min with 10 ng/ml rhGDF-15. LFA-1 was activated by adding 1 mM $MgCl_2$ + 1 mM EGTA + 100 ng/ml CXCL12α for 30 min. For staining, the AF647-labeled hICAM-1-Fc immune complex or, respectively, the AF647-labeled activation-specific anti-LFA-1 antibody mAb24 (used at 5 μg/ml) was applied together with 2 μg/ml anti-CD3-AF488 and 2 μg/ml CD8a-AF532 (all from ThermoFisher Scientific) for 20 min at 37 °C[65]. After washing, cells were fixed for 15 min with 2% formaldehyde and 0.25% glutaraldehyde, washed and transferred into a PBS-based buffer (pH 7.4) with 100 mM β-mercaptoethylamine (Sigma-Aldrich, Taufkirchen, Germany) to allow reversible photoswitching of Alexa Fluor 647. *d*STORM measurements were performed as described[35], using an IX-71 inverted microscope equipped with an APON 60XOTIRF oil-immersion objective and an IX2-NPS nosepiece stage (all from Olympus, Hamburg, Germany). AF647, AF532 and AF488 were excited with appropriate laser systems (Genesis MX 639 and MX514-500 from Coherent, Göttingen, Germany; iBeam smart 488 nm, Toptica, Gräfelfing, Germany). The excitation light was spectrally cleaned by appropriate bandpass filters and focused onto the backfocal plane of the objective. To switch between different illumination modes (Epi and TIRF illumination) the lens system and mirror were arranged on a linear translation stage. A polychromatic mirror (HC 410/504/582/669, Semrock, Rochester, NY, USA or ZT405/514/635rpc, Chroma) was used to separate excitation (laser) and emitted (fluorescence) light. The fluorescence emission was collected by the same objective and transmitted by the dichroic beam splitter and several detection filters (HC 440/521/607/700, Semrock; HC 679/41, Semrock, for Alexa Fluor 647; HQ 582/75, Chroma (Bellows Falls, VT, USA), for Alexa 532; ET 525/50, Chroma, for Alexa 488), before being projected on two electron-multiplying CCD cameras (both iXon Ultra 897, Andor, Belfast, UK; beam splitter 635 LP, Semrock). The final pixel size of 128 nm was generated by placing additional lenses in the detection path. Excitation intensity was about 3.3 kW/cm$^2$. Typically, 15,000 frames were recorded at a frame rate of ~50 Hz (20 ms exposure time). From the recorded image stack a table with all localizations as well as a reconstructed *d*STORM image were generated using the localization software rapidSTORM 3.3[66]. Only CD3$^+$ CD8$^+$ cells were further analyzed for active LFA-1 expression. Quantification of mAb24 binding was performed with a custom-written Python script. The analysis routine included the following steps: Fluorescent spots containing less than 800 photons per frame were discarded. Repeated localizations coming from one antibody were grouped using a DBscan algorithm with 3 localizations in the locality of 20 nm. It was confirmed that the overall density of detected antibodies was small enough to yield well separated nearest neighbors. Antibody densities were calculated from the number of grouped localizations divided by the area of the bottom plasma membrane of each cell as determined with a region of interest (ROI)-selector. 10–20 cells per donor and condition were analyzed to obtain average mAb24 or ICAM-1-Fc density distributions.

### Bead-based adhesion signaling assay

In all, 3 mg Protein G beads (Dynabeads™, thermofisher) were loaded for 20 min with 24 μg of recombinant human ICAM-Fc and 24 μg recombinant human E-selectin-Fc and washed. $4\times10^6$ purified T cells were added to 1.5 mg ICAM-coated beads in a total volume of 200 μl PBS with $Ca^{2+}$ and $Mg^{2+}$, in the absence or presence of GDF-15 (100 ng/ml for 20 min). After 7 min, supernatant was discarded and cells were lysed by adding 100 μl hot SDS Lysis buffer (1% SDS, 10 mM Tris pH 8.0, 1 mM EDTA, 10 mM PMSF, 25 mM ß-Glycerophosphat, 1 mM NaF, 1 mM $NaV_2O_5$, all from Sigma, Proteinase Inhibitor (Carl Roth, Karlsruhe, Germany) and PhosStop (Roche)). After boiling for 3 min at 100 °C, cells were 4–5 times sonicated for 30 s. When all cells were lysed, beads were magnetically removed and protein concentrations were determined (RotiQuant, Carl Roth). 20 μg protein/lane were loaded on an SDS-polyacrylamide (7,5-15%) gel, separated by SDS gel electrophoresis (BioRad) and transferred onto a PVDF membrane (Roche), which was cut into two at around 50 kDa and blocked for 15 min with Everyblot Blocking buffer (BioRad). The upper part of the membrane was probed with anti phospho-Talin Ser$^{425}$ (clone D2P2M, Cell Signaling, expected molecular weight (MW): 270 kDa), the lower part with anti-CD3ε (Clone UCHT1, Biolegend, expected MW: 21 kDa). Signals were visualized with hrp-labeled secondary antibodies (Cell Signaling) and detected in an iBright CL750 chemoluminescence imager (ThermoFisher), using Westar Supernova (Cyanagen, Bologna, Italy).

### Immunohistochemistry

Formalin-fixed paraffin-embedded (FFPE) tissues from melanoma brain metastases from 70 patients were collected and processed as tissue microarrays (TMAs). All specimens had been obtained from the cancer registry tumor bank "Blut-und Gewebebank zur Erforschung des malignen Melanoms" (Department of Dermato-Oncology and Department of Neuropathology, University Hospital Tübingen, Germany) and approved by the local ethics committee. Immunohistochemistry was performed using 3 μm thick slides and standard protocols on the automated IHC staining system Discovery XT (Roche/Ventana, Tucson, Arizona, USA) using anti-GDF-15 (HPA011191, dilution 1:50, Sigma/Atlas, protocol #730), anti-CD3 (clone A0452, dilution 1:500), anti-CD8 (clone C8/144B, dilution 1:100, both from DAKO, Glostrup, Denmark), and anti-FOXP3 (clone 236 A/E7; dilution 1:100; eBioscience, San Diego, U.S.A.) antibodies. CD3$^+$, CD8$^+$ and FOXP3$^+$ immune infiltrates in brain metastases had been quantified previously[45]. Slides were counterstained with hematoxylin and mounted. All stained samples were scored according to the frequency of positive cells related to all cells (as percentage) on the stained TMA core. For hGDF-15 expression, a score as previously described in detail[67] was used: frequency 0–1% score 0; 1–10% score 1; 10–25% score 2; 25–50% score 3; >50% score 4; additionally the frequency score was multiplied with the intensity of staining (1 = weak, 2 = moderate, 3 = strong staining), finally resulting in the ordinal scaled hGDF-15 score (0, 1, 2, 3, 4, 6, 8, 9, 12).

For analysis of murine tissues, tissue sections were stained with anti-CD8a (clone 4SM15, 1:100 dilution; eBioscience), or anti-Granzyme B (clone E5V2L) or anti-FoxP3 (clone D6O8R, both from Cell Signaling Technologies) on the automated IHC staining system Discovery XT and evaluated by an observer who was blinded with regard to the sample or treatment group. Immune infiltrates of CD8-positive cells were scored as percentage of all cells. Local perinecrotic infiltration was scored on an ordinal scale ranging from zero to three (zero: no infiltration; one: low; two: moderate; three: strong infiltration).

### qRT-PCR for *GFRAL* mRNA expression (supplement)

RNA was isolated from 5-10×10$^6$ cells using TRI reagent (Sigma-Aldrich). DNA was removed with the Ambion® DNA-free™ kit (ThermoFisher). RNA was reverse transcribed using iScript cDNA synthesis kit (BioRad, Munich, Germany). qRT-PCR was performed on a StepOnePlus (Applied Biosystems) qRT-PCR cycler, using the 2x TaqMan Fast advanced Master mix (ThermoFisher) and probes (all from ThermoFisher) for *human GDNF family receptor alpha like/gfral* (assay ID

Hs01087628_m1) and for *TATA-box binding protein/tbp* (assay ID: Hs00427620_m1).

## Animal experiments

**Cells.** *gdf15* was cloned into the pT2B-puro Sleeping Beauty transposon vector described in patent application EP1616 kindly provided by Zsuzsanna Izsvák (Berlin, Germany). MC38 colon tumor cells were provided by the respective CRO (Charles River Laboratories or Fluofarma). MC38[blank] and MC38[tghGDF-15] cells were generated by transfection with either a GDF-15 expression plasmid or an empty vector control and selected with puromycin (InvivoGen, San Diego, CA, USA). Firefly luciferase-expressing Panc02 (Panc02 FUGLW) and EMT6 cells were cultured in DMEM supplemented with 10% heat inactivated FCS, Penicillin (100 U/ml), Streptomycin (100 μg/l), L-glutamine (2 mM) (all from Gibco) at 37 °C and 5% $CO_2$. Cells were harvested with Trypsin (Gibco).

**Tumor inoculation.** Mice were co-housed in specific pathogen-free (SPF) facilities where temperature was controlled between 22 and 25 °C, humidity was maintained between 40 and 70%, and artificial lighting was maintained for 12 h per day. $2 \times 10^5$ of either MC38[blank] or MC38[tghGDF-15] colon cancer cells in PBS were subcutaneously inoculated in the right flank of 7-week-old female C57BL/6NCrl (#027) or NCI[ath/nu] (#490) mice (Charles River). For orthotopic inoculation with Panc02 cells, all mice received pre-, peri- and post-operative analgesia. For inoculation into the head of the pancreas (approved by the governmental review board of the State of Bayern, Regierung von Unterfranken, under the authorization number 55.2.2-2532-410), the lateral side of anesthetized male albino C57Bl/6J mice (B6(C)Rj-Tyr c−c, Janvier Laboratories, Saint Berthevin, France, #SC-C57AL-M) was opened with a surgical incision. Spleen and pancreas were externalized. Using a 100 μl syringe (710 RN 100 ml, Gauge 28, 10 mm, Point Style 4 needle, Hamilton), $1 \times 10^4$ tumor cells were then injected in 30 μl PBS. Afterwards the wound was closed with 6–0 coated Vicryl Polyglactin BV-1 (Ethicon). Inoculation of Panc02 into the tail of the pancreas was performed by Crown Bio. $1 \times 10^6$ Panc02-Luc cells in 20 μl growth-factor reduced Matrigel were orthotopically injected in anesthetized 8–9 weeks old female C57BL/6 albino (C57BL/6BrdCrHsd-Tyr[c]) (Envigo, Venray, Netherlands, #103) as follows: A traverse incision was made over the spleen. Then an incision was made in the underlying abdominal muscle. The spleen was gently removed from the abdominal cavity and secured along a second saline-wetted cotton bud to expose the underlying pancreas. The tail of the pancreas was located adjacent to the spleen. Using a 29G insulin syringe, 20 μL of the Matrigel-cell suspension was injected into the pancreas. Following injection, the syringe was held in the pancreas for ~30–60 s until the Matrigel had solidified. The site of injection was inspected to ensure no leakage had occurred. The spleen and pancreas were carefully returned to the abdominal cavity. The abdomen and the skin were then be closed using a suture line followed by application of clips. To induce breast tumors (as approved by the governmental review board of the State of Bayern, Regierung von Unterfranken, under the authorization number 55.2.2-2532-1242-12), $2 \times 10^5$ EMT6 murine breast cancer cells were orthotopically injected into the fourth mammary fat pad of female 7 to 10-week-old BALB/c inbred (BALB/cOlaHsd, Envigo, #16204F) mice. Animals were excluded if tumors failed to form or if health concerns were reported. Humane end points that prompted termination of an experiment were defined by absolute termination criteria that included weight loss >20% in all models, palpable tumor volume >2000 mm³ in NCI[nu/nu]-mice with sub-cutaneous MC38 tumors, >1500 mm³ in C57Bl/6j-mice with sub-cutaneous MC38 tumors, >520 mm³ in BALB/c-mice with orthotopic EMT6 tumors, and a bioluminescent tumor signal >$10^9$ photons/s in SC-C57AL-M mice inoculated with Panc02-luc cells. Further, behavior, posture, respiration, activity, mobility, coordination, condition of body, fur, eye lids, retina, and signs for pain were used to derive a composite score that indicated when an animal had to be euthanized. For euthanasia, cervical dislocation or $CO_2$ asphyxiation were deemed acceptable.

**Bioluminescence imaging.** In vivo tumor growth was monitored at least weekly from day 5 after tumor inoculation by bioluminescence imaging for all groups. D-luciferin (300 mg/kg) was injected i.p. and anesthesia was induced with 1-2% isoflurane. After 10 min the animals were imaged with an IVIS Spectrum (Perkin-Elmer) or Spectrum BL and imaged for luminescence in left lateral and ventral view. Imaging data were analyzed with Living Image software (Caliper LS, US).

**Anti-GDF-15 and anti-PD-1/anti-GDF-15 combination treatment of tumor-bearing mice.** In the sub-cutaneous MC38 model, intraperitoneal treatment with anti-GDF-15 antibody (20 mg/kg), anti-PD-1 antibody (clone Rmp1-14, used at 5 mg/kg), isotype control antibody (either IgG1κ MOPC-21, BioLegend, or mIG2a B12 anti-HIV, Evitria, Schlieren, Switzerland, as appropriate) or vehicle was applied twice weekly from day 3 after tumor inoculation. Tumors were measured with a caliper twice weekly for the duration of the study. Individual body weight loss ≥30% for one measurement, or ≥25% for three measurements was defined as exclusion criterion. In the orthotopic Panc02 model, animals were randomized on day 5 based on their bioluminescence image signal intensity (and hence tumor load) into the different treatment groups (n = 12 per group at randomization, n = 10–12 per group at final evaluation). Starting from day 5, animals were treated intraperitoneally (i.p.) with either 10 mg/kg isotype control antibody (clone MOPC-21; Biolegend), anti-murine GDF-15 (0297.mIgG1), anti-murine-PD-1 (RMP1-14.mIgG1) or a combination of anti-murine GDF-15 and anti-murine PD-1 antibody. Antibody treatment was given biweekly until day 27 or, respectively, day 34 for a total of 8–9 treatments. On day 29 or, respectively, 35 all animals were finally monitored with bioluminescence imaging and euthanized. The Tumor Burden Change was calculated as (Photon flux on d35-d5)/(Photon flux on d5). Mice with EMT6 tumors were injected i.p. with 10 mg/kg anti mGDF15 antibody (0297.mIgG1) or isotype control IgG1 (MOPC-21) on day 6, 9 and 12 after tumor cell implantation. Tumor size was monitored by caliper measurement. Tumor volume was calculated based on the ellipsoid formula: $\pi/6 \times$ (length × width²). Tumor weight was determined after tumors had been explanted.

**Analysis of tumor-infiltrating T cells by flow cytometry.** Mouse tumor samples were dissociated according to the gentleMACS™ protocol "Tumor Dissociation Kit" provided by Miltenyi. Tumor-infiltrating lymphocytes were stained with antibodies against CD45 (clone 30-F11), CD3 (clone 17A2, both from BioLegend) and CD8 (clone 53-6.7, BD Biosciences). Pancreata of Panc02-bearing mice were harvested on day 12 and placed in Hank´s Buffered Saline Solution (HBSS) on ice. Tissue was cut into small pieces in a 50 ml falcon lid and placed in a 50 ml falcon tube with 9.5 ml RPMI 1640 media, 100 μg/ml DNase I and 400 μg/ml Collagenase P. Tissue digestion required incubation at 37 °C and 400 rpm for 30 min and trituration with a 10 ml pipette every 10 min. Afterwards, cells were disseminated through a 70 μm cell strainer, resuspended in PBS, centrifuged, washed, and stained with Zombie Aqua Fixable Viability Dye (Biolegend) and two different antibody panels: Panel 1 consisted of anti-CD45-BV711, anti-CD3-PE-Cy7, anti-CD8-BV421 (clones 30-F11, 145-2C11, 53-6.7; Biolegend) and anti-CD4 PerCP Cyanine5.5 (clone RM4-5; Invitrogen). Panel 2 contained anti-CD45-BV711, anti-CD11b-APC, anti-IA-IE-FITC, anti-F4/80-BV421, anti-Ly-6C-PE, anti-Ly-6G-BV605 (clones 30-F11, M1/70, M5/114.15.2, BM8, HK1.4, 1A8; Biolegend), anti-CD11c-PE-Cy7 (clone N418; Invitrogen).

In the EMT6 mouse model, adoptive T cell transfer was performed on day 13 post tumor inoculation, one week after the onset of treatment with either anti-GDF-15 or isotype antibody. CD3[+] T cells were thus isolated from pooled spleens of BALB/c organ donor mice (MoJo mouse T cell isolation kit; Biolegend #480024) and labeled with 5 μM

carboxyfluorescein succinimidyl ester (CFSE) (ThermoFisher, #65-08-50-85). $5 \times 10^6$ labeled T cells were then injected into the tail vein of EMT6 tumor-bearing mice. 24 h later animals were euthanized by cervical dislocation, weighed and tumors and draining lymph nodes were harvested. Harvested tumors were measured using a caliper and weighed to determine the volume and size. For FACS analysis, axillary and brachial lymph nodes and primary tumor were collected. Extracted lymph nodes were mechanically disrupted and digested at 37 °C for 60 min in 500 μl of a mixture of 1 mg/ml collagenase A and D (Roche, #10103578001, #11088858001) and 0.4 mg/ml DNase I (Roche, #10104159001) in PBS, under constant rotation (1000 rpm) in a thermo-mixer (Eppendorf). After addition of 10 mM EDTA (Sigma, #60-00-4), cells were passed through a 70-μm mesh. Cells were then stained with fixable viability dye live/dead eFluor 780 (eBioscience, #65-0865-14) and combinations of the following fluorescently conjugated antibodies: anti-CD3 AF 700 (clone 500A2, BioLegend), anti-CD4 APC (clone GK1.5, BioLegend), and CD8 BV421 (clone 53-6.7, BioLegend). FACS measurements were performed on an Attune NxT (Thermo Scientific) analyzer and evaluated using FlowJo software (Treestar, version 10.8.1).

**Analysis of human tumor-infiltrating immune cells in anti-GDF-15-treated humanized mice.** For humanization, female NOD/SCID/γc$^{-/-}$FcRγ$^{-/-}$ mice (Jackson #005557) bred in house were sublethally irradiated with 1.4 Gy within the first 24 h after birth. 6–18 h after irradiation, mice were intravenously injected with $2-5 \times 10^4$ hematopoietic stem cells isolated from human umbilical cord blood using the hCD34 isolation kit (Miltenyi Biotec). At the age of 10–12 weeks, the percentages of human immune cells in the peripheral blood were monitored via FACS analysis and animals with sufficient chimerism were selected for consecutive experiments. $5 \times 10^5$ HV-18-MK melanoma cells were injected subcutaneously into the neck of humanized mice. Intraperitoneal treatment with anti-GDF-15 or hIgG4 isotype antibody (20 mg/kg) was applied twice weekly from day 3 after tumor inoculation. Mice from different treatment groups were co-housed in individually ventilated cages under specific pathogen-free conditions. Animals were monitored at least on a daily basis and were immediately euthanized if they showed abnormal behavior (reduced activity, reduced food/water intake, separation for other animals) or signs of wasting (cachexia, lower body temparature, loss of body weight). For analysis of human tumor-infiltrating lymphocytes, mice were euthanized by CO$_2$ asphyxiation, tumors were excised, cut into pieces, and enzymatically digested for 3 h with collagenase D. Cells were stained with antibodies against human CD45 (clone 2D1), CD3 (clone SK7), CD8 (HIT8a) and CD19 (clone HIB19, all from BioLegend) and analyzed by flow cytometry.

## Serum analyses

Human and mouse GDF-15 was measured in serum or cell culture supernatant with sensitive and specific ELISA kits (DY957 and DY6385 DuoSet, R&D Systems, Biotechne) according to manufacturer's instructions. A customized ELISA was used to detect mouse anti-human GDF-15 antibodies. Briefly, diluted serum samples were applied to microtiter plates coated with human GDF-15 (0.5 μg/ml). Captured antibodies were detected with an anti-mouse IgG HRP (#7076, Cell signaling technology) secondary antibody. Quantitation was based on standard curves generated in the same matrix (e.g., serum).

## Patients

Eighty-six patients (53 male, 33, female) with histologically confirmed oropharyngeal squamous cell carcinoma (54 of the HPV$^{pos}$, 32 of the HPV$^{neg}$ subtype) were included at Leiden University Medical Center (LUMC) after they had signed the informed consent. The presented study was part of a larger observational study entitled: "Identification of immune response against HPV and p53 antigen in patients with a

squamous cell tumor arising from the head and neck region" (P07-112). The study was approved by the *Medical Ethics Committee Leiden The Hague Delft* and was in agreement with the Dutch law. Patient enrolment was from November 2007 until November 2015.

Surplus sera obtained during routine blood draws at baseline of ipilimumab treatment were available from 22 male and 15 female patients with stage IV melanoma at Würzburg University Hospital (age range: 36–79, median age: 68 years). Patients had received up to 4 applications of ipilimumab as monotherapy between 2011 and 2014 and were followed-up subsequently. The retrospective analysis of GDF-15 levels was approved by the *Ethik-Kommission der Universität Würzburg* (file number 20210310 01).

Surplus sera from 34 melanoma patients (25 male, 9 female, 26–80 year old, median age: 58 years) at baseline of pembrolizumab treatment were collected at the University Hospital Zurich (USZ) Biobank during routine blood draws from consenting metastatic melanoma patients according to institutional review board (*Ethik-Kommission der Universität Zürich*) approval (BASEC-Nr.PB_2017-00494) and following the Declaration of Helsinki on Human Rights. The University Research Priority Program in Translational Cancer Research (URPP) biobank processed the surplus material according to standard operating procedures established for routine biobanking at the USZ. Blood chemistry measurements were conducted by the routine hematology lab at the USZ according to standard procedures. Hemolysis prevented the determination of sLDH levels in 4 samples.

Surplus sera from 88 melanoma patients (54 male, 34 female, age range: 27–83 years, median age: 62 years) at the baseline of anti-PD-1 treatment (pembrolizumab in 48, nivolumab in 40 patients) were collected in the Department of Dermatology at Tübingen University Hospital. All patients with histologically confirmed melanoma were identified in the Central Malignant Melanoma Registry (CMMR) database[68]. All patients had given written informed consent to have clinical data recorded by the CMMR registry. The *Ethik-Kommission an der Medizinischen Fakultät der Eberhard-Karls-Universität und am Universitätsklinikum Tübingen* has approved the study (ethical vote 125/2015BO2).

GDF-15 serum concentrations were quantified in duplicates using a commercial ELISA kit according to the manufacturer´s instructions (R&D Systems). Investigators who analyzed GDF-15 serum levels were blinded with regard to patient data. Serum LDH was categorized as elevated vs. normal according to cut-off values used in clinical routine (upper limit of normal 250 U/l, respectively).

Total RNA was extracted from 40 short term melanoma cell cultures by QIAGEN RNA Mini kits. RNA capture was performed with TruSeq RNA Library Prep Kit v2 (Illumina) and sequenced on a HiSeq4000. RNA counts were quantified from single-end reads using STAR aligner (https://bioinformaticshome.com/tools/rna-seq/descriptions/STAR.html). Variant calling was performed with Haplotype Caller. GDF-15 expression and mutational load correlation was calculated with Pearson's correlation. Gene expression profiles from the Zurich cohort have been deposited under https://www.ncbi.nlm.nih.gov/geo/query/acc.cgi?acc=GSE198776. The analyzed TCGA data had been retrieved from https://portal.gdc.cancer.gov/.

## Pan-cancer analysis of GDF-15 expression

Gene expression profiles (RNA sequencing V2 data) and clinical information for 33 different cancer type cohorts with additional 3 combined projects of The Cancer Genome Atlas (TCGA) were retrieved via firebrowse.org (courtesy Broad Institute of MIT & Harvard). Samples were filtered for primary solid tumors (sample code 01) and corresponding normal tissue (sample code 11). Using raw count data, differentially expressed genes were identified with the statistical software environment R (version 3.3.2; www.r-project.org) and DESeq2 package. Differentially expressed GDF-15 mRNA for different tumor types were visualized as volcano plot.

## Statistical analysis

GraphPad Prism version 9 was used to generate most of the statistical analyses and graphics. Correlations between tumor volume and GDF-15 serum levels were assessed via Pearson's correlation coefficient whereas correlations between GDF-15 score or serum levels and percentage CD3[+], CD8[+] or FoxP3[+] or infiltrating CD8[+] T cells were assessed via Spearman´s rank-correlation. Comparisons of continuous outcomes between two independent samples were compared by unequal variance t-tests or by Mann−Whitney tests and, in case of paired samples, by paired t-tests or Wilcoxon rank sum tests, depending on whether the normality assumption was made or not. Apart from the one-sided comparison against a normalized control in 2l, all tests were two-sided. Comparisons of continuous outcomes among more than two independent samples were done by one-way ANOVA or Kruskal−Wallis tests. One-way ANOVA was used in the presence of blocks, usually due to different donors. Where appropriate, Tukey´s adjustment for multiple comparisons was applied. Adjustments for multiple comparisons made by t-tests were either done by Bonferroni or, to avoid p-values exceeding 1, by Bonferroni–Holm[69,70]. For non-parametric tests Dunn´s correction was used. Time-to-event outcomes such as overall survival were visualized by Kaplan–Meier curves, which were compared by log-rank (Mantel-Cox) tests. For the analysis of overall survival, patients who were alive at the last follow-up were censored while patients who had died were considered an "event". For the Tübingen melanoma cohort, where responses were categorized as complete response, partial response, stable disease or progressive disease, a Kruskal−Wallis test was performed. Due to a discordance between RECIST v1.1-based staging and clinical course, one patient was neglected in the statistical analyses for Fig. 6f, g, h. Using R (Version 4.04), overall survival was analyzed in both the Zurich and the Tübingen cohort of melanoma patients by Cox proportional hazards models with serum GDF-15 concentration as continuous explanatory variable, to estimate hazard ratios (HR) associated with a change in GDF-15 concentration by 1 ng/ml (compare Supplementary Table 2). Similarly, Cox proportional hazard models with tumor size as continuous explanatory variable were performed for Fig. 3t. Dichotomization of GDF-15 was used to visualize the data by Kaplan−Meier curves in Fig. 6e, i, j. Response to treatment (responder vs. non-responder) and, in the Tübingen cohort, also disease control was analyzed by logistic regression with GDF-15 as continuous predictor, to estimate odds ratios (OR) associated with a change in GDF-15 concentration by 1 ng/ml. Logistic curves are shown in Fig. 6c, g, h. In multi-variate analyses, Cox proportional hazard models were used to study the relationship between predictor features and overall survival time. The results are described by means of hazard ratios and p-values (Wald test). These analyses were carried out using SPSS 24 (IBM). The only figures based on technical replicates are Supplementary Figs. S1a, b and S3c. All other figures are based on biological replicates. Further detail on the type of analysis used for the respective experiment or substudy is given in the corresponding Figure legends.

### Reporting summary

Further information on research design is available in the Nature Portfolio Reporting Summary linked to this article.

## Data availability

Source data for Figs. 1a–j, 2a–g, j, l, 3a–h, j–o, q–t, 4a–d, f–m, 5b–h, 6a–j, S1a, S2a–i, S3c, S4a–f, S5a–c, S6a, S7a–f and S8a are provided as source data file. An unprocessed image of the western blot shown in Fig. 2k and flow cytometric gating strategies applied for Figs. 1a, b, 2f, g, 3m–o, 4c, d, f, g, h, l, m and S5a, b are provided as Supplementary information. Gene expression profiles from the Zurich cohort have been deposited under https://www.ncbi.nlm.nih.gov/geo/query/acc.cgi?acc=GSE198776. Additional questions will be answered by the corresponding author on reasonable request. Source data are provided with this paper.

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

## Acknowledgements

This work was supported by the following grants: #B-37 from the Interdisciplinary Centre for Clinical Research (IZKF), Medical Faculty, University of Würzburg, to J.W., #109319 from Deutsche Krebshilfe to J.W., #AZ-1365-18 (Bayerischer Forschungsverbund FORTiTher "Tumordiagnostik für individualisierte Therapie") from the Bayerische Forschungsstiftung to J.W. (TP1-WP2), #BMBF 01KT2310 TRANSCAN-3 (ERA-NET: Sustained collaboration of national and regional programmes in cancer research—JTC 2021) iParaCyts to J.W., #FKZ031A148 from the German Federal Ministry for Education and Research (BMBF) to M.H. and other team members of J.W., grant #2014-6696 from the Dutch Cancer Society to S.v.B. and M.W., European Regional Development Fund (EFRE project) "Center for Personalized Molecular Immunotherapy" to M. Sauer, #FNR PEARL P16/BM/11192868 from the Luxembourg National Research Fond (FNR) to M. Mittelbronn, #TRR305-B02 (Deutsche Forschungsgemeinschaft) to F.N., #TRR201-Z02 (Deutsche Forschungsgemeinschaft) to A.B. We thank the University Research Priority Program (URPP) in translational cancer research at the University of Zurich for funding the biobanking activities at the University of Zurich Hospital. We acknowledge the Department for Transfusion Medicine at the University Hospital Würzburg for providing blood cones, Markus Junker for generating the anti-GDF-15 antibody used in this study, Jens Strelau (Heidelberg, Germany) for having generously provided GDF-15-/- mice used for antibody generation and a GDF-15 expression plasmid, Zsuzsanna Izsvák (Berlin, Germany) for sharing the pT2B-puro Sleeping Beauty transposon vector, Johannes Dietl for institutional support and Sandy Westermann from SCIGRAPHIX for help with preparing the figures. This publication was supported by the Open Access Publication Fund of the University of Würzburg.

## Author contributions

Conceptualization: J.W. Methodology: M.H., T.S., P.N.H., P.E., N.V., F.W., A.I.E., F.N., C.S.W., M. Sauer, A.B., A.R., M. Mehling, J.W. Bioinformatics: H.H., F.H.G., Validation: B.H., C.S.W. Formal analysis: S.v.F., B.H., M.H., J.W. Investigation: B.H., T.S., P.N.H., G.M., P.E., N.V., F.W., B.F., J.D., F.M., A.K., V.T., L.S., G.A., M. Selle, T.M.B., K.T., J.S., M.C.G., C.R., A.I.E., K.E., K.F., P.S.Z., R.G., P.R.R., H.K.J., D.P., P.F.C., M. Mehling. Resources: M.J.P.W., K.W.H., R.B., R.W.R., A.T., A.W., J.E.D., M.P.L., M.G., M. Mittelbronn, S.H.v.d.B., B.W., R.D., Data curation: M.H., T.S., B.H., M.J.P.W., A.T., P.R.R., M.R., E.L., M.P.L., M.G., B.W., J.W. Writing—original draft: J.W. Writing—review & editing: M.H., B.H., T.S., P.N.H., P.E., N.V., F.W., B.H., S.G., M.J.P.W., V.T., L.S., H.H., R.W.R., M.R., E.L., P.F.C., M.P.L., M.G., M. Sauer, F.N., C.S.W., S.v.F., M. Mittelbronn, S.H.v.d.B., A.B., A.R., M. Mehling, B.W., R.D., J.W. Visualization: P.N.H., P.E., N.V., B.H. Supervision: M.H., T.S., P.N.H., N.V., F.W., S.G., M.J.P.W., M.G., M. Sauer, F.N., C.S.W., M. Mittelbronn, S.H.v.d.B., A.B., A.R., B.W., R.D., J.W. Project administration: M.H., M.R., E.L., C.S.W., J.W. Funding acquisition: M.H., M.J.P.W., M.R., M Mittelbronn, S.H.v.d.B., A.B., J.W.

## Funding

## Competing interests

M.H., T.S., M. Mehling, M. Selle, R.D., B.W., and J.W. are inventors on patents related to GDF-15 as a biomarker and therapeutic target. M.H., T.S., and J.W. are co-founders of the biotech company CatalYm GmbH involved in translating GDF-15-based cancer therapies and diagnostics into the clinics. M.H., T.S., M.R., E.L., and C.S.W. are current or former employees and stock owners of CatalYm. N.V., S.G., M.C.G., K.E., and P.R.R. are current or former employees of CatalYm. R.R. is a partner of Forbion Capital partners who are invested in CatalYm. R.D. has intermittent, project-focused consulting and/or advisory relationships with Novartis, Merck Sharp & Dohme (MSD), Bristol-Myers Squibb (BMS), Roche, Amgen, Takeda, Pierre Fabre, Sun Pharma, Sanofi, Catalym, Second Genome, Regeneron, Alligator, T3 Pharma, MaxiVAX SA and touchIME outside the submitted work. A.B. is scientific cofounder and advisor of Aamuthera Biotech GmbH and DualYX NV that are not related to this work. K.W.H., H.H., F.N., R.D., and J.W. have received research funding via Catalym GmbH. Other authors declare no competing interests in this work.

## Additional information

Markus Haake [1,2], Beatrice Haack[1], Tina Schäfer[1], Patrick N. Harter[3,4,5,6], Greta Mattavelli[7], Patrick Eiring[8], Neha Vashist[1,2], Florian Wedekink[1], Sabrina Genssler[2], Birgitt Fischer[1,2], Julia Dahlhoff[9], Fatemeh Mokhtari[9], Anastasia Kuzkina[1], Marij J. P. Welters [10], Tamara M. Benz[1], Lena Sorger [1], Vincent Thiemann[1], Giovanni Almanzar[1,11], Martina Selle[1], Klara Thein[1], Jacob Späth [1], Maria Cecilia Gonzalez[2], Carmen Reitinger [12], Andrea Ipsen-Escobedo[12], Kilian Wistuba-Hamprecht [13,14,15], Kristin Eichler[1,2], Katharina Filipski[3,4,5], Pia S. Zeiner[3,4,5,16], Rudi Beschorner [17], Renske Goedemans[10], Falk Hagen Gogolla[18], Hubert Hackl [18], Rogier W. Rooswinkel[19], Alexander Thiem [20,31], Paula Romer Roche[1,2], Hemant Joshi [1,32], Dirk Pühringer[1], Achim Wöckel[1], Joachim E. Diessner[1], Manfred Rüdiger [2], Eugen Leo[2], Phil F. Cheng [21], Mitchell P. Levesque [21], Matthias Goebeler [20], Markus Sauer [8], Falk Nimmerjahn [12], Christine Schuberth-Wagner[2], Stefanie von Felten [22,23], Michel Mittelbronn [24,25,26,27,28,29], Matthias Mehling[30], Andreas Beilhack [9], Sjoerd H. van der Burg[10], Angela Riedel[7], Benjamin Weide[13], Reinhard Dummer [19] & Jörg Wischhusen [1] ✉

[1]Department of Gynecology, University Hospital Würzburg, Würzburg, Germany. [2]CatalYm GmbH, Am Klopferspitz 19, 82152 Munich, Germany. [3]German Cancer Consortium (DKTK), German Cancer Research Center (DKFZ), Heidelberg, Germany. [4]Neurological Institute (Edinger Institute), University Hospital, Goethe University, Frankfurt/Main, Germany. [5]Frankfurt Cancer Institute (FCI), Frankfurt/Main, Germany. [6]Center for Neuropathology and Prion Research, Munich, Ludwig-Maximilians-University, Munich, Germany. [7]Mildred Scheel Early Career Center, University Hospital of Würzburg, Würzburg, Germany. [8]Department of Biotechnology and Biophysics, Julius Maximilians University Würzburg, Am Hubland, 97074 Würzburg, Germany. [9]Department of Medicine II, University Hospital of Würzburg, Würzburg, Germany. [10]Department of Medical Oncology, Oncode Institute, Leiden University Medical Center, Albinusdreef 2, Leiden 2333 ZA, The Netherlands. [11]Department of Pediatrics, University Hospital Würzburg, Würzburg, Germany. [12]Division of Genetics, Department of Biology, University of Erlangen, 91058 Erlangen, Germany. [13]Department of Dermatology, University Medical Center Tübingen, Tübingen, Germany. [14]Department of Immunology, University of Tübingen, Tübingen, Germany. [15]Section for Clinical Bioinformatics, Department of Internal Medicine I, University Medical Center Tübingen, Tübingen, Germany. [16]Dr. Senckenberg Institute of Neurooncology, University Hospital Frankfurt, Goethe University, Frankfurt am Main, Germany. [17]Department of Neuropathology, University of Tübingen, Tübingen, Germany. [18]Institute of Bioinformatics, Biocenter, Medical University of Innsbruck, Innrain 80, 6020 Innsbruck, Austria. [19]Forbion, Naarden, The Netherlands. [20]Department of Dermatology, Venereology and Allergology, University Hospital Würzburg, Würzburg, Germany. [21]Department of Dermatology, University of Zurich, University of Zurich Hospital, Wagistrasse 18, 8952 Zürich, Switzerland. [22]oikostat GmbH, Statistical Analyses and Consulting, Lucerne, Switzerland. [23]Department of Biostatistics, Epidemiology, Biostatistics and Prevention Institute, University of Zurich, Hirschengraben 84, 8001 Zürich, Switzerland. [24]Department of Oncology (DONC), Luxembourg Institute of Health (LIH), Luxembourg, Luxembourg. [25]Luxembourg Centre of Neuropathology (LCNP), Luxembourg, Luxembourg. [26]National Center of Pathology (NCP), Laboratoire National de Santé (LNS), Dudelange, Luxembourg. [27]Luxembourg Centre for Systems Biomedicine (LCSB), University of Luxembourg, Esch-sur-Alzette, Luxembourg. [28]Department of Life Sciences and Medicine (DLSM), University of Luxembourg, Luxembourg, Luxembourg. [29]Faculty of Science, Technology and Medicine (FSTM), University of Luxembourg, Esch-sur-Alzette, Luxembourg. [30]Department of Biomedicine and Neurology Department, University Hospital Basel, 4031 Basel, Switzerland. [31]Present address: Clinic for Dermatology and Venereology, Rostock University Medical Center, Rostock, Germany. [32]Present address: Division of Infectious Diseases, Department of Pediatrics, Washington University School of Medicine, St. Louis, MO 63130, USA. ✉e-mail: Wischhusen_J@ukw.de

