## [Peer Review File · Nature Communications]

Tumor-derived GDF-15 blocks LFA-1 dependent T cell recruitment and suppresses responses to anti-PD-1 treatmentREVIEWER COMMENTS

Reviewer #1 (Remarks to the Author):

In this comprehensive study by Hakke and colleagues, the authors address the role of the TGF β family member GDF-15 in suppression of anti tumor T cell activities and raises a new concept, namely that secretion of this cytokine in the tumor microenvironment (TME) promotes a novel mechanism of tumor evasion at least in some types of tumors. To this end, the authors use elegant sets of in vitro read-outs with nice clinical data from different groups of cancer patients. Overall, this is an interesting study but I have several reservations which need to be carefully addressed experimentally or textually.

GDF-15 is a known inhibitor of LFA-1 activation- a classical paper by Kempf (Nature Medicine, 2011) has dissected the molecular basis of this inhibition, especially on neutrophils. The finding that GDF-1 inhibits T cell infiltration via suppression of LFA-1 is therefore not particularly novel. What is intriguing is the final outcome- the apparently poor infiltration and killing functions of CTLs on tumor targets combined with the elevated levels of this cytokine in tumors, but the mechanistic details of these observations are not systematically dissected in the tumor mice models presented in this work.

Major comments

1. CXCL12 is not physiologically relevant for tumor killing T cells- although a nice tool, a CXCL12 based LFA-1 activation readout is insufficient. Matching experiments assessing CXCR3-mediated LFA-1 activation of adhesion (using canonical tumor chemokines such as CXCL9 or CXCL10 will be nice to include.
2. Tregs express LFA-1 and could be also a subject of GDF-15 inhibition. It would be nice to confirm it in vitro. If indeed Tregs are as sensitive to GDF-15 as CTLs are, enumerating Tregs inside tumors expressing elevated GDF-15 should also address this question. If Tregs are insensitive to GDF-15, this could be a valuable result.
3. The idea to express human GDF-15 in a murine tumor is elegant, especially for the purposes of direct GDF-15 blocking experiments but it is critical to repeat these experiments with lines that express the murine analogue and show that its effect resemble those of the human cytokine since after all the responding immune cells in the TME are murine and not human. Another complication of this model is the induction of anti-human GDF-1 in some of the mice. This introduces noise and complicates the interpretation.
4. The effects of the GDF-1 overexpression on tumor growth are different- for instance 3B vs. 4D. Are these different protocols? This deserves careful explanations.
5. A major difficulty in interpreting these data is the notion that GDF-15 may not only interfere with T cell LFA-1 but with LFA-1 and with activation of additional β 2 integrins expressed by other leukocytes. A rigorous analysis of other subsets of cells in the TME is required to either exclude this possibility or highlight the pleotropic functions of GDF-15. For instance could GDF-15 affects tumor killer macrophages (M1) more than M2 macrophages?
6. The in vitro inhibition studies with anti LFA-1 and GDF-15 are convincing but it is essential to perform also combination experiments to further strengthen these data. For instance, in the presence of LFA-1 blocking, does GDF-15 exert any additional inhibitory activity on T effector adhesion? I wouldn't think so but it is absolutely required to demonstrate it.
7. Elegant dSTORM experiments that aim to quantify LFA-1 activation were performed and presented in Figure 2 but the images are unclear and very hard to follow. The rationale of using this fancy methodology is also unclear- FACs based assays with β 2 integrin conformation reporter mAbs are as sensitive and are commonly used to assess interference with chemokine mediated inside-out LFA-1 activation.
8. Is human GDF-1 as potent as murine GDF-1 towards murine effector T cells? This must be shown in vitro.
9. In some of the figures, one of the donors doesn't respond to LFA-1 inhibition or to GDF-15 treatment. This puzzling result is not explained. Either more examples must be provided and if multiple donors share this binary behavior, I'd consider dividing these figures to two subgroups of patient T cells. As is, it is a strange result given that all T cells express LFA-1 and are expected to bind ICAM-1 in vitro with similar efficacy.
10. There are no discussions on how GDF-15 affects the composition of tumor draining lymph

nodes (skin). Adoptive transfer experiments with CFSE labeled CD8 and CD4 populations will allow to track both the homing and initial activation of naïve T cells in TdLNs. Memory polyclonal T cells raised against the tumor in donor mice should be also tested.

11. GDF-15 seem to exert multiple effects on the tumor environment via metabolic changes which are associated with cachexia (Fig. 3a). The authors don't attempt to link this finding to LFA-1 inhibition or to interference with T cell infiltration. This is puzzling and confusing.

12. The use of humanized mice in this study is questionable and should be re-considered since it is a very complex experimental system. I don't see what advantages it has over immunocompetent mice since as noted, among other deficiencies APCs in these mice are scarce or abnormal. It is not well explained how the different T cells emerge in this system, and how they distribute in the spleen and blood. I'd consider removing this model and describing it in another paper.

13. The clinical data and especially the correlations between elevated serum levels of GDF-15 and survival are nice and statistically significant. Nevertheless, in light of the multiple effects of this cytokine it is hard to directly link these results to the main theme of the paper which is LFA-1 inhibition and attenuation of trafficking and function of tumor killer immune cells in the TME. So is the inclusion of data from pregnant women. It is unclear how these data strengthen the main theme of this work. I think it does the very opposite.

Minor comments

1. The terminology 'recruitment' used in the legend of Fig. 1 is confusing. In vitro readouts do not involve recruitment. Please change it. The explanation of these in vitro readouts in the main text should be also improved.

2. The title "GDF-15 acts on naïve and memory CD8+, and on CD4+ memory T cells" must be revisited. As far as I can tell, naïve T cells are less responsive to GDF-15.

3. The mentioning of "catch bonds" is relevant to the discussion but not to the result section since this biophysical property is not directly measured in this study.

4. Fig. 4a- the intensity of staining (CD45 and CD8) is very different for the different tumors. Please explain.

5. Adjuvant immunization of tumor bearing mice is mentioned in the methods but never discussed in the main text. Why?

6. Why can't alternative adhesive pathways such as VLA-4-VCAM-1 compensate for LFA-1 inhibition in the various models. This point has to be discussed.

7. LFA-1 inhibition by GDF-1 may result in reduced functionality due to poor ability of the CTLs to generate killing synapses with tumor cells? This point should be further emphasized.

Reviewer #2 (Remarks to the Author):

In this study, Haake and colleagues have investigated the influence of GDF-15 on T-cell recruitment within the tumor microenvironment and on the response to anti-PD-1 immunotherapy. In the first part of the work, the authors provided in vitro evidence of a role of GDF-15 in interfering with the LFA-1/ICAM1 interaction, which is known to be essential for T-cell migration and extravasation. In a second part, using the murine MC38 colon cancer model, they attempted to functionally validate these observations in vivo, and investigated the consequences of GDF15 overexpression on T-cell infiltration and response/resistance to anti-PD1 therapy. In addition, they assessed the potential of using anti-GDF15 to improve anti-PD-1 effect in this setting. Data confirmed the prognostic value of GDF15, and a trend to a synergistic effect for the anti-PD1/anti-GDF15 combination was observed in this model. In the last part, the authors examined various datasets and cohort of patients (Melanoma and oropharyngeal squamous cell carcinomas) to test the impact of GDF15 expression levels (by IHC and serum measures) on patient clinical outcomes and tumor T-cell infiltration. They show a positive association of GDF-15 with poor survival, and inverse correlation with T-cell infiltrates (CD8, CD3, Treg). They also report evidence in two series of metastatic melanoma patients that elevated GDF-15 serum levels are associated with resistance to PD-1 blockade monotherapy, suggesting GDF-15 as a novel predictive biomarker in this cancer type.

The topic is of interest and original. The study provides results of relevance for the onco-immunology field. However, several technical and conceptual concerns need to be addressed.

Major comments:

1- in the first part of the manuscript, in vitro experiments and associated conclusions derive from T cells purified from healthy donor PBMCs. The significance would be stronger if the authors could expand the work and confirm the findings using T lymphocytes from cancer patients.

2- page 8 and Supp. Figure S2, authors claim that "GDF-15 acts on naïve and memory CD8+, and on CD4+ memory T cells". But it is unclear how they have differentiated the memory subsets from naïve T cells. Information is missing in the method and Figure. It would be important to provide flow cytometry data to support the observed differences and statements (GDF15/anti-LFA1 treatments). Of note, in figure 2, adhesion data are shown for pan-, naïve-, Memory and EM (effector memory) CD4+ T cells. However, for CD8+ cells, the EM category is missing. Moreover, the quality of the images in panels a and b needs to be improved.

3- Data in figure 3 are difficult to interpret. For example, sample size and statistics in panels c, f, g make the current data questionable and hard to interpret (with 2-to-4 points in each group)

4- Fig 3i, j, k, tumor growth curves for each treatment are presented as various sets. It would be keen for the authors to include combined curves to help visualization and interpretation of the added value of these treatments relative to the control condition.

5- Figure 3k and Page 10 (L14), authors claim "Adding anti-GDF-15 to anti-PD-1, however, resulted in complete clearance of MC38hGDF-15 tumors in 4/10 mice (Fig. 3k), indicating that anti-GDF-15 and PD-1-based immune checkpoint blockade synergize to increase survival in mice". This is a key statement but this relies only on 4 events, with no significant differences for survival between the two conditions ($p=0.17$).

This could be an interesting observation, but parts of the experiments should be repeated to foster the significance before to draw firm conclusions on a potential synergistic effect of their drug. Another improvement would be to assess the anti-PD-1/anti-GDF15 combination in another murine syngeneic model, as the data presented in humanized mice did not address the anti-PD-1/-GDF15 combination.

6- Figure 4, regarding analysis of T-cell infiltration (e.g. in panels f, i, j), the sample size and statistics make it hard to interpret/conclude (2-to-4 points in each condition). It's unclear why the sample size is so small, compared to the number of mice analyzed in the previous figure (10 in each treatment group). These experiments need to be confirmed with at least 6-to-10 mice per group?

7- Figure 4 m-n data: as indicated in the legend, CD45 and CD3 data presented in panel (m) derived from two independent experiments, and in (n), results from the second experiment only are presented to inform on the CD4 and CD8 infiltrates. However, (n) shows data from the control condition derived from 1 mouse. Therefore, it's unclear from where come the measures in (m). For clarification, it would be reasonable to show the distribution of CD4 and CD8 from the two independent experiments.

8- Page13 L20: authors claim "GDF-15 hence represents an independent marker for failure of anti-PD-1 therapy" It is not clear whether the authors have done a multivariate analysis to conclude that GDF-15 separates from other known risk factors. This would be an important point given the poor HR and significant p values presented in Table S2. If so, other variables considered in the model should be presented with HR and p values.

9- Also, have the authors examined the potential association with PD-L1? Or test if GDF-15 is a better biomarker than PD-L1?

10- Figure 6: For clarity, authors should annotate with the name of each cohort and insert the sample size "in each subgroup". Some dots have been removed in some conditions like in panel b, but it is important to know how many patients have been included for statistical analyses.

11- Discussion, p16 and abstract: author claim: "We show that tumor-derived GDF-15 inhibits LFA-1 activation on human cytotoxic T cells, thereby interfering with T cell adhesion, rolling and

diapedesis”.

Using the term “cytotoxic” in this sentence sounds like an overstatement as experiments were not performed to test the cytotoxic features of effector cells (cytolytic markers, cytotoxic activity of CTL or NK cells, etc...). Moreover, as mentioned above, the mechanistic part (GDF15/LFA1) was performed with peripheral T lymphocytes from healthy donors.

12- as shown for the MC38 model (Fig. 4h), it would be useful to provide IHC staining from the humanized model to visualize the CD8 T-cell infiltrates in the isotype and anti-GDF15 conditions.

Minor comments:

13- figure 4 a, h: the quality of the photomicrographs should be improved. Additionally, in h, as in a, authors should indicate what immunostaining was performed.

14- Authors claims no relationship was found between GDF15 expression and tumor mutational burden (supp Figure 8). Since GDF-15 is a downstream target of TP53, it would be interesting to include a comparison of the GDF-15 low/high groups in patients stratified for TP53 status and assess if it could relate to ICB response.

15- page 8 L15: to identify the T-cell subsets affected by GDF-15, purified immune cells were briefly exposed to GDF-15. If appropriate, authors should specify “isolated from healthy donor PBMC”.

16- Figure 3h: is for survival comparison, authors should edit the Y axis legend to make it clear.

17- discussion should be improved: for instance, authors should better highlight what is new in their study with respect to previous reports on the link between GDF-15 and tumor immune escape (ref. 23 and 27). They also need to clearly state the limitations of their study and discuss the characteristic of the cohort (e.g. Pts with Brain metastasis) relative to other types of patients treated with ICB. A discussion about the potential of GDF15 as a biomarker for prognostic or predictive.

18- Method p23 immunohistochemistry: authors should indicate who performed and if the analyses were blinded to the treatment outcomes.

19- Figure 5 legend and P23 method: “Formalin-fixed paraffin-embedded tissues from 70 patients with melanoma brain metastases”, it is unclear if the tissues analyzed are from brain metastases or primary tumor tissues obtained from metastatic patients.

20- P1 L3, “Having established a link between GDF-15 and T cell migration in mice, we assessed GDF-15 expression”: T-cell “infiltration” or “adhesion” seems more appropriate

21- The statement in the title « Tumor-derived GDF-15 acts as roadblock for LFA-1-dependent T-cell recruitment and suppresses response to immune checkpoint inhibition» is entirely proven since the outcome values for anti-CTLA-4 in the studied cohort were not always conclusive. Moreover, anti-CTLA-4 treatment has not been investigated in the MC38 model. Authors should consider revising the title and adjusting stated claims

Reviewer #3 (Remarks to the Author):

The manuscript by Haake and colleagues postulates a new mechanism by which GDF-15 contributes to immune evasion by tumor cells, namely suppression of LFA-1 dependent recruitment. They also show the potential clinical significance of this by providing data that suggest GDF-15 suppresses responses evoked by anti-PD-1 therapy. The strengths of this manuscript are its novelty given that investigations into the contribution of GDF-15 into tumor immunology are relatively rare (PMIDs 20534737, 32508832) and the use of syngeneic murine models, humanised mice models and patient samples to confirm these effects. The main

weaknesses is the lack of evidence suggesting that the mechanism by which GDF-15 suppresses immune responses in vivo is T cell intrinsic and trafficking dependent, much less that this is LFA-1 specific.

Major comments

- The manuscript shows some evidence that GDF-15 affects T cell migration in vitro but there is limited evidence that a) Therapeutic effects in vivo are T cell dependent b) Effects are T cell intrinsic or c) mediated through T cell trafficking. Some evidence is presented pertaining to point c in Figure 4 but this is performed in a very limited cohort size and would need a greater n number to be convincing.
- The authors provide data inferring that MC38 tumors can be further sensitised to anti-PD-1 therapy through blockade of GDF-15. However, MC38 is already quite PD-1 sensitive and therefore it would add further significance to the paper if results could be replicated in a second syngeneic model that is less sensitive to anti-PD-1 at baseline.
- One point of concern is that there appears to be major inconsistencies in tumor growth between experiments even when the same conditions are tested. For example in Figure 3H approximately 50% of parental MC38 tumors are cleared but in 3i this is 0%. Similarly in Figure 3J MC38tgGDF-15 tumors (vehicle treated) uniformly reach 2000mm³ by approximately day 25 but this appears to be <500mm³ in 3K. I could not see a reason for these differences, can this be explained?
- In Figure 5 and 6 the authors present very interesting patient data that indicates patients with elevated GDF-15 have poorer overall survival. Of course with any such dataset one must keep in mind responses may be correlative but not necessarily causative. With this in mind, to what extent do GDF-15 levels correlate with TGFbeta (part of the same family and known to lead to an immune excluded phenotype) and does this explain these observations in patients?
- The authors should discuss why they chose to overexpress human GDF-15 in the mouse model as opposed to murine GDF-15. The immune response mounted against human GDF-15 is clearly a caveat of the current work.

Minor comments

LFA-1 is also important for T cells binding to tumor cells and antigen presenting cells. Did the authors consider/ evaluate whether these parameters are affected by GDF-15? Simple in vitro analyses could be performed e.g. killing assays, conjugate assays to address this.

In terms of the T cell intrinsic nature of the in vivo effects, the authors indicate that the GFRAL receptor is not expressed on lymphocytes but the mechanism by which they are affected is not really discussed. Could the authors elaborate on this topic.

Figure 2a/b the staining is difficult to see. Is it possible to provide more convincing images for this effect?

Point-by-point reply to the reviewer comments

Reviewer #1 (Remarks to the Author):

In this comprehensive study by Haake and colleagues, the authors address the role of the TGF β family member GDF-15 in suppression of anti-tumor T cell activities and raises a new concept, namely that secretion of this cytokine in the tumor microenvironment (TME) promotes a novel mechanism of tumor evasion at least in some types of tumors. To this end, the authors use elegant sets of in vitro read-outs with nice clinical data from different groups of cancer patients. Overall, this is an interesting study, but I have several reservations, which need to be carefully addressed experimentally or textually.

GDF-15 is a known inhibitor of LFA-1 activation- a classical paper by Kempf (Nature Medicine, 2011) has dissected the molecular basis of this inhibition, especially on neutrophils. The finding that GDF-15 inhibits T cell infiltration via suppression of LFA-1 is therefore not particularly novel. What is intriguing is the final outcome- the apparently poor infiltration and killing functions of CTLs on tumor targets combined with the elevated levels of this cytokine in tumors, but the mechanistic details of these observations are not systematically dissected in the tumor mice models presented in this work.

Major comments

1. CXCL12 is not physiologically relevant for tumor killing T cells- although a nice tool, a CXCL12 based LFA-1 activation readout is insufficient. Matching experiments assessing CXCR3-mediated LFA-1 activation of adhesion (using canonical tumor chemokines such as CXCL9 or CXCL10) will be nice to include.

We thank the referee for this suggestion. We have now cited a very recent publication on CXCL9 and CXCL10 driving T cell trafficking to tumors (Hoch, T., et al. Multiplexed imaging mass cytometry of the chemokine milieu in melanoma characterizes features of the response to immunotherapy. Science Immunology 7, eabk1692 (2022), new reference 34). Consequently, we have also included an experiment to show that GDF-15 impairs (CD8⁺) T cell adhesion to HUVEC also in the presence of CXCL9 + CXCL10 (new Figure 1e and data below. Different colors depict different donors).

Figure R1: GDF-15 impairs adhesion of activated T cells to stimulated HUVEC in the presence of CXCL9 and CXCL10. Phase contrast microscopy in chamber slides was used to enumerate T cells that adhere to HUVECs under flow conditions, in the absence or presence of rhGDF-15 or the blocking anti-LFA-1 antibody TS1/18.

Please note: While effects of GDF-15 on T cell adhesion to HUVECs in the presence of CXCL9 and CXCL10 may appear slightly smaller than the corresponding effect in the presence of CXCL12, effects of anti LFA-1 were in a similar range as those achieved with GDF-15 (please see above), and the *p* value is highly significant ($p < 0.0001$). We thus consider this variability to be donor- and assay-dependent, and abstain from discussing effect sizes in the context of CXCR3 or CXCR4 mediated T cell adhesion. As shown below, CXCL9- and CXCL10-induced CD8⁺ T cell adhesion to HUVECs can not only be inhibited by GDF-15. The inhibitory effect of GDF-15 can also be reverted by our anti GDF-15 antibody CTL-002, but not by an isotype control antibody. GDF-15 can thus exert highly significant and specific effects in the presence of different chemokines. As CXCL12 is indeed a nice tool for *in vitro* assays, and as the generated data were now also reproduced with CXCL9 and CXCL10, and as all these findings are also supported by additional tumor models *in vivo*, there can be no reasonable doubt that GDF-15 impairs T cell adhesion to endothelial cells.

Figure R2: GDF-15 – mediated inhibition of T cells adhesion to activated HUVEC can be reverted by the blocking anti-GDF-15 antibody CTL-002. This assay was performed in the presence of the adhesion-stimulating chemokines CXCL9 + CXCL10.

2. Tregs express LFA-1 and could be also a subject of GDF-15 inhibition. It would be nice to confirm it *in vitro*. If indeed Tregs are as sensitive to GDF-15 as CTLs are, enumerating Tregs inside tumors expressing elevated GDF-15 should also address this question. If Tregs are insensitive to GDF-15, this could be a valuable result.

The referee is right in asserting that LFA-1 was described to be critical for regulatory T cell homeostasis and function (Wohler, J., Bullard, D., Schoeb, T. & Barnum, S. LFA-1 is critical for regulatory T cell homeostasis and function. *Mol Immunol* 46, 2424-2428 (2009)). GDF-15 might therefore also affect Treg. As already indicated in the original Figure 5b-d, there is an inverse correlation between the immunohistochemical GDF-15

score and the number of intratumoral *Foxp3*⁺ cells ($\rho=-0.285$, $p=0.017$). However, correlation coefficients and corresponding *p* values indicate that the inverse correlation between GDF-15 and total CD3⁺ ($\rho=-0.372$, $p=0.0015$) and CD8⁺ T cells ($\rho=-0.371$, $p=0.0016$) are even more pronounced. *In vitro*, flow adhesion assays with purified, expanded and activated human Treg (now shown in Supplementary Figure 2h) confirmed that GDF-15 also affects Treg adhesion to huLEC. Again, the effect was weaker than for effector T cells (compare Figure 1c). Still, these data imply that blocking GDF-15 might also recruit Treg into tumors. On the other hand, GDF-15 was recently reported to stabilize *Foxp3* expression in T cells, resulting in more and functionally enhanced Treg in hepatocellular carcinoma (Wang, Z., et al. GDF15 induces immunosuppression via CD48 on regulatory T cells in hepatocellular carcinoma. *J Immunother Cancer* 9(2021), new reference 25). GDF-15 inhibition may thus enhance Treg cell trafficking to tumors, but also impair intratumoral Treg functionality. Accordingly, the overall effect of GDF-15 blockade on Treg-mediated tumor immune evasion is difficult to predict. This is now also discussed (page 9, lines 8-11).

3. The idea to express human GDF-15 in a murine tumor is elegant, especially for the purposes of direct GDF-15 blocking experiments, but it is critical to repeat these experiments with lines that express the murine analogue and show that its effect resemble those of the human cytokine since after all the responding immune cells in the TME are murine and not human. Another complication of this model is the induction of anti-human GDF-15 in some of the mice. This introduces noise and complicates the interpretation.

*We thank the referee for recognizing the rationale underlying overexpression of GDF-15 in an otherwise PD-1-sensitive murine tumor cell line. Regarding inter-species cross-talk, there is ample evidence from the literature that human GDF-15 also acts on mouse cells. In their landmark paper (Kempf, T., et al. GDF-15 is an inhibitor of leukocyte integrin activation required for survival after myocardial infarction in mice. *Nat Med* 17, 581-588 (2011)) Kempf et al. rescued GDF-15^{-/-}-mice by administering recombinant human GDF-15. Moreover, as shown in Figure 3a and described by Johnen et al. (Johnen, H., et al. Tumor-induced anorexia and weight loss are mediated by the TGF-beta superfamily cytokine MIC-1. *Nat Med* 13, 1333-1340 (2007)) and many others, human GDF-15 also induces cachexia in mice. Therefore, there is little doubt regarding cross-species activity of GDF-15. However, the induction of noise due to anti-human GDF-15 antibodies in some of the mice turned out to be a confounder. In order to rule out pertaining doubts, additional experiments have now been performed in the syngeneic orthotopic Panc02 pancreatic cancer and EMT6 breast cancer mouse models, which both show endogenous GDF-15 expression (new Figures 4a-i).*

4. The effects of the GDF-15 overexpression on tumor growth are different- for instance 3B vs. 4D. Are these different protocols? This deserves careful explanations.

In Figure 3B, MC38^{blank} and MC38^{tgGDF-15} cells were injected in immunodeficient NCI^{mu/nu}-mice, in the former Figure 4d (now 3l) MC38^{blank} and MC38^{tgGDF-15} cells were injected in immunocompetent C57Bl6/J mice. The different effects indicate that tumor-promoting effects of transgenic GDF-15 can only be observed in the presence of a functional immune system. We have now rearranged Figure 3 to more prominently show the different tumor growth in nude vs. immune-competent mice (3b vs. 3d). Figure 3l provides another example for the growth advantage of MC38^{tgGDF-15} cells in immune competent mice (page 10, lines 2-6). We hope this clarifies the issue.

5. A major difficulty in interpreting these data is the notion that GDF-15 may not only interfere with T cell LFA-1 but with LFA-1 and with activation of additional $\beta 2$ integrins expressed by other leukocytes. A rigorous analysis of other subsets of cells in the TME is required to either exclude this possibility or highlight the pleiotropic functions of GDF-15. For instance could GDF-15 affects tumor killer macrophages (M1) more than M2 macrophages?

*While we cannot prove that GDF-15 acts on LFA-1 only, the effects described in this manuscript can all be explained by the action of GDF-15 on LFA-1 in T cells. Nevertheless, the referee is absolutely right that inhibition of LFA-1 will also affect further cell types. There are also published data that GDF-15 impairs macrophages (Ratnam, N.M., et al. *NF-kappaB regulates GDF-15 to suppress macrophage surveillance during early tumor development. J Clin Invest* 127, 3796-3809 (2017)) and dendritic cells (Zhang, Y., et al. *GDF15 Regulates Malat-1 Circular RNA and Inactivates NFkappaB Signaling Leading to Immune Tolerogenic DCs for Preventing Alloimmune Rejection in Heart Transplantation. Front Immunol* 9, 2407 (2018)). We also find that adding GDF-15 to LPS and IFN- γ during the M-CSF-induced polarization of monocytes skews the resulting macrophages towards a phenotype characterized by lower expression of CD86 and HLA-DR (see below).*

Figure R3: Characterization of M-CSF-induced, monocyte-derived macrophages that were matured with IFN- γ , LPS and (where indicated) rhGDF-15. Expression of maturation markers was assessed by flow cytometry.

Nevertheless, this (as the referee writes already “comprehensive”) study is about T cell-related effects of GDF-15, as these effects are novel and presumably relevant. A description of the many different effects exerted by GDF-15 on different cell types would by far exceed the limitations of a research article and could easily provide material for a special issue. We have, however, now further emphasized the pleiotropic effects of GDF-15 and clarified that effects of GDF-15 on other cell types can also contribute to the immune-paralyzing effects of GDF-15 (e.g. page 17, lines 2-3, page 20, lines 1-2).

6. The *in vitro* inhibition studies with anti LFA-1 and GDF-15 are convincing but it is essential to perform also combination experiments to further strengthen these data. For instance, in the presence of LFA-1 blocking, does GDF-15 exert any additional inhibitory activity on T effector adhesion? I wouldn't think so but it is absolutely required to demonstrate it.

*We thank the referee for this question. As GDF-15 has also been described to reduce agonist-induced activation of β_1 - and β_3 -integrins on platelets (Rossaint, J., Vestweber, D. & Zarbock, A. GDF-15 prevents platelet integrin activation and thrombus formation. *J Thromb Haemost* 11, 335-344 (2013)), effects of GDF-15 on other integrins appear plausible. The cited study is, however, still awaiting independent confirmation. In order to address this issue we have now performed T cell-based flow adhesion assays on stimulated HUVEC, which are likely to address most physiologically relevant cell adhesion molecules. Activated CD4⁺ or CD8⁺ T cells were either left untreated, or pre-treated with an anti-LFA-1 antibody (TS1/18), or with GDF-15, or both (new Figure 2d,e). While all three treatments resulted in a highly significant inhibition of T cell adhesion, there was no additional effect by adding GDF-15 to the anti LFA-1 antibody. Thus, the combination experiments show that GDF-15 does not affect other adhesion molecules expressed on HUVEC. Of note, while anti LFA-1 was even more potent than GDF-15 in inhibiting T cell adhesion to HUVEC, effects of GDF-15 were consistently highly significant.*

7. Elegant dSTORM experiments that aim to quantify LFA-1 activation were performed and presented in Figure 2 but the images are unclear and very hard to follow. The rationale of using this fancy methodology is also unclear- FACS based assays with β_2 integrin conformation reporter mAbs are as sensitive and are commonly used to assess interference with chemokine mediated inside-out LFA-1 activation.

*We have performed FACS-based assays with the conformation-specific β_2 -integrin antibody mAb24, and with rhICAM-1-Fc. However, these assays revealed no significant effects of GDF-15 (new Figure 2h,i). In contrast to what Kempf et al. ((Kempf, T., et al. GDF-15 is an inhibitor of leukocyte integrin activation required for survival after myocardial infarction in mice. *Nat Med* 17, 581-588 (2011)) have described in their Fig. 5a for human PMNs, the GDF-15 obtained by us apparently does not lock LFA-1 in a closed conformation on stimulated T cells. Nevertheless, LFA-1 can also assume a loosely open conformation with incrementally lower binding affinity (Nordenfelt, P., Elliott, H.L. & Springer, T.A. Coordinated integrin activation by actin-dependent force during T-cell migration. *Nat Commun* 7, 13119 (2016)). While the resulting small differences in ICAM-1-Fc or mAb24 binding will likely be obliterated by the spread of the respective flow cytometric peaks, single-molecule based dSTORM microscopy achieves a higher power of resolution, leading to a more accurate quantification. As this indeed revealed a slightly reduced binding of ICAM-1 and m24 in the presence of GDF-15 (Figure 2j), our observations are still very much in line with the landmark paper by Kempf et al., who have first described effects of GDF-15 on LFA-1 activation. Therefore, we would like to keep the dSTORM data as part of our manuscript. We hope that the improved visual representation and the additional binding assays shown in the revised Figures 2h,i now help the reader to better understand the findings. In addition, we have now also analyzed Talin phosphorylation in activated T cells that were either treated or not with rhGDF-15 before being added to bead-bound immobilized ICAM-1-Fc. The observed differences in Talin S425 phosphorylation (new Figures 2k,l) show that GDF-15 disrupts the link from immobilized ICAM-1 via LFA-1 to the actin*

cytoskeleton. As Talin phosphorylation is (in this experimental setup) triggered by immobilized ICAM-1 binding to LFA-1, the signaling pathway could be interrupted either at the level of LFA-1 binding to ICAM-1, or intracellularly by uncoupling LFA-1 from the actin cytoskeleton. While a precise elucidation of the intracellular signaling is beyond the scope of the current manuscript, the lack of effect on ICAM-1 or m24 binding when assessed by flow cytometry (Figure 2f,g), the weak effect observed via dSTORM microscopy (Figure 2h-j), and the strong effect on Talin phosphorylation (Figure 2k,l) suggest that GDF-15 likely destabilizes the linkage of LFA-1 to the actin cytoskeleton – which would be in line with the very slight effects on ICAM-1-Fc binding, while also explaining the strong effects on adhesion to ICAM-1.

8. Is human GDF-15 as potent as murine GDF-15 towards murine effector T cells? This must be shown in vitro.

Comparing the potency of human vs. murine GDF-15 would require GDF-15 preparations with standardized (or, at least, reliably quantified) immunomodulatory bioactivity as starting material. Unfortunately, none of the current manufacturers has such an assay. The flow adhesion assay depends on primary T cells, which inevitably introduces some variety. It is also too complex and labor-intensive to be widely used. Still, using this and another proprietary assay we have observed >10-fold differences in bioactivity between different batches of rhGDF-15 from the same manufacturer. We further find a decline in immunomodulatory GDF-15 bioactivity with storage. Thus, we would not be able to say whether different bioactivities are due to a general difference in potency between human and murine GDF-15, or to a different bioactivity of the respective GDF-15 aliquots used for the comparison. There are, however, in vivo data from Kempf et al. showing that rhGDF-15 can substitute for mGDF-15 in GDF-15^{-/-} mice (Kempf, T., et al. GDF-15 is an inhibitor of leukocyte integrin activation required for survival after myocardial infarction in mice. Nat Med 17, 581-588 (2011)). These, and our data clearly show that human GDF-15 is active towards murine immune cells. Interestingly, Weng et al. (Weng, J.H., et al. Colchicine acts selectively in the liver to induce hepatokines that inhibit myeloid cell activation. Nat Metab 3, 513-522 (2021)) have been unable to restore GDF-15-dependent anti-inflammatory effects by applying recombinant murine GDF-15 to GDF-15^{-/-} mice, which has prompted a discussion on the bioactivity of different GDF-15 preparations (Lockhart, S.M. & O’Rahilly, S. Colchicine—an old dog with new tricks. Nature Metabolism 3, 451-452 (2021)). Considering that some preparations of human GDF-15 can rescue effects of a GDF-15 knock-out in mice, while some preparations of murine GDF-15 fail to do so, it is clear that GDF-15 bioactivity has to be assessed for the respective protein preparation. We have now tried to clarify this important issue (page 9, lines 12-14, page 19, lines 16-19).

9. In some of the figures, one of the donors doesn't respond to LFA-1 inhibition or to GDF-15 treatment. This puzzling result is not explained. Either more examples must be provided and if multiple donors share this binary behavior, I'd consider dividing these figures to two subgroups of patient T cells. As is, it is a strange result given that all T cells express LFA-1 and are expected to bind ICAM-1 in vitro with similar efficacy.

We presume that the referee is referring to Figure S2. In Figures 1a,b,c,f, 2d,e,f, etc., effects of GDF-15 on different donors are qualitatively similar. Puzzling results are, however, sometimes observed when working with recombinant GDF-15, as rhGDF-15

is difficult to handle. Still, when cells from the same donors were assayed again, there were no generally unresponsive donors. This is now explicitly stated (page 9, lines 12-14). Even in the presence of a carrier protein like albumin GDF-15 can rapidly lose its immunomodulatory bioactivity upon storage. This became clear when we compared effects of freshly used vs. stored GDF-15 in flow adhesion assays (similar to the ones shown in Figure 1C). Of note, these differences were also observed when frozen T cells from the same donor were used, which excludes a donor dependency of the effect.

Figure R4: Stimulated T cells were treated or not with fresh (left panel) or re-used (right) aliquots of rhGDF-15 for 20 min before being run in μ -slides over a layer of activated huLEC. 10 predefined fields of view were video-imaged for 5s and the number of T cells adhering under hydrodynamic flow conditions was counted. Data points from different experiments with different T cell donors are depicted differently shaded.

We thus believe that the differences visible in Figure S2 are more likely due to the vulnerable bioactivity of rhGDF-15 rather than to donor-specific differences. One possible explanation is that GDF-15 is an extremely sticky protein that binds to plastics, and can therefore easily be lost upon storage or pipetting. We have now commented on this possible reason for the observed differences (page 9, lines 12-14). Moreover, the recent publication by Weng et al. (Weng, J.H., et al. Colchicine acts selectively in the liver to induce hepatokines that inhibit myeloid cell activation. *Nat Metab* 3, 513-522 (2021)) also describes that recombinant GDF-15 can be inactive. Having seen a number of negative experiments with frozen aliquots of cells from otherwise responsive donors, we are convinced that the difference between a “positive” and a “negative” assay is more likely related to the condition of the GDF-15 used rather than to a specific donor. Interestingly, the activation of GFRAL, which we can measure in a reporter gene assay, is less sensitive to effects of GDF-15 storage. It is therefore conceivable that GDF-15 can undergo proteolytic or oxidative changes, which selectively affect the immunomodulatory bioactivity of GDF-15. Nevertheless, as we have just preliminary data to support this hypothesis, we do not want to indulge in excessive speculations. We would further like to point out that an assay with 3 stimulation conditions, 1 cell type and 2 donors requires one full week (5 days) of work from a technician. A systematic exploration of different donor behavior across different immune cell subsets is thus simply not feasible.

10. There are no discussions on how GDF-15 affects the composition of tumor draining lymph nodes (skin). Adoptive transfer experiments with CFSE labeled CD8 and CD4 populations will allow to track both the homing and initial activation of naïve T cells in TdLNs. Memory polyclonal T cells raised against the tumor in donor mice should be also tested.

We thank the referee for this suggestion. As the network of lymphatic vasculature and lymph nodes responsible for draining the pancreas is extremely complex, with many lymphatic vessels in the interlobular spaces of the pancreas and a not-yet-standardized classification of pancreatic nodes, we could not address this question in the Panc02 model, which we have used for most experiments performed during the course of this revision. We thus chose the GDF-15 expressing EMT6 breast cancer model to address this question and performed adoptive transfer experiments with CFSE-labelled T cells which were then traced in axillary and brachial lymph nodes. Interestingly, GDF-15 neutralization on days 6, 9, and 12 after tumor inoculation greatly improved T cell infiltration, in particular in the more distant brachial lymph node. While the onset of treatment may have been too late to achieve the maximum effect on the axillary lymph node, where tumor antigens arrive early, effects on CD8⁺ T cell lymph node entry were still significant in axillary lymph nodes. We can thus confirm that GDF-15 also affects trafficking of T cells to the lymph nodes where they should get primed (see new Figure 4f-h).

Raising memory polyclonal T cells against the rather poorly immunogenic EMT6 cells (Zhong, W., et al. Comparison of the molecular and cellular phenotypes of common mouse syngeneic models with human tumors. BMC Genomics 21, 2 (2020)) was neither covered by an active application for animal experimentation, nor an established procedure in our or the collaborating lab of Angela Riedel. Therefore, we could not address this suggestion within the current revision. Still, the new experiments showed that blocking GDF-15 by a neutralizing antibody injected from day 6 onward increased CD4⁺ and, in particular, CD8⁺ T cell trafficking to tumor draining lymph nodes, with highly significant differences in brachial lymph nodes (new Figures 4F-I).

11. GDF-15 seem to exert multiple effects on the tumor environment via metabolic changes which are associated with cachexia (Fig. 3a). The authors don't attempt to link this finding to LFA-1 inhibition or to interference with T cell infiltration. This is puzzling and confusing.

We agree with the referee that GDF-15 is sometimes puzzling and confusing. However, recent data show clearly that immunomodulatory effects of GDF-15 do not depend on the cachexia-/anorexia-inducing receptor GFRAL (Weng, J.H., et al. Colchicine acts selectively in the liver to induce hepatokines that inhibit myeloid cell activation. Nat Metab 3, 513-522 (2021)). Klein et al. have further reported that an anorexia-inducing effect is induced by pharmacologically administered GDF-15, but not by GDF-15 that is physiologically produced in exercise-stressed tissue (Klein, A.B., et al. Pharmacological but not physiological GDF15 suppresses feeding and the motivation to exercise. Nat Commun 12, 1041 (2021)).

Moreover, we have also preliminary data with the GDF-15^{V87R} mutant described in Yang et al. (Yang, L., et al. GFRAL is the receptor for GDF15 and is required for the anti-obesity effects of the ligand. Nat Med 23, 1158-1166 (2017), Reference 22). This mutation abrogates binding of GDF-15 to GFRAL (shown by sandwich ELISA with rhGFRAL-Fc used for capture and polyclonal anti-human GDF-15 for detection):

Figure R5: Dose-dependent binding of wild-type or V87R rhGDF-15 to GFRAL was assessed in a sandwich ELISA with rhGFRAL-Fc used for capture and polyclonal anti-human GDF-15 for detection. Optical density as measured by ELISA reader is indicated.

Consequently, this mutant also fails to activate GFRAL/RET signaling as assessed via an SRE reporter gene assay:

Figure R6: HEK293 cells stably transfected with GFRAL/RET51 were stimulated with wild-type or V87R rhGDF-15. The concentration-dependent increase in Erk1/2 phosphorylation was assessed by homogeneous time-resolved fluorescence (HTRF) with Cisbio ERK phospho-T202/Y204 HTRF kit (#64ERKPEG) according to the manufacturer's instructions.

In flow adhesion assays, however, this variant potently interferes with T cell adhesion to huLEC:

Figure R7: Stimulated T cells were treated with different preparations of rhGDF-15 and/or antibodies for 20 min before being run in μ -slides over a layer of activated huLEC. T cells adhering under hydrodynamic flow conditions were counted. Three donors were tested.

Conversely, an anti-GFRAL antibody does not rescue T cell adhesion in the presence of GDF-15:

Figure R8: Stimulated T cells were treated with different preparations of rhGDF-15 and/or antibodies for 20 min before being run in μ -slides over a layer of activated huLEC. T cells adhering under hydrodynamic flow conditions were counted. Three donors were tested.

Thus, tissue-protective, immunomodulatory and metabolic effects of GDF-15 are likely mediated via different receptors. Having seen no signs of cachexia in immunocompetent animals with GDF-15 expressing tumors, we believe that we should not attempt to link different functions of GDF-15 which may, or may not, be related. We have thus added a remark referring to possible metabolic effects of GDF-15 in the tumor microenvironment, but refrained from excessive speculation (page 19, lines 15-17). Based on recent data from the literature, we have also added the statement “Immunomodulatory effects of GDF-15 thus appear to be independent of GFRAL” (page 19, lines 12-15), which is also supported by the most recent literature on GDF-15 (e.g. Weng, J.H., et al. Colchicine acts selectively in the liver to induce hepatokines that inhibit myeloid cell activation. Nat Metab 3, 513-522 (2021)).

12. The use of humanized mice in this study is questionable and should be re-considered since it is a very complex experimental system. I don't see what advantages it has over immunocompetent mice since as noted, among other deficiencies APCs in these mice are scarce or abnormal. It is not well explained how the different T cells emerge in this system, and how they distribute in the spleen and blood. I'd consider removing this model and describing it in another paper.

The project has a clear translational focus. Thus, humanized mice were chosen in order to enable the use of an anti GDF-15 antibody suitable to enter clinical trials. While we agree with the referee that the system has caveats and limitations, it is well-established in translational research. Our co-author Falk Nimmerjahn and his group, who have performed these experiments, are leading experts in this area (Lux, A. & Nimmerjahn, F. Of mice and men: the need for humanized mouse models to study human IgG activity in vivo. J Clin Immunol 33 Suppl 1, S4-8 (2013)). In their competent hands the system is sufficiently well characterized to generate meaningful data. In fact, Referee 3 has even highlighted this model as a particular strength of the manuscript, which strongly argues against removing it. However, additional experiments performed in syngeneic mice with a surrogate antibody against mouse GDF-15 now complement these data.

13. The clinical data and especially the correlations between elevated serum levels of GDF-15 and survival are nice and statistically significant. Nevertheless, in light of the multiple effects of this cytokine it is hard to directly link these results to the main theme of the paper which is LFA-1 inhibition and attenuation of trafficking and function of tumor killer immune cells in the TME. So is the inclusion of data from pregnant women. It is unclear how these data strengthen the main theme of this work. I think it does the very opposite.

We have now removed the data from pregnant women. We also agree with the referee that GDF-15 exerts multiple effects on the immune system, which we have also discussed. Nevertheless, we respectfully point out that the main topic of the manuscript is the functional relevance of GDF-15 as a new target for cancer immunotherapy. Given the known relevance of CD8⁺ T cell infiltration for responses to anti PD-1, effects of GDF-15 on LFA-1 on T cells provide a very plausible explanation for the clinical findings. The observed effects of T cell trafficking to tumor-draining lymph nodes, which were assessed on request of the referee, provide a further strong rationale for

neutralizing GDF-15 in cancer patients. We also explicitly concede that effects on macrophage polarization or dendritic cell maturation may also contribute to GDF-15-dependent immune escape. Nevertheless, a comprehensive evaluation of all possible mechanisms would be an excellent topic for a dedicated issue of a journal. In a single article for Nature Communications we can only focus on findings that appear most salient with regard to novelty and relevance.

Minor comments

1. The terminology ‘recruitment’ used in the legend of Fig. 1 is confusing. In vitro readouts do not involve recruitment. Please change it. The explanation of these in vitro readouts in the main text should be also improved.

The term recruitment has been replaced by adhesion for all in vitro assays apart from Figure 1i. Figure 1i is part of a sequence depicting different steps in the interaction of T cells with endothelial cells. Here, the terms adhesion (1g), transmigration (1h) and recruitment (1i) (meaning recruitment to the other side of the HUVEC layer) are routinely used. Therefore, we would like to stick to the terminology. However, we hope that all in vitro readouts will now be clear.

2. The title “GDF-15 acts on naïve and memory CD8⁺, and on CD4⁺ memory T cells” must be revisited. As far as I can tell, naïve T cells are less responsive to GDF-15.

We have now performed additional experiments which have added to the previous picture. Initially, we had just observed trends with pan and with naïve CD4⁺ T cells. Now, the $p < 0.05$ level of significance has also been reached for these subsets. In fact, adhesion of naïve CD4⁺ T cells ($p = 0.001$) and of CD8⁺ T cells ($p < 0.0001$) to HUVEC are also greatly impaired. Further, we have also observed an effect of GDF-15 on regulatory CD4⁺ T cells. Therefore, we have now revisited the above-mentioned subtitle and changed it into “GDF-15 broadly affects T cell adhesion to endothelial cells”. The question whether specific subsets are more or less responsive may be addressed in the future. Our current insights are accurately represented by the data shown in Figure S2f. Therefore, we would rather confine ourselves to showing the data and concluding that GDF-15 acts broadly on different T cell subsets (which is also supported by preliminary data from our clinical phase 1 trial). Readers who want to infer further details from the data are, of course, free to do so.

3. The mentioning of “catch bonds” is relevant to the discussion but not to the result section since this biophysical property is not directly measured in this study.

The referee is right in pointing out that this biophysical property was not directly measured. Still, readers who are less familiar with integrin signaling will be surprised that minimal differences in m24 and ICAM-1-Fc binding can translate into strongly reduced adhesion. We have now re-arranged and extended Figure 2. By first showing the loss of adhesion to ICAM-1, then presenting the data which indicate that opening of the “headpiece” of LFA-1 is hardly affected by GDF-15, and finally finding reduced intracellular Talin phosphorylation, we have hopefully managed to clarify that GDF-15 likely affects the linkage of LFA-1 to the actin cytoskeleton.

4. Fig. 4a- the intensity of staining (CD45 and CD8) is very different for the different tumors. Please explain.

*We thank the referee for pointing this out. When labeling the figure, we made a mistake and confused rows and columns. While the upper left and the lower right panel were still correctly assigned, the panel labeled “MC38^{tghGDF-15} CD45” (upper right) was actually “MC38^{blank} CD8”. The panel on the lower left (initially labeled as “MC38^{blank}CD8”) was “MC38^{tghGDF-15}CD45”. Accordingly, the fainter signals come from the anti CD45 staining where myeloid cells may display different intensities of CD45 immunoreactivity (Badie, B. & Schartner, J.M. Flow cytometric characterization of tumor-associated macrophages in experimental gliomas. *Neurosurgery* 46, 957-961; discussion 961-952 (2000), Brandenburg, S., et al. Myeloid cells expressing high level of CD45 are associated with a distinct activated phenotype in glioma. *Immunol Res* 65, 757-768 (2017)), while the very prominent signals were obtained with the anti CD8a antibody. This mistake has now been corrected in the revised Figure (now 3i). In addition, we have slightly adjusted the color/brightness on the complete images to improve the visual impression.*

5. Adjuvant immunization of tumor bearing mice is mentioned in the methods but never discussed in the main text. Why?

We apologize for a lack of clarity. The term “adjuvant immunization” referred to immune stimulatory treatment with poly(I:C) and anti-CD40 antibody. However, as the newly added data direct the focus of the manuscript even more on immune infiltration and on the synergy between anti GDF-15 and anti PD-1, we have now omitted the data obtained with (poly(I:C) + anti-CD40) treatment, as these might rather distract the reader. (Moreover, we have also reached the word limit allowed by the journal.) Nevertheless, if the referee feels that these data should still be incorporated, we could also add them again to the manuscript.

6. Why can't alternative adhesive pathways such as VLA-4-VCAM-1 compensate for LFA-1 inhibition in the various models. This point has to be discussed.

*Effects of GDF-15 on LFA-1 occur within seconds to minutes, which leaves little room for compensatory mechanisms to be activated. Moreover, expression of integrin ligands differs between different tissues, which limits the potential for compensation through other integrins. Finally, an early study on mice lacking the LFA-1 alpha chain CD11a found that LFA-1-deficient mice did not reject immunogenic tumors grafted into footpads and did not demonstrate priming response against tumor-specific antigen. The observation that these mice mounted normal T cell responses to lymphocyte choriomeningitis virus (LCMV) and vesicular stomatitis virus (VSV) infections and exhibited normal ex vivo CTL function indicates that compensatory mechanisms may exist for some, but not for all functions of LFA-1 (Schmits, R., et al. LFA-1-deficient mice show normal CTL responses to virus but fail to reject immunogenic tumor. *J Exp Med* 183, 1415-1426 (1996), new reference 13). A very recent reference also shows that LFA-1 activation enriches tumor-specific T cells in an immunologically “cold” tumor model and synergizes with CTLA-4-based immune checkpoint blockade (Hickman, A., et al. LFA-1 activation enriches tumor-specific T cells in a cold tumor model and synergizes with CTLA-4 blockade. *J Clin Invest* 132(2022), new reference 14). Therefore, it would be very surprising if an almost instantaneous inhibition of the LFA-1:ICAM-1 axis could be compensated for.*

7. LFA-1 inhibition by GDF-15 may result in reduced functionality due to poor ability of the CTLs to generate killing synapses with tumor cells? This point should be further emphasized.

The referee is right that LFA-1 also stabilizes immunological synapses. However, while this is likely true for killing synapses between CTLs and ICAM-1 expressing targets, reference data on www.proteinatlas.org indicate that “most malignant tissues were negative (for ICAM-1). Rare cases of squamous cell carcinoma, liver cancer and breast cancer showed moderate to strong cytoplasmic and membranous immunoreactivity.” Thus, effects on killing synapses will be highly variable. ICAM-1 can, however, greatly stabilize immune synapses between dendritic cells and T cells. This is now also mentioned in page 19, lines 21-22.

Reviewer #2 (Remarks to the Author):

In this study, Haake and colleagues have investigated the influence of GDF-15 on T-cell recruitment within the tumor microenvironment and on the response to anti-PD-1 immunotherapy. In the first part of the work, the authors provided in vitro evidence of a role of GDF-15 in interfering with the LFA-1/ICAM1 interaction, which is known to be essential for T-cell migration and extravasation. In a second part, using the murine MC38 colon cancer model, they attempted to functionally validate these observations in vivo, and investigated the consequences of GDF15 overexpression on T-cell infiltration and response/resistance to anti-PD1 therapy. In addition, they assessed the potential of using anti-GDF15 to improve anti-PD-1 effect in this setting. Data confirmed the prognostic value of GDF15, and a trend to a synergistic effect for the anti-PD1/anti-GDF15 combination was observed in this model. In the last part, the authors examined various datasets and cohort of patients (Melanoma and oropharyngeal squamous cell carcinomas) to test the impact of GDF15 expression levels (by IHC and serum measures) on patient clinical outcomes and tumor T-cell infiltration. They show a positive association of GDF-15 with poor survival, and inverse correlation with T-cell infiltrates (CD8, CD3, Treg). They also report evidence in two series of metastatic melanoma patients that elevated GDF-15 serum levels are associated with resistance to PD-1 blockade monotherapy, suggesting GDF-15 as a novel predictive biomarker in this cancer type. The topic is of interest and original. The study provides results of relevance for the onco-immunology field. However, several technical and conceptual concerns need to be addressed.

Major comments:

- 1- in the first part of the manuscript, in vitro experiments and associated conclusions derive from T cells purified from healthy donor PBMCs. The significance would be stronger if the authors could expand the work and confirm the findings using T lymphocytes from cancer patients.

We thank the referee for this suggestion. Unfortunately, our existing ethical approval (15/18-me) does not allow us to obtain sufficient numbers of T cells from cancer patients to perform flow-adhesion assays. Flow cytometry-based ligand capture experiments with hICAM-1-Fc and stainings with m24 anti-human active LFA-1, which can be performed with small blood samples, turned out to be poorly suited to assess responses to GDF-15 - no matter whether cells were obtained from cancer patients or from healthy controls (see Figure 2f,g and also our response to Q7 by referee 1). This is now mentioned (page 8, lines 9-11).

We would further like to add that our neutralizing anti GDF-15 antibody Visugromab has successfully completed phase 1 development demonstrating excellent tolerability as a monotherapy as well as in combination with a PD-1 checkpoint inhibitor. In this dose escalation trial named GDFATHER-1 (GDF-15 Antibody-mediated Human Effector cell Relocation phase 1, NCT04725474), patients with solid tumor indications known to be difficult to treat with immune-oncology (I/O) therapy have consistently shown enhanced intratumoral T cell infiltration. Moreover, some patients who had failed on anti-PD-(L)1 therapy have shown deep and lasting tumor regressions upon treatment with visugromab + nivolumab. Despite being preliminary and not yet ready for publication, our clinical data thus show that effects of GDF-15 are not confined to healthy donor PBMC. We hope that this information can help to sort out any concerns regarding the applicability of the concept to T lymphocytes from cancer patients.

2- page 8 and Supp. Figure S2, authors claim that “GDF-15 acts on naïve and memory CD8+, and on CD4+ memory T cells”. But it is unclear how they have differentiated the memory subsets from naïve T cells. Information is missing in the method and Figure. It would be important to provide flow cytometry data to support the observed differences and statements (GDF15/anti-LFA1 treatments). Of note, in figure S2, adhesion data are shown for pan-, naïve-, Memory and EM (effector memory) CD4+ T cells. However, for CD8+ cells, the EM category is missing. Moreover, the quality of the images in panels a and b needs to be improved.

We have now performed additional experiments which have slightly changed the picture. While previously only trends had been observed with pan CD4⁺ and with naïve CD4⁺ T cells, the $p < 0.05$ level of significance has now also been reached for these subsets. Therefore, we have now changed the title into “GDF-15 broadly affects T cell adhesion to endothelial cells”. We have further indicated the kits that were used for purification (page 23, lines 5-12) and listed the markers used for negative isolation in Supplementary Figure 2. We have now also clarified that we have only analyzed the subsets obtained with >90% purity. We fully agree with the referee that CD8⁺ T_{EM} would be desirable, in particular as the gene expression database immunecellatlas.net indicates a fairly high Integrin alpha L chain/ITGAL/LFA-1 expression in this subset:

(http://immunecellatlas.net/ICA_Skyline.php?gene=ITGAL&celltype=all&organ=all&datatype=rnaseq&scale=Local).

However, reliable flow adhesion assays can only be performed with negatively selected cells. Unfortunately, there are no negative selection kits available kits for isolating the CD8⁺ T_{EM} subset. Isolating CD8⁺ Memory T Cells with subsequent depletion of CCR7⁺ cells is suboptimal, as CD57⁺ late-stage effector memory cells would be lost during the isolation of memory T cells. Similar purification issues arise with CD8⁺ effector memory RA⁺ T cells (T_{EMRA}). Therefore, the T_{EMRA} subset was omitted, too. Given that CD8⁺ memory T cells, which show high LFA-1 expression, were also highly responsive in our assays, while naïve CD4⁺ T cells, which show the lowest LFA-1 expression among T cell subsets, also turned out to be least responsive in our assays, we certainly would have liked to further explore subsets of CD8⁺ memory T cells. Still, there will always be subsets that have not been tested. Focusing on the cell types that can be obtained untouched, in sufficient quantity and with >90% purity thus appears to be the best we can do.

3- Data in figure 3 are difficult to interpret. For example, sample size and statistics in panels c, f, g make the current data questionable and hard to interpret (with 2-to-4 points in each group)

The data in panels 3c,f and g were meant to show the caveats of our initial mouse model. In our opinion, the limited information could still convey the message that anti-human GDF-15 antibodies were detected in some, but not in all mice. This observation likely explains the heterogeneity of some subsequent findings that are based on larger sample sizes. Increasing these n numbers to underline that antibody responses against human GDF-15 are heterogeneous would not have led to new conclusions or insights. Accordingly, the use of further animals did not appear justified for this purpose. Instead, we have now included two fully syngeneic mouse models to strengthen the key message of the manuscript (see new Figure 4a-i).

4- Fig 3i, j, k, tumor growth curves for each treatment are presented as various sets. It would be keen for the authors to include combined curves to help visualization and interpretation of the added value of these treatments relative to the control condition.

Data were shown as growth curves depicting each individual mouse, and as Kaplan-Meier plots showing all groups together. For clarification, the individual tumor growth curves have now been moved into the supplements (Figure S4) and standard Kaplan-Meier plots are shown. Moreover, as both the GDF-15 overexpressing MC38 as well as the newly included Panc02 tumors were treated with either isotype control, or anti-GDF-15, or anti-PD-1 or both antibodies, we have now sharpened the focus on the synergy between anti-GDF-15 and anti-PD-1 treatment and therefore omitted the anti-CD40 + poly(I:C) treatment. We hope that this more focused presentation will also facilitate the interpretation of our findings. (Please let us know if you would like us to re-integrate the data on poly(I:C) + anti-CD40 treatment into the manuscript.)

Regarding the proposed combination of curves, we agree that some aspects like the faster growth of the isotype-treated tumors and the fact that tumor-free mice are exclusively found in the group receiving the combination treatment become apparent. Still, we are afraid that the picture below might be perceived by most readers as overly crowded, in particular, when we adjust the size to the size of the other figures in our manuscript. Thus, we have combined curves to help visualization, as proposed, and we are happy to show the result below. But we are not sure whether it should actually become part of the manuscript.

Figure R9: Combined tumor growth curves from the experiment shown in Figure 3t (former Figure 3i).

5- Figure 3k and Page 10 (L14), authors claim “Adding anti-GDF-15 to anti-PD-1, however, resulted in complete clearance of MC38hGDF-15 tumors in 4/10 mice (Fig. 3k), indicating that anti-GDF-15 and PD-1-based immune checkpoint blockade synergize to increase survival in mice”. This is a key statement but this relies only on 4 events, with no significant differences for survival between the two conditions (p= 0.17).

This could be an interesting observation, but parts of the experiments should be repeated to foster the significance before to draw firm conclusions on a potential synergistic effect of their drug. Another improvement would be to assess the anti-PD-1/anti-GDF15 combination in another murine syngeneic model, as the data presented in humanized mice did not address the anti-PD-1/-GDF15 combination.

We agree that a larger number of mice would be desirable. However, the experiment now shown in Figure 3s and t already included 60 mice for overall survival analysis, and additional mice for the analysis of immune cell infiltration, which was at the limit of feasibility. We further have to accept that German laws for animal protection do not allow for experiments to be repeated. Still, having 40% versus 0% tumor clearance is, in our opinion, a remarkable result, and a very clear difference by all clinical standards. The referee is nevertheless right in asserting that Kaplan-Meier analysis (with correction for multiple testing) did not give a p value that would generally be considered significant. We have therefore performed further statistical interrogations of these data using a Cox proportional hazards model with anti-PD1 and anti-GDF-15 as explanatory variables. As animals had to be sacrificed once the tumor size exceeded 1000 mm³, an event (tumor-related death) was recorded when the termination criterion was reached. The clinically relevant question is whether adding anti-GDF-15 to anti-PD-1 is superior to anti-PD-1 monotherapy. Thus, we compared all groups against standard therapy with anti-PD-1. As expected, the mice treated with vehicle had the worst hazard ratio (HR 2.62; 95% CI 1.00 – 6.89; p=0.05). Mice treated with anti GDF-15 rather than with anti PD-1 had a non-significantly elevated hazard ratio (HR 1.39; 95% 0.50 – 3.84; p=0.52), indicating that anti PD-1 should not be replaced by anti GDF-15. However, the mice receiving the combination therapy had a significantly reduced hazard ratio (HR 0.28; 95% CI 0.08-0.96; p=0.044) compared to the mice treated with anti PD-1 monotherapy. Thus, the increased power of Cox proportional hazard models shows that adding anti-GDF-15 on top of anti-PD-1 significantly improves survival in the MC38^{tgGDF-15} model. This is now indicated in Figure 3t.

To further confirm the observed synergy, we have now performed similar experiments in the orthotopic Panc02 pancreatic cancer model. Since Panc02 cells express GDF-15 at low to moderate levels, we abstained from ectopically overexpressing GDF-15 (which we had done in the MC38 model). As the model can be performed in different ways, we once inoculated male mice with 1x10⁴ Panc02-luc cells in the head of the pancreas (new Figure 4a). The new Figure 4b shows female mice that were inoculated with 10⁶ Panc02-luc cells in the tail of the pancreas. Tumor growth was monitored by luciferase-based in vivo bioluminescence imaging. Both experiments showed that monotherapy with anti-PD-1 or anti-GDF-15 is hardly effective in most mice, whereas the combined anti-PD-1 + anti-GDF-15 treatment resulted in tumor clearance in 7/12 or, respectively, 5/12 mice (unadjusted p values: p=0.007 in a) and p=0.0205 in b) as calculated by Mann-Whitney test, adjusted p values p=0.021 in a) and p=0.063 in b), after correction for multiple testing according to Bonferroni-Holm). Thus, we thank the referee for suggesting to assess the anti-PD-1/anti-GDF15 combination in another murine syngeneic model. These new data greatly strengthen our manuscript.

Applying the anti-PD-1 + anti-GDF-15 combination to humanized mice would not be promising as these mice largely lack dendritic cells (see page 13, lines 4-7 and reference 45). Based on a personal communication from Sara Colombetti (Roche), Roche only uses humanized mouse models to study immune cell trafficking or in conjunction with agonistic antibodies/biologicals, but no longer with anti PD-L1 or other immune checkpoint blockers.

6- Figure 4, regarding analysis of T-cell infiltration (e.g. in panels f, i, j), the sample size and statistics make it hard to interpret/conclude (2-to-4 points in each condition). It's unclear why the sample size is so small, compared to the number of mice analyzed in the previous figure (10 in each treatment group). These experiments need to be confirmed with at least 6-to-10 mice per group?

Tumor-infiltrating T cells can only be analyzed from tumors that are not cleared by the immune system. Accordingly, analyzing TILs in “non-responder tumors” from experiments with responders would have introduced a massive bias. Thus, additional mice (6 per treatment group) were included to analyze immune cell infiltration at earlier time points. Unfortunately, however, MC38^{blank} cells were spontaneously rejected in some of these animals, which explains the groups with only 2 animals in the former Figure 4e,f,g (now shifted to Figure 3m-o). As the mice designated for the analysis of tumor-infiltrating leukocytes were injected last, we cannot exclude that the injected cells had been rested on ice for too long. It can nevertheless be safely assumed that the number of infiltrating T cells in eradicated tumors was at least as high as in non-eradicated ones.

To make up for this statistically underpowered experiment, we have now also analyzed TILs in Panc02 tumors on day 12 after tumor inoculation, i.e. 6 days after initiation of the antibody treatment (please see new Figure 4d,e). Here, FACS-based analysis shows that anti GDF-15 treated tumors show a trend towards increased immune infiltration by mostly CD8⁺ T cells ($p=0.0735$). Moreover, the combination treatment with anti-PD-1 + anti-GDF-15, but not the monotherapy with anti-PD-1 resulted in a significantly increased number of tumor-infiltrating CD4⁺ ($p=0.0221$) and CD8⁺ T cells ($p=0.0272$) (new Figure 4c,d). Changes in tumor-associated macrophages and intra-tumoral macrophages were, in contrast, not significant (new Supplementary Figure 5). Together with the data from the MC38^{IGHGDF-15} tumors, with the data from the humanized mouse model and the new data on T cell trafficking to tumor-draining lymph nodes in the EMT6 model, we now provide ample evidence that blocking GDF-15 increases T cell trafficking into tumors in mice.

7- Figure 4 m-n data: as indicated in the legend, CD45 and CD3 data presented in panel (m) derived from two independent experiments, and in (n), results from the second experiment only are presented to inform on the CD4 and CD8 infiltrates. However, (n) shows data from the control condition derived from 1 mouse. Therefore, it's unclear from where come the measures in (m). For clarification, it would be reasonable to show the distribution of CD4 and CD8 from the two independent experiments.

This seems to be a misunderstanding. While CD45⁺, CD3⁺ and CD19⁺ tumor-infiltrating cells were analyzed in two independent experiments, the T cell subset analysis presented in Figure 4n was only included in the second experiment. In this experiment, animals Iso1, Iso 3 and Iso 4 showed no T cell infiltrates at all. Apparently, this fact, which led to the absence of bars in these three mice, created the misleading impression that only one mouse had been analyzed (only one bar shown from four mice). We apologize for not having represented the negative T cell infiltration data from these three mice more clearly. This has now been clarified (see revised Figure 4m).

8- Page13 L20: authors claim “GDF-15 hence represents an independent marker for failure of anti-PD-1 therapy” It is not clear whether the authors have done a multivariate analysis to conclude that GDF-15 separates from other known risk factors. This would be an important

point given the poor HR and significant p values presented in Table S2. If so, other variables considered in the model should be presented with HR and p values.

We have now performed additional multivariate analyses on the Tübingen cohort, where we had additional data on age, sex, line of treatment, absence or presence of brain metastasis, serum LDH and S100B levels. In this analysis, the prognostic relevance of GDF-15 (defined by a cut-off value of 2.0 ng/ml) still yielded a p value of 0.001, and a hazard ratio of 3.312 (95% CI 1.617 – 6.783), which was only second to brain metastasis ($p < 0.001$), but superior to LDH ($p = 0.008$). S100B, age, sex and line of treatment were no significant predictors in this analysis. These data are now shown in Supplementary Figure 7.

Regarding the apparently poor HR we would like to point out that this hazard ratio indicates the change in survival probability per integer change in GDF-15 level measured in ng/ml, in a cohort with GDF-15 levels ranging from 0 ng/ml to 1640 (!) ng/ml. If the HR had been indicated per integer change in GDF-15 level measured in $\mu\text{g/ml}$, the respective value would be 1000-fold larger. Likewise, eliminating the extreme values would result in a much higher HR (compare Supplementary Table 2). As mentioned above, dichotomizing GDF-15 levels with a cut-off of 2 ng/ml would, e.g., yield a hazard ratio of 3.312 (95% confidence interval from 1.617 to 6.783), which is clearly not a poor HR. Still, as indicating the increased hazard ratio per ng/ml GDF-15 allows for a more accurate calculation of the individualized hazard ratio than a dichotomized HR, we believe that we should not change this. We have, however, also included the p value obtained when the two most extreme values are omitted (Table S2). We hope that the additional multi-variate analysis, the actual shape of the curve and the p values may now help the reader to grasp this information more easily.

9- Also, have the authors examined the potential association with PD-L1? Or test if GDF-15 is a better biomarker than PD-L1?

Using the “R2: Genomics Analysis and Visualization Platform (<http://r2.amc.nl>)”, we have performed an in silico correlation analysis to explore whether GDF-15 mRNA expression correlates with PD-L1/CD274 mRNA expression in melanoma. However, none of the available data sets showed a significant positive correlation between GDF-15 and PD-L1. The negative correlation observed in one data set is very weak ($R = -0.107$, $p = 0.020$) and unlikely to explain the strong clinical impact of GDF-15 levels.

Figure R10: Dot plots depicting GDF-15 and PD-L1 (CD274) mRNA expression in three different cohorts of patients with cutaneous melanoma.

Cohort	Identifier	platform	n	Correlation GDF-15 – PD-L1	
				R	p
Cutaneous melanoma (primary & metastatic)	ps_avgpres_tcgaskcm470_tcgars	tcgars	470	-0.046	0.319
	Cancer Genome Atlas (2015). Genomic Classification of Cutaneous Melanoma. Cell 161(7), 1681-1696.	tcagrs	375	-0.074	0.151
	ps_avgpres_tcgaskcmv32a473_gencode36	gencode36	473	-0.107	0.020
Metastatic melanoma prior to anti-PD-1	Hugo, W et al. (2016). Genomic and Transcriptomic Features of Response to Anti-PD-1 Therapy in Metastatic Melanoma. Cell 165(1), 35-44.	RNAseq	28	-0.255	0.190

This table has now been integrated in Supplementary Figure 5.

To test whether GDF-15 is a better biomarker than PD-L1, we would have needed tissue and sera from the same patient cohort. Unfortunately, no tissues had been preserved from the Tübingen cohort. From the cohort treated at Zurich university hospital, we got access to tissue blocks from 14 melanoma patients. These 14 formalin-fixed paraffin embedded (FFPE) blocks were cut into 4 um sections. Slides were stained using the automated Bond RXm system using the Bond Polymer Refine Red Detection Kit (DS9390 Leica). Antibodies were stained with the following dilutions and settings.

Antibody	Dilution	Antigen Retrieval
S100 (NCL-L-S100p, Leica)	1:600	Leica Bond ER2 (30min 95°C)
MLANA (A19-P, Novis biologicals)	1:200	Leica Bond ER2 (30min 95°C)
PD-L1 (E1L3N, Cell Signaling)	1:400	Leica Bond ER2 (20min 100°C)

Examples for the stainings (one positive and one negative example) are shown below:

MLANA_B14.37017_0.4x MLANA_B14.37017_10x

PD-L1.37017_0.4x

PD-L1_B14.37017_10x

S100.37017_0.4x

S100_B14.37017_10x

MLANA_B14.2561_0.7x

MLANA_B14.2561_10x

PD-L1.2561_0.7x

PD-L1_B14.2561_10x

S100.2561_0.7x

S100_B14.2561_10x

Figure R11: Immunohistochemical staining of melanoma sections for MLANA (Melan-A), PD-L1 and S100B.

However, as only 1 out of 14 melanomas was positive for PD-L1, a prediction of response based on PD-L1 would not have been possible in this cohort. Our data thus imply that GDF-15 was a better marker than PD-L1 in this specific cohort.

10- Figure 6: For clarity, authors should annotate with the name of each cohort and insert the sample size “in each subgroup”. Some dots have been removed in some conditions like in panel b, but it is important to know how many patients have been included for statistical analyses.

Panel 6b depicts the subset of patients classified as responders to anti-CTLA-4. By definition, this is thus a subgroup of the total anti-CTLA-4 treated cohort shown in Figure 6a. However, no dots have been removed. The only patient who was neglected for statistical analysis was a melanoma patient in Figure 6f, who was initially classified as a complete responder, but then rapidly diagnosed with further metastases. In this case, the discordance between diagnosis and clinical course raised serious doubts regarding the initial classification. Therefore, removal from statistical analysis appeared to be the most appropriate option.

11- Discussion, p16 and abstract: author claim: “We show that tumor-derived GDF-15 inhibits LFA-1 activation on human cytotoxic T cells, thereby interfering with T cell adhesion, rolling and diapedesis”.

Using the term “cytotoxic” in this sentence sounds like an overstatement as experiments were not performed to test the cytotoxic features of effector cells (cytolytic markers, cytotoxic activity of CTL or NK cells, etc...). Moreover, as mentioned above, the mechanistic part (GDF15/LFA1) was performed with peripheral T lymphocytes from healthy donors.

We thank the referee for this comment. We have now replaced cytotoxic by CD8⁺ (where appropriate) or just omitted the adjective “cytotoxic”.

12- as shown for the MC38 model (Fig. 4h), it would be useful to provide IHC staining from the humanized model to visualize the CD8 T-cell infiltrates in the isotype and anti-GDF15 conditions.

We apologize. As the complete tumors were dissected and analyzed by flow cytometry, there were no tumor samples left for IHC staining.

Minor comments:

13- figure 4 a, h: the quality of the photomicrographs should be improved. Additionally, in h, as in a, authors should indicate what immunostaining was performed.

Regarding the former Figure 4a (now 3i), we had made a mistake and confused rows and columns when we labeled the figure. This has now been corrected. We have also slightly adjusted the color/brightness on the complete images to improve the visual impression. We would also like to point out that myeloid cells give weaker signals than lymphoid cells when stained for CD45. The low number of tumor-infiltrating T cells observed in the MC38^{tgGDF-15} tumor may therefore lead to a weaker CD45 signal.

With regard to the former Figure 4h (now 3p), we have taken new photomicrographs using a higher magnification. We now provide these more detailed pictures.

14- Authors claims no relationship was found between GDF15 expression and tumor mutational burden (supp Figure 8). Since GDF-15 is a downstream target of TP53, it would be interesting

to include a comparison of the GDF-15 low/high groups in patients stratified for TP53 status and assess if it could relate to ICB response.

Unfortunately, in our two clinical cohorts the p53 status was not systematically assessed. Moreover, we have previously shown that high-level expression of wild-type p53 in melanoma cells is frequently associated with inactivity in p53 reporter gene assays (Houben, R., et al. High-level expression of wild-type p53 in melanoma cells is frequently associated with inactivity in p53 reporter gene assays. PloS one 6, e22096 (2011)), which indicates frequent post-transcriptional silencing of p53. Therefore, a proper assessment of p53 activity is far from straightforward in melanoma. However, the previously described p53-dependent induction of GDF-15 is supported by a data set from TCGA, which illustrates that GDF-15 is more abundantly expressed in p53 wild-type tumors.

Figure R12: GDF-15 mRNA expression in skin cutaneous melanoma with either mutant (left, n=49), undetermined (middle, n=57) or wild-type (right, n=269) p53.

Nevertheless, the plot already shows that GDF-15 expression varies considerably within the p53 wild-type and p53 mutant group of melanomas, with considerable overlap between these two groups. Moreover, GDF-15 expression can also be induced by numerous other transcription factors. A p53 and HIF-1 α -independent induction of p53 under hypoxia was already described by Albertoni et al. (Albertoni, M., et al. Anoxia induces macrophage inhibitory cytokine-1 (MIC-1) in glioblastoma cells independently of p53 and HIF-1. Oncogene 21, 4212-4219 (2002).). The in silico TRANSFAC analysis tool provided by QIAGEN (Kel, A.E., et al. MATCH: A tool for searching transcription

factor binding sites in DNA sequences. Nucleic Acids Res 31, 3576-3579 (2003)) indicates AP-1, ATF-2, c-Jun, C/EBPbeta, PPAR-gamma1, PPAR-gamma2 and STAT3 as the top transcription factor binding sites in the GDF15 gene promoter. GDF-15 can also be induced via the via PERK/eIF2/CHOP pathway (L'Homme, L., et al. Saturated Fatty Acids Promote GDF15 Expression in Human Macrophages through the PERK/eIF2/CHOP Signaling Pathway. Nutrients 12(2020)). A recent high-impact publication further described GDF-15 as key important mediator of Nrf2-dependent anti-inflammatory effects (Eisenstein, A., et al. Activation of the transcription factor NRF2 mediates the anti-inflammatory properties of a subset of over-the-counter and prescription NSAIDs. Immunity 55, 1082-1095 e1085 (2022)). Thus, GDF-15 expression is clearly no surrogate marker (or reporter) for p53 activity.

Moreover, p53 is an inducer of baseline and, in particular, IFN- γ inducible PD-L1 expression in melanoma (Thiem, A., et al. IFN-gamma-induced PD-L1 expression in melanoma depends on p53 expression. J Exp Clin Cancer Res 38, 397 (2019)), which is associated with an improved response to anti PD-(L)1 therapy. GDF-15, in contrast, correlates with a poor response to anti PD-(L)1 therapy. Thus, we do not believe that the effects of GDF-15 can be explained by its (weak) association with p53.

15- page 8 L15: to identify the T-cell subsets affected by GDF-15, purified immune cells were briefly exposed to GDF-15. If appropriate, authors should specify “isolated from healthy donor PBMC”.

The wording has been adjusted as suggested by the referee.

16- Figure 3h: is for survival comparison, authors should edit the Y axis legend to make it clear.

In this figure, which has now become Figure 3d in the revised manuscript, we have re-labeled the Y axis as suggested. For accuracy, we have now indicated in the legend that termination criterion is defined as tumor size > 1200 mm³.

17- discussion should be improved: for instance, authors should better highlight what is new in their study with respect to previous reports on the link between GDF-15 and tumor immune escape (ref. 23 and 27). They also need to clearly state the limitations of their study and discuss the characteristic of the cohort (e.g. Pts with Brain metastasis) relative to other types of patients treated with ICB. A discussion about the potential of GDF15 as a biomarker for prognostic or predictive.

We have done our best to improve the discussion as suggested. We have also clarified that correlations between GDF-15 and T cell infiltration in human tumors were assessed in brain metastases from melanoma, as these lesions are typically only resected once they have grown to a size that allows for a proper histopathological assessment. Other melanoma metastases are, in contrast, either removed as early as possible, or not at all (page 17, lines 14-17). However, the clinical cohorts shown in Figure 6 were not limited to patients with brain metastasis.

18- Method p23 immunohistochemistry: authors should indicate who performed and if the analyses were blinded to the treatment outcomes.

All stainings were performed on the automated IHC staining system Discovery XT and evaluated by an observer who was blinded with regard to the sample or treatment group. This is now indicated (page 28, lines 15-18).

19- Figure 5 legend and P23 method: “Formalin-fixed paraffin-embedded tissues from 70 patients with melanoma brain metastases”, it is unclear if the tissues analyzed are from brain metastases or primary tumor tissues obtained from metastatic patients.

The stained tissue specimens were obtained from surgically resected brain metastases. This has now been clarified (page 17, lines 14-17). A challenge when analyzing immune infiltration in melanoma is the fact that easily accessible metastases are resected once they become visible. On the other hand, as surgery is not curative in metastatic melanoma, difficult-to-access lesions are rarely excised. Removal of brain metastases can, in contrast, often improve the quality of life for patients with incurable disease. Therefore, brain metastases are often removed once they have grown a certain size, which makes them highly suited for analyzing T cell infiltration.

20- P1 L3,” Having established a link between GDF-15 and T cell migration in mice, we assessed GDF-15 expression”: T-cell “infiltration” or “adhesion” seems more appropriate

We thank the referee for this suggestion. We have now used the term infiltration.

21- The statement in the title « Tumor-derived GDF-15 acts as roadblock for LFA-1-dependent T-cell recruitment and suppresses response to immune checkpoint inhibition» is entirely proven since the outcome values for anti-CTLA-4 in the studied cohort were not always conclusive. Moreover, anti-CTLA-4 treatment has not been investigated in the MC38 model. Authors should consider revising the title and adjusting stated claims

We have now rephrased the title as « Tumor-derived GDF-15 blocks LFA-1-dependent T-cell recruitment and suppresses response to PD-1-based immune checkpoint inhibition», which is now also in line with the recommended maximum length (15 words) for a title.

Reviewer #3 (Remarks to the Author):

The manuscript by Haake and colleagues postulates a new mechanism by which GDF-15 contributes to immune evasion by tumor cells, namely suppression of LFA-1 dependent recruitment. They also show the potential clinical significance of this by providing data that suggest GDF-15 suppresses responses evoked by anti-PD-1 therapy. The strengths of this manuscript are its novelty given that investigations into the contribution of GDF-15 into tumor immunology are relatively rare (PMIDs 20534737, 32508832) and the use of syngeneic murine models, humanised mice models and patient samples to confirm these effects. The main weaknesses is the lack of evidence suggesting that the mechanism by which GDF-15 suppresses immune responses in vivo is T cell intrinsic and trafficking dependent, much less that this is LFA-1 specific.

Major comments

- The manuscript shows some evidence that GDF-15 affects T cell migration in vitro but there is limited evidence that a) Therapeutic effects in vivo are T cell dependent b) Effects are T cell intrinsic or c) mediated through T cell trafficking. Some evidence is presented pertaining to point c in Figure 4 but this is performed in a very limited cohort size and would need a greater n number to be convincing

We thank the referee for recognizing the strengths of this manuscript and appreciate the thoughtful advice. We have now added a second syngeneic tumor model to support our findings. The question whether effects are T cell dependent is an interesting one. GDF-15 broadly affects immune responses. Effects on dendritic cells and macrophages are therefore likely to contribute to GDF-15-mediated immune evasion. Still, the MC38 colon cancer model is paradigmatic for T cell-mediated tumor clearance (Yadav, M., et al. Predicting immunogenic tumour mutations by combining mass spectrometry and exome sequencing. Nature 515, 572-576 (2014)). Likewise, the observed synergy between anti GDF-15 and anti PD-1 also suggests an involvement of T cells, as T cells are the main effectors of anti PD-1 induced tumor cell killing. Finally, analysis of immune infiltrates upon combined GDF-15 and PD-1 blockade showed significant effects on CD4⁺ and CD8⁺ T cells in the orthotopic Panc02 model, but neither on macrophages nor on dendritic cells (Figure 4c,d, Supplementary Figure 5). Still, by causing a more M2-like polarization of macrophages and by inhibiting dendritic cells, GDF-15 can also indirectly impair T cell activity against tumors. However, to further dissect the respective contributions a cell-type specific deletion of the still unknown GDF-15 immune receptor would be required. Without knowing this receptor we can only describe effects of GDF-15 on T cells and show that GDF-15 exerts functional effects that are fully in line with the proposed mechanism. This limitation is now discussed (page 20, lines 1 – 4).

Moreover, the additional experiments performed in the syngeneic Panc02 pancreatic cancer model, where we show effects on T cell infiltration and on tumor eradication, and in the EMT6 breast cancer model, where GDF-15 affects the trafficking of adoptively transferred T cells to tumor-draining lymph nodes, substantially strengthen the revised manuscript.

- The authors provide data inferring that MC38 tumors can be further sensitized to anti-PD-1 therapy through blockade of GDF-15. However, MC38 is already quite PD-1 sensitive and therefore it would add further significance to the paper if results could be replicated in a second syngeneic model that is hardly sensitive to anti-PD-1 at baseline.

The referee is right that the MC38 model is highly sensitive to anti PD-1. This is also confirmed by our experiment with mock-transfected MC38 cells (now Figure 3s and Supplementary Figure 4c). In fact, as wild-type MC38 cells express hardly any GDF-15, blocking GDF-15 would make little sense with these cells. However, we then show that overexpression of hGDF-15 renders the MC38 model almost completely resistant to anti PD-1 treatment, which can be (partly) reversed with an anti GDF-15 antibody (Figure 3t and Supplementary Figure 4d). Using an intrinsically PD-1 resistant tumor model would not have enabled us to show that ectopically expressed GDF-15 can protect a tumor against anti PD-1 treatment. However, while murine tumors generally express much less GDF-15 than human ones, we have now generated additional data to show that the effect of anti PD-1 treatment is synergistically enhanced by anti GDF-15 in the hard-to-treat orthotopic Panc02 tumor model (new Figure 4a,b). So we now show that a PD-1-sensitive, GDF-15^{low} tumor can be rendered PD-1 resistant by overexpressing GDF-15, which can be reverted by blocking GDF-15, and we also show that a PD-1 resistant, GDF-15^{high} tumor can be sensitized to anti-PD-1 treatment by blocking GDF-15.

- One point of concern is that there appears to be major inconsistencies in tumor growth between experiments even when the same conditions are tested. For example in Figure 3H approximately 50% of parental MC38 tumors are cleared but in 3i this is 0%. Similarly in Figure 3J MC38tgGDF-15 tumors (vehicle treated) uniformly reach 2000mm³ by approximately day 25 but this appears to be <500mm³ in 3K. I could not see a reason for these differences, can this be explained?

We apologize for any potential misunderstanding, but we have not observed 50% tumor clearance in the former Figure 3h (which has now become 3d). We have just observed a significantly slower tumor growth of MC38^{blank} compared to MC38^{ghGDF-15} tumors in immunocompetent mice, with a 100% tumor take rate in this specific experiment (please compare tumor growth curves in Supplementary Figure 4b). Strikingly, these sublines showed an opposite growth behavior in immunodeficient mice. Still, the referee is right that growth rates differed between different experiments. As these experiments were performed by a contracted CRO (Charles River US), we cannot tell whether these differences were linked to differences regarding the fitness of cells, the person performing the experiment, or to microbiota, where minor changes in the animal facility can have strong effects in immunocompetent tumor models. However, as we have always included the relevant controls in the same experiment, we believe that this biological variation does not affect the validity of our data. Seeing that GDF-15 dependent effects persist despite certain differences between individual experiments rather indicates to us that the effects are robust.

- In Figure 5 and 6 the authors present very interesting patient data that indicates patients with elevated GDF-15 have poorer overall survival. Of course with any such dataset one must keep in mind responses may be correlative but not necessarily causative. With this in mind, to what extent do GDF-15 levels correlate with TGFbeta (part of the same family and known to lead to an immune excluded phenotype) and does this explain these observations in patients?

We thank the referee for this valuable remark. There is indeed a list of publications that wrongly assigned effects of TGF- β to GDF-15. This was mainly due to the presence of contaminating TGF- β in preparations of recombinant GDF-15 (Olsen, O.E., Skjaervik, A., Stordal, B.F., Sundan, A. & Holien, T. TGF-beta contamination of purified recombinant GDF15. PloS one 12, e0187349 (2017), reference 54). However, the

sequence identity in the mature protein is just 21%. Thus, GDF-15 is more dissimilar to TGF- β than often appreciated. With regard to tissue expression in cancer, we have no data on TGF- β in the clinical cohorts available for our study. However, potential correlations can be easily tested based on RNAseq data available in repositories. Using the “R2: Genomics Analysis and Visualization Platform (<http://r2.amc.nl>)”, we have thus performed in silico correlation analyses to test whether GDF-15 mRNA expression correlates with TGFB1 or TGFB2 mRNA expression in melanoma. As three cohorts with $n \geq 375$ samples each showed unanimously weak negative correlation coefficients between GDF-15 and TGFB1 (correlation coefficients between -0.058 and -0.106 for TGFB1, between -0.084 and -0.126 for TGFB2), there is certainly no co-regulation of GDF-15 and TGF- β 1 or TGF- β 2. A smaller fourth cohort with metastatic melanoma patients prior to anti-PD-1 treatment (Hugo, W., et al. Genomic and Transcriptomic Features of Response to Anti-PD-1 Therapy in Metastatic Melanoma. Cell 165, 35-44 (2016)), i.e. a patient population very similar to our two cohorts, gave similar results, albeit with slightly higher negative R values. Thus, it is extremely unlikely that our observation that GDF-15 excludes T cells and correlates with failure of anti PD-1 therapy would be caused by an (overlooked) association between GDF-15 and TGF- β expression. This is now discussed (page 17, lines 11-14) and a key reference describing TGF- β -mediated T cell exclusion is now cited (Tauriello, D.V.F., et al. TGFbeta drives immune evasion in genetically reconstituted colon cancer metastasis. Nature 554, 538-543 (2018), new reference 55). We have now also added two plots and a table showing the lack of positive association between GDF-15 and TGF- β 1/2 gene expression to Supplementary Figure 8. Please find the plots from all four investigated cohorts below.

Figure R13: Dot plots depicting GDF-15 and TGFb1 or TGFb2 mRNA expression in four different cohorts of patients with cutaneous melanoma.

- The authors should discuss why they chose to overexpress human GDF-15 in the mouse model as opposed to murine GDF-15. The immune response mounted against human GDF-15 is clearly a caveat of the current work.

While we had to recognize that the immune response mounted against human GDF-15 is indeed a caveat, the choice of model enabled us to use our own antibody, which has now entered a phase 2 clinical trial, to block GDF-15. From a translational perspective, this was an important advantage. Moreover, the strong immunosuppressive effect of human GDF-15 in mice is supported by the observation that only 50% of the mice developed measurable antibody responses against human GDF-15. Apparently, the immunosuppressive function of GDF-15 was thus sufficient to prevent endogenous antibody formation against human GDF-15 in 50% of the mice. Nevertheless, we have now included further syngeneic animal models based on endogenous GDF-15 (see new Figure 4a-i), and used an anti-murine GDF-15 antibody. While this antibody is - from a pre-clinical development perspective - just a surrogate, it circumvents the problem. As the new data are fully in line with our previous findings and interpretations, we have now strengthened our study by including data from experimental systems that do not include this caveat.

Minor comments

LFA-1 is also important for T cells binding to tumor cells and antigen presenting cells. Did the authors consider/ evaluate whether these parameters are affected by GDF-15? Simple in vitro analyses could be performed e.g. killing assays, conjugate assays to address this.

Effects of GDF-15 on antigen-presenting cells have already been demonstrated by others (Zhou, Z., et al. Growth differentiation factor-15 suppresses maturation and function of dendritic cells and inhibits tumor-specific immune response. PloS one 8, e78618 (2013), reference 24, Zhou et al. 2013, and Zhang, Y., et al. GDF15 Regulates Malat-1 Circular RNA and Inactivates NFkappaB Signaling Leading to Immune Tolerogenic DCs for Preventing Alloimmune Rejection in Heart Transplantation. Front Immunol 9, 2407 (2018), reference 26), and we are also working to figure out the precise role of GDF-15 on LFA-1 activity in antigen-presenting cells. However, simple in vitro assays have already been addressed in the aforementioned manuscripts, while a detailed consideration of the effect of GDF-15 in antigen-presenting cells is a complex issue that deserves consideration in a separate manuscript.

In terms of the T cell intrinsic nature of the in vivo effects, the authors indicate that the GFRAL receptor is not expressed on lymphocytes but the mechanism by which they are affected is not really discussed. Could the authors elaborate on this topic.

There is mounting evidence that a second GDF-15 receptor must be expressed on immune cells. This has been discussed by Lockhart and O'Rahilly (Lockhart, S.M. & O'Rahilly, S. Colchicine—an old dog with new tricks. Nature Metabolism 3, 451-452 (2021)) in their editorial on Weng et al. (Weng, J.H., et al. Colchicine acts selectively in the liver to induce hepatokines that inhibit myeloid cell activation. Nat Metab 3, 513-522 (2021), new reference 26). These authors show that that an immunoreactive GDF-15 moiety in plasma mediates the anti-inflammatory effects of colchicine. Importantly, this effect of GDF-15 is still active in GFRAL knock-out mice. Weng et al. have thus

provided clear evidence for an additional GDF-15 receptor, without actually identifying it. While we are skeptical regarding recent reports that GDF-15 might signal via CD44 on dendritic cells (Gao, Y., et al. Growth differentiation factor-15 promotes immune escape of ovarian cancer via targeting CD44 in dendritic cells. Exp Cell Res 402, 112522 (2021)) or via CD48 on regulatory T cells (Wang, Z., et al. GDF15 induces immunosuppression via CD48 on regulatory T cells in hepatocellular carcinoma. J Immunother Cancer 9(2021), new reference 25), our data also point to the existence of a yet unknown immune receptor for GDF-15. This is now discussed (page 19, lines 8-13). Please see also our response to question 11 by Referee 1, where we show that a mutant GDF-15 which no longer binds to GFRAL can still impair T cell adhesion to endothelia, while a neutralizing anti-GFRAL antibody has no effect on GDF-15-mediated adhesion inhibition. However, without having identified the immune receptor for GDF-15, we do not want to indulge in undue speculation. Showing our gene expression data and citing the relevant literature allows us to safely conclude that “Immunomodulatory effects of GDF-15 ... appear to be independent of GFRAL” (page 19, lines 12-13), which is as much as we can currently say.

Figure 2a/b the staining is difficult to see. Is it possible to provide more convincing images for this effect?

We have tried to optimize the contrast in these picture. However, as the bright dots are from individual LFA-1/ICAM-1 or LFA-1/m24 binding pairs (on a single molecule basis), we are already at the current limit of resolution for super-resolution microscopy. Moreover, the effect of GDF-15 on ICAM-1/LFA-1 or on m24/LFA-1 binding is very weak. Still, in this revised version of the manuscript we now show that GDF-15 may have a lesser effect on ICAM-1/LFA-1 binding than on the signal transmission into the cell and towards the actin cytoskeleton. We hope that this clarifies the issue.

REVIEWER COMMENTS

Reviewer #1 (Remarks to the Author):

The authors made a very elaborated effort and extensively revised the MS. The textual revisions are very helpful in particular the discussion points that the effects of GDF-15 on anti cancer immunity are beyond its direct role on inhibition of LFA-1 adhesiveness.

As for specific experiments I asked for- the combinatorial in vitro inhibition studies with anti LFA-1 and GDF-15 are convincing and the explanations on the use of human GDF-15 in murine systems are helpful. Some of the figures addressed to the reviewers (and listed as R1-R13) should be incorporated as supplementary figures available to the readers. In particular I would strengthen the new parts that implicate GDF-15 as a multi-functional suppressor of anti-tumor immunity beyond its role in lymphocyte trafficking (see please Figures R3, R8). I would also recommend a slight change in the title (responses instead of response).

Minor:

In the discussion, the authors suggest that interference with the LFA-1-ICAM-1 axis inhibits rolling. This is wrong, at least for T cells. LFA-1-ICAM-1 is mainly involved in post rolling arrest, crawling and diapedesis across activated blood vessels.

Reviewer #2 (Remarks to the Author):

The authors have adequately responded to most of my comments. However, they were unable to address some concerns for technical reasons or limited access to biological samples. Nevertheless, they should:

Major concerns

1- Discuss why flow cytometry failed to show an effect of GDF-15 on the binding of the conformation-specific anti-active-LFA-1 antibody mAb24 (Figure 2f) or an Fc-tagged ICAM-1 complex (Figure 2g) to CD8+ T cells from healthy donors or cancer patients.

2- Perform additional experiments using positive selection of T lymphocytes with anti-CD8 or anti-CD4 (not anti-CD3) antibodies to limit interference on the results.

3- The authors should be more rigorous in setting their mouse model and interpreting their results. They should also discuss the origin of heterogeneity in anti-human GDF-15 antibodies from one mouse to another.

4- OK

No need to include the data with poly(I:C) + anti-CD40 treatment.

5- OK with the Cox model and the additional Panc02 pancreatic cancer model.

6- Mice that spontaneously reject MC-38 tumors should not be recorded, therefore larger groups of mice need to be included. Moreover, the authors have to provide a more convincing interpretation of the observed results. The explanation that "the injected cells had been rested on ice for too long" is not acceptable. Similarly, "... safely assumed that the number of infiltrating T cells in eradicated tumors was at least as high as in non-eradicated ones" is not acceptable.

7- Again, the number of experiments and number of mice in each experiment is too limited to obtain strong and statistically convincing results.

8- OK with the multivariate analysis.

9- OK with Supplementary Fig. 5 and integration of the negative, but informative, correlation.

10- The authors should be more precise with the patient subsets and explain why they removed a

particular patient for statistical analysis reason.

11- OK with CD8 T cells.

12- IHC staining could have been performed on tumors from additional mice, not necessarily from the same groups of mice than FACS.

Minor comments:

13- OK with the correction of the mistake in Fig. 3i and with the new photomicrographs in Fig. 3p.

14- Authors may briefly discuss these statements in the discussion section.

15- OK with the wording.

16- Ok with Figure 3d relabeling.

17- OK with the new version of the discussion, which could be completed as indicated above.

18- OK with immunohistochemistry staining and analysis.

19- OK with FFPE brain metastasis samples.

20- OK with T-cell infiltration.

21- OK with the new title.

Reviewer #3 (Remarks to the Author):

I thank the authors for addressing all questions, the manuscript is now significantly improved.

Point-by-point reply to the reviewer comments

Reviewer #1 (Remarks to the Author):

The authors made a very elaborated effort and extensively revised the MS. The textual revisions are very helpful in particular the discussion points that the effects of GDF-15 on anti cancer immunity are beyond its direct role on inhibition of LFA-1 adhesiveness.

As for specific experiments I asked for- the combinatorial in vitro inhibition studies with anti LFA-1 and GDF-15 are convincing and the explanations on the use of human GDF-15 in murine systems are helpful. Some of the figures addressed to the reviewers (and listed as R1-R13) should be incorporated as supplementary figures available to the readers. In particular I would strengthen the new parts that implicate GDF-15 as a multi-functional suppressor of anti-tumor immunity beyond its role in lymphocyte trafficking (see please Figures R3, R8). I would also recommend a slight change in the title (responses instead of response).

We thank the referee for these thoughtful and positive comments. We have now incorporated Figure R8 into Figure 1 (as new Figure 1f). Figure R3 has been incorporated as Supplementary Figure 5c. Also the title has been changed as proposed.

Minor:

In the discussion, the authors suggest that interference with the LFA-1-ICAM-1 axis inhibits rolling. This is wrong, at least for T cells.

This has been corrected. The sentence (page 17, lines 3-6) now reads "Here we show that GDF-15 also prevents a stable, Talin-dependent linkage between LFA-1 and the actin cytoskeleton. Tumor-derived GDF-15 thus inhibits the LFA 1:ICAM 1 axis in human T cells, thereby interfering with T cell post rolling arrest, firm adhesion to endothelia, crawling and diapedesis across activated blood vessels."

Reviewer #2 (Remarks to the Author):

The authors have adequately responded to most of my comments. However, they were unable to address some concerns for technical reasons or limited access to biological samples. Nevertheless, they should:

Major concerns

1- Discuss why flow cytometry failed to show an effect of GDF-15 on the binding of the conformation-specific anti-active-LFA-1 antibody mAb24 (Figure 2f) or an Fc-tagged ICAM-1 complex (Figure 2g) to CD8+ T cells from healthy donors or cancer patients.

We apologize if our explanation was not clear enough. On page 8, lines 14-16, we have now written: "Adhesion, however, requires ligand binding and a stable linkage of LFA-1 to the actin cytoskeleton. The latter is triggered by mechanical tension resulting from interactions with immobilized (but not soluble) ICAM-1³⁷." As in the previous version, we also refer to Nordenfelt, P., Elliott, H.L. & Springer, T.A. Coordinated integrin activation by actin-dependent force during T-cell migration. Nat Commun 7, 13119 (2016), where the authors describe that an open headpiece– which is indicated via binding assays of soluble ICAM-Fc or conformation-specific antibodies – in itself is not sufficient for integrins to bear adhesion forces. In addition, their cytoplasmic domain must be linked to Talin and the actin cytoskeleton. While all intracellularly cross-linked integrins appear to have an open headpiece, intracellular tethering of the cytoplasmic domain to Talin and the actin cytoskeleton occurs in a second step triggered when shear forces act on the ligated extracellular domain. The most likely explanation for our data is that GDF-15 blocks this second step.

In lines 18-23 (unchanged) we further write: "As phosphorylation is required to relieve Talin auto-inhibition³⁸, GDF-15-treated T cells cannot establish a stable link between LFA-1 and the cytoskeleton³⁹. GDF-15 thus modulates adhesion of T cells to the endothelium mainly by interfering with the intracellular stabilization of extracellular LFA-1:ICAM-1 interactions."

Page 17, lines 3-6 now read: "Here we show that GDF-15 also prevents a stable, Talin-dependent linkage between LFA-1 and the actin cytoskeleton. Tumor-derived GDF-15 thus inhibits the LFA-1:ICAM-1 axis in human T cells, thereby interfering with T cell post rolling arrest, firm adhesion to endothelia, crawling and diapedesis across activated blood vessels."

Experimentally, a capillary-based protein analysis has further confirmed the inhibitory effect of GDF-15 on Talin1 phosphorylation.

Figure R1: Surplus protein samples from stimulated and GDF-15- or vehicle-treated human T cells (as analyzed by conventional Western Blotting in Figure 2k) were run on the SimpleWestern system Jess (). Samples were probed with anti-Phospho-Talin^{Ser42} (D2P2M) Rabbit mAb #13589 (Cell Signaling Technologies) in the capillaries and normalized for total protein content.

Importantly, Kempf et al. (Nature Medicine 2011) have already described that GDF-15 inhibits LFA-1 activation by interfering with Rap1-GTP formation, which acts upstream of Talin activity (Lagarrigue et al., Talin-1 is the principal platelet Rap1 effector of integrin activation, Blood 2020 Sep 3;136(10):1180-1190.doi: 10.1182/blood.2020005348). Thus, our data are in good agreement with their findings. Still, we could not reproduce the reduction in mAb24 binding to chemokine-activated human PMN shown in Figure 5a of their publication.

We would also like to point out that these binding assays are strongly dependent on incubation times and temperature. Initially, we had also obtained some results suggesting an effect of GDF-15 on mAb24 or ICAM-1-Fc binding, with some data pointing in the opposite direction. In order to nail down the positive “findings”, we meticulously standardized all assay conditions. After applying strict temperature control, only very small effects compatible with affinity data published by Nordenfelt, P., Elliott, H.L. & Springer, T.A. Coordinated integrin activation by actin-dependent force during T-cell migration. Nat Commun 7, 13119 (2016) remained. Due to the inevitable spread of FACS peaks, these small differences required dSTORM microscopy for quantification.

Given the word limit and the interest of the average reader, we cannot include this extended discussion in the manuscript. Still, we hope that the additional information helps the referee to better understand how we have arrived at our conclusion. Most importantly, we hope that the revised version now also clarifies the issue for our readers.

2- Perform additional experiments using positive selection of T lymphocytes with anti-CD8 or anti-CD4 (not anti-CD3) antibodies to limit interference on the results.

In the first revision we have added the requested information on how we have differentiated memory subsets from naïve T cells, and we have increased the n number for the subsets that initially failed to achieve the level of significance ($p < 0.05$). This has confirmed that GDF-15 acts very broadly on different T cell subsets. The only still open request was the investigation of the

CD8⁺ T_{EM} subset, for which there is no suitable purification kit available. We have now solved this by first purifying memory CD8⁺ T cells, and subsequently depleting CCR7⁺ cells, which results in a CD8⁺ T_{EM} population without CD57⁺ late stage effector cells. This assay now been added as Supplementary Figure 2h. Thereby, we have now fulfilled all requests raised during the initial review.*

The new request to also use positively selected cells for flow adhesion experiments entails a caveat. Using either human CD8⁺ micro beads (Mitenyi Biotec #130-045-201) for positive selection, or the human CD8⁺ T cells selection kit (Mitenyi Biotec #130-096-495) for negative selection, CD8⁺ T cell purity (91% with #130-045-201, 93% with #130-096-495) and the proportion of CD45RO⁺ memory CD8⁺ T cells purity (37% with #130-045-201, 40% with #130-096-495) were very similar. Despite small differences with LFA-1 blockade, adhesion data were also quite similar for negatively and for positively selected CD8⁺ T cells. However, beads that remain attached to positively selected cells largely abrogate transmigration. Given that our manuscript explores effects of GDF-15 on T cell infiltration into a tumor tissue, which requires adhesion and transmigration, we believe that we should stick with negatively selected cells.

Figure R2: Negatively or positively isolated human CD8⁺ T cells were run over a layer of activated human human umbilical vein endothelial cells (HUVEC). T cells adhering (left) and transmigrating (right) under hydrodynamic flow conditions were quantified under a fluorescent microscope.

We would further like to point out that this supplementary figure is only meant to show that GDF-15 broadly affects the adhesion properties of T cell subsets. A differential analysis of GDF-15 effects on individual cell types is an interesting question that could be addressed in a future manuscript.

** Original comment by Referee 2: “page 8 and Supp. Figure S2, authors claim that “GDF-15 acts on naïve and memory CD8+, and on CD4+ memory T cells”. But it is unclear how they have differentiated the memory subsets from naïve T cells. Information is missing in the method and Figure. It would be important to provide flow cytometry data to support the observed differences and statements (GDF15/anti-LFA1 treatments). Of note, in figure S2, adhesion data are shown for pan-, naïve-, Memory and EM (effector memory) CD4+ T cells. However, for CD8+ cells, the EM category is missing. Moreover, the quality of the images in panels a and b needs to be improved.”*

3- The authors should be more rigorous in setting their mouse model and interpreting their results. They should also discuss the origin of heterogeneity in anti-human GDF-15 antibodies from one mouse to another.

While rigor in designing and interpreting results is indeed key to successful preclinical research, we proudly point out that our preclinical work has been translated into phase 1 and phase 2 clinical trials in human cancer patients. Based on our data we predicted that GDF-15 neutralization can enhance current aPD-1 therapies. This was confirmed in patient biopsies by increased intratumoral T cell infiltration, and by some impressive clinical responses in heavily pre-treated, aPD-1-refractory last-line patients with solid tumors. Please find a brief summary of early clinical trial data in https://www.catalym.com/wp-content/uploads/2022/09/220910_ESMO_CTL-002-001-Phase-1-Trial-presentation_MELERO_v1.0.pdf

While we cannot include the data generated by a large group of clinical investigators to the present pre-clinical manuscript (which is already filled with data), there can be no better validation of pre-clinical work than its successful translation into patients.

Moreover, protection of animals enjoys legal and constitutional status in Germany, and regulations are becoming continuously stricter. We are only allowed to use animals for research when there is no other way of obtaining the desired result. Once a question has been addressed in an animal experiment, a permission to repeat this experiment will not be granted again. Instead, authorities expect investigators to draw all conclusions from the already available animal data. We cannot ignore and overcome this legislation, which is clearly not in favor of science. Accordingly, we sometimes have to accept that unanticipated problems like poor engraftment of tumor cells in a group of mice may result in less-than-perfect results. Therefore, we have now written (page 10, line 24 to page 11, line 2) “While the number of tumors that could be harvested and assessed by flow cytometry after day 23 was too small to allow for valid conclusions, effects on immune cell infiltration still showed a trend (Figure 3m-o).” We believe that this is an apt description of these admittedly less-than-ideal data. Still, with highly concordant data from four different animal models (transgenic sub-cutaneous MC38 colon cancer, orthotopic Panc02 pancreatic cancer, orthotopic EMT6 breast cancer, and humanized patient-derived xenograft models), detailed mechanistic data and early clinical data that all confirm our hypothesis, the conclusions drawn in the manuscript are extremely well substantiated.

Regarding the heterogeneous development of endogenous antibodies, we have now written on page 10, lines 15-17: “Still, immunosuppressive effects of human GDF-15 likely prevented antibody formation in 50% of the mice, and immunosuppressive effects of human GDF-15 more than outweighed its immunogenicity in mice.”

4- OK

No need to include the data with poly(I:C) + anti-CD40 treatment.

Thank you.

5- OK with the Cox model and the additional Panc02 pancreatic cancer model.

Thank you.

6- Mice that spontaneously reject MC-38 tumors should not be recorded, therefore larger groups of mice need to be included. Moreover, the authors have to provide a more convincing interpretation of the observed results. The explanation that “the injected cells had been rested on ice for too long” is not acceptable. Similarly, “... safely assumed that the number of infiltrating T cells in eradicated tumors was at least as high as in non-eradicated ones” is not acceptable.

While we agree with the referee that mice should not be recorded when a tumor does not engraft, we urge the referee to take a closer look at Supplementary Figure 4d:

Here, it is clearly visible that several tumors were rejected after engraftment. Excluding these animals would constitute a selection on non-responders and thereby introduce a bias. As rejection of MC38 tumors depends on tumor-infiltrating T cells, it was never observed in T cell deficient mice. We thus maintain that such tumors were evidently infiltrated by T cells. Ignoring these animals would therefore introduce a substantial bias.

Still, the referee is absolutely right that the small number of mice in some groups is clearly suboptimal, and we have also acknowledged this in the manuscript. Page 10, line 24 to page 11, line 2: “While the number of mice that could be assessed by flow cytometry after day 23 was too small to allow for actual conclusions, effects on immune cell infiltration still showed a trend (Figure 3m-o).”

However, as outlined above we are not free to perform animal experiments whenever we consider them useful. We have to accept the limitations imposed by law, which does not allow for repeat experiments. We therefore also discussed with the governmental review board of the State of Bayern, Regierung von Unterfranken. Still, from these discussions it became clear that repeating the whole set of animals with MC38^{tg^hGDF-15} tumors will not be approved. Scientifically, however, an addition of individual animals to selected groups is no valid option. As immune responses depend on the microbiome and on the condition of the mouse facility, only littermates must be compared within one experiment. Therefore, we cannot optimize in a way that would be scientifically valid and compatible with the rules for animal experimentation.

Still, the data we obtained from the transgenic model show unambiguously that expression of GDF-15 confers a survival advantage to tumor cells grown in immune competent, but not in immunodeficient animals. The model further shows that GDF-15 reduces CD8⁺ T cell infiltration in tumors, which can be reverted by a neutralizing anti-GDF-15 antibody (which also shows that the antibody works in vivo). This effect on T cell infiltration was subsequently confirmed in other models (see Figures 4c-e, Figures 4f,g, Figure 4l) and supported by correlations obtained using human tumor tissues (Figure 5). Data from humans treated with a neutralizing anti-GDF-15 antibody has been presented at ESMO2022 (not shown in this manuscript). The ability of GDF-15 to exclude T cells from the tumor microenvironment is thus very well supported by data. Accordingly, GDF-15 overexpression protects MC38 tumors from anti-PD-1-mediated clearance. Again, the effect of GDF-15 can be reverted by a neutralizing anti-GDF-15 antibody. All these effects are statistically significant and relevant. However, the confounding factor that has shown up in these experiments, namely the development of endogenous antibodies against human GDF-15 in some mice, would likely be observed again if the same experiment was repeated and expanded. Therefore (and due to the legal framework for animal experimentation), we had to find a work-around. Initially, we circumvented the problem by using a humanized patient-derived xenograft model. In the course of the first revision, we then performed additional experiments in an orthotopic Panc02 pancreatic cancer and an orthotopic EMT6 breast cancer model – and obtained concordant results in all four models. In our opinion, showing that GDF-15 acts as a “T cell repellent” in several different models is even more convincing than demonstrating the effect in a maximally optimized model. Importantly, the other two referees who also felt the need for additional animal data, have accepted our strategy as a valid approach to support our hypothesis.

7- Again, the number of experiments and number of mice in each experiment is too limited to obtain strong and statistically convincing results.

In the first response to the referees, we had already outlined that the referee’s perception of the animal number was wrong. We apologize if we have not been clear enough in indicating that several mice treated with the isotype antibody showed no detectable tumor-infiltrating human T cells at all and were therefore difficult to display. Moreover, statistical analysis of tumor-infiltrating immune cells (Figure 4l, unchanged, based on 9 mice per group) yielded p values of 0.029 for CD45⁺ and 0.012 for CD3⁺ T cells. B cells, which are less dependent on LFA-1 for trafficking, were assessed as a negative control (p=0.718). At the p<0.05 level of significance, the results for CD45⁺ and CD3⁺ cells are statistically convincing. The lack of impact on B cell infiltration was expected. We hope this clarification solves the issue. By calculating the overall percentage of CD4⁺, CD8⁺, CD4⁺CD8⁺ double-positive and CD4⁻CD8⁻ double-negative T cells in the disseminated tumors, we have now also been able to perform additional statistics. While no clear data were obtained for double-positive or double-negative T cells (p values 0.20 and 0.62, respectively), treatment with anti-GDF-15 antibody resulted in a statistically significant enrichment of tumor-infiltrating CD4⁺ T cells (p=0.037). For CD8⁺ T cells, the level of significance was missed by a narrow margin (p=0.069, using Mann-Whitney-U test). However, as CD8⁺ T cells were found in 4/4 anti-GDF-15-treated, but only in 1/4 isotype antibody-treated tumors, these results clearly support the findings from other models. Additional animal experiments would thus not be justified. Moreover, humanization of mice based on the chosen protocol exceeds the time foreseen for this revision.

Finally, we refer once again to the clinical data obtained with GDF-15 blockade in last-line patients (https://www.catalym.com/wp-content/uploads/2022/09/220910_ESMO_CTL-002-001-Phase-1-Trial-presentation_MELERO_v1.0.pdf). We hope that we will soon be able to share this additional validation with the broader scientific community.

8- OK with the multivariate analysis.

Thank you.

9- OK with Supplementary Fig. 5 and integration of the negative, but informative, correlation.

Thank you.

10- The authors should be more precise with the patient subsets and explain why they removed a particular patient for statistical analysis reason.

*One single patient was removed from the statistical comparison of GDF-15 serum levels between different response groups (Figures 6f,g,h). The reason were discordant data between clinical staging and clinical course. Though this patient was staged as a complete responder, he developed symptomatic metastases within weeks after the staging and died from melanoma soon thereafter – which strongly suggests that his initial response was less than complete. Unfortunately, we cannot retrospectively resolve this discrepancy. Therefore, we cannot confidently assign this patient to a response group. (CR would most likely be wrong, though compatible with the staging result. Any other assignment would signify a retrospective change of the staging result, which would not be correct, either.) In the Figure legend we have now written “In **f,g,h**, one patient whose clinical course contradicted the classification as complete responder was omitted from statistical consideration” (previous version “disregarding one misidentified complete responder”, page 52, lines 4-5). In the Materials and Methods part, section on statistics, we have now added “Due to a discordance between staging and clinical course, one patient was neglected in the statistical analyses for Figures 6f,g,h” (page 36, lines 18-19). In the extended Figure legends we had already written: “One patient who relapsed and died from melanoma despite being classified as complete responder was treated as on outlier in the CR group and thus disregarded” (page 71, lines 15-17). We have now added the information that this patient died within months from melanoma.*

Still, we thank the referee for directing our attention again towards this detail, as excluding this patient only from statistical analysis in Figure 6f was inconsistent. As the analyses in Figures 6g and h are also based on clinical staging, we have now also excluded this patient from these analyses. In the corresponding statistics odds ratios were thus reduced from 0.796 to 0.778 in Figure 6g and from 0.782 to 0.773 in Figure 6h, while p values changed from 0.0347 to 0.0315 in Figure 6g and from 0.00906 to 0.0085 in Figure 6h. Thus, results are hardly affected by our decision – which is not surprising as just one of 88 patients in this cohort, and none of the 35 patients in our second cohort showed such discordant data. Still, we are convinced that this is the best way of dealing with the contradictory data, and we are now consistent in excluding the questionable staging result. As we would not exclude a patient without good reason, we have still included this patient in the analysis of overall survival, where data were unambiguous.

11- OK with CD8 T cells.

Thank you.

12- IHC staining could have been performed on tumors from additional mice, not necessarily from the same groups of mice than FACS.

As the absolute number of tumor-infiltrating CD8⁺ T cells is quite low in this humanized mouse model, immunohistochemical analyses suffer from a substantial sampling bias. While we had stained some sections when setting up the model, flow cytometry-based analysis of dissected tumors yields much more representative and therefore also more reliable data. Thus, we have now written on page 13, lines 10-11: "Due to the overall low TIL numbers and the resulting sampling bias, immunohistochemical analyses were inferior to flow cytometry-based infiltration assessments."

Minor comments:

13- OK with the correction of the mistake in Fig. 3i and with the new photomicrographs in Fig. 3p.

Thank you.

14- Authors may briefly discuss these statements in the discussion section.

Thank you.

15- OK with the wording.

Thank you.

16- Ok with Figure 3d relabeling.

Thank you.

17- OK with the new version of the discussion, which could be completed as indicated above.

Thank you.

18- OK with immunohistochemistry staining and analysis.

Thank you.

19- OK with FFPE brain metastasis samples.

Thank you.

20- OK with T-cell infiltration.

Thank you.

21- OK with the new title.

Thank you.

Reviewer #3 (Remarks to the Author):

I thank the authors for addressing all questions, the manuscript is now significantly improved.

Thank you.

REVIEWERS' COMMENTS

Reviewer #2 (Remarks to the Author):

The authors have now addressed all my concerns and improved their manuscript accordingly.

Point-by-point reply to the reviewer comments

Reviewer #2 (Remarks to the Author):

The authors have now addressed all my concerns and improved their manuscript accordingly.

We thank the referee for approving our efforts to revise the manuscript according to his/her suggestions.

As no further comments from referees were transmitted to the authors, we assume to have adequately responded to all of the referees' requests.